# An Optimal and Scalable Matrix Mechanism for Noisy Marginals under Convex Loss Functions

**Yingtai Xiao**
Penn State University
yxx5224@psu.edu

**Guanlin He**
Penn State University
gbh5146@psu.edu

**Danfeng Zhang**
Penn State University
dbz5017@psu.edu

**Daniel Kifer**
Penn State University
duk17@psu.edu

## Abstract

Noisy marginals are a common form of confidentiality-protecting data release and are useful for many downstream tasks such as contingency table analysis, construction of Bayesian networks, and even synthetic data generation. Privacy mechanisms that provide unbiased noisy answers to linear queries (such as marginals) are known as matrix mechanisms.

We propose ResidualPlanner, a matrix mechanism for marginals with Gaussian noise that is both optimal and scalable. ResidualPlanner can optimize for many loss functions that can be written as a convex function of marginal variances (prior work was restricted to just one predefined objective function). ResidualPlanner can optimize the accuracy of marginals in large scale settings in seconds, even when the previous state of the art (HDMM) runs out of memory. It even runs on datasets with 100 attributes in a couple of minutes. Furthermore ResidualPlanner can efficiently compute variance/covariance values for each marginal (prior methods quickly run out of memory, even for relatively small datasets).

## 1 Introduction

Marginals are tables of counts on a set of attributes (e.g., how many people there are for each combination of race and gender). They are one of the most common formats for the dissemination of statistical data [8, 2], studying correlations between attributes, and are sufficient statistics for loglinear models, including Bayesian networks and Markov random fields. For this reason, a lot of work in the differential privacy literature has considered how to produce a set of noisy marginals that is both privacy-preserving and accurate.

One line of work, called the *matrix mechanism* [32, 52, 30, 53, 37, 51, 46, 18, 42] designs algorithms for answering linear queries (such as marginals) so that the privacy-preserving noisy answers are accurate, unbiased, and have a simple distribution (e.g., multivariate normal). These crucial properties allow statisticians to work with the data, model error due to data collection (sampling error) and error due to privacy protections. It enables valid confidence intervals and hypothesis tests and other methods for quantifying the uncertainty of a statistical analysis (e.g,. [20, 29, 50, 25, 26]). Incidentally, sets of noisy marginals are also used to generate differentially private synthetic data (e.g., [54, 4, 41, 10]).

For the case of marginals, significant effort has been spent in designing optimal or nearly optimal matrix mechanisms for just a single objective function (total variance of all the desired marginals) [32, 49, 13, 52, 53, 31, 51] and each new objective function requires significant additional effort [6, 18, 42, 46]. However, existing optimal solutions do not scale and additional effort is needed to design scalable, but suboptimal, matrix mechanisms for marginals [37, 38]. This is because prior work used mathematical properties specific to their chosen objective function in order to improve runtime. Furthermore, computing the individual variances of the desired noisy marginals is a slow process and more difficult is computing the covariance between cells in the same marginal.

37th Conference on Neural Information Processing Systems (NeurIPS 2023).

**Contributions.** Our paper addresses these problems with a novel matrix mechanisms called ResidualPlanner. It can optimize for a wide variety of convex objective functions and return solutions that are guaranteed to be optimal under Gaussian noise. It is highly scalable – running in seconds even when other scalable algorithms run out of memory. It also efficiently returns the variance and covariances of each cell of the desired marginals. It leverages the following insights. Since a dataset can be represented as a vector $\mathbf{x}$ of counts, and since a marginal query on a set $\mathbf{A}$ of attributes can be represented as a matrix $\mathbf{Q_A}$ (with $\mathbf{Q_A x}$ being the true answer to the marginal query), we find a new linearly independent basis that can parsimoniously represent both a marginal $\mathbf{Q_A}$ and the "difference" between two marginals $\mathbf{Q_A}$ and $\mathbf{Q_{A'}}$ (subspace spanned by the rows of $\mathbf{Q_A}$ that is orthogonal to the rows of $\mathbf{Q_{A'}}$). Using parsimonious linear bases, instead of overparametrized mechanisms, accounts for the scalability. Optimality results from a deep analysis of the symmetry that marginals impose on the optimal solution – the same linear basis is optimal for a wide variety of loss functions. Our code is available at https://github.com/dkifer/ResidualPlanner.

## 2 Preliminaries

The Kronecker product between a $k_1 \times k_2$ matrix $\mathbf{V} = \begin{bmatrix} v_{1,1} & \cdots & v_{1,k_2} \\ \vdots & \cdots & \vdots \\ v_{k_1,1} & \cdots & v_{k_1,k_2} \end{bmatrix}$ and an $\ell_1 \times \ell_2$ matrix $\mathbf{W}$, denoted by $\mathbf{V} \otimes \mathbf{W}$, is the $k_1\ell_1 \times k_2\ell_2$ matrix that can be represented in block matrix form as: $\begin{bmatrix} v_{1,1}\mathbf{W} & \cdots & v_{1,k_2}\mathbf{W} \\ \vdots & \cdots & \vdots \\ v_{k_1,1}\mathbf{W} & \cdots & v_{k_1,k_2}\mathbf{W} \end{bmatrix}$. A dataset $\mathcal{D} = \{r_1, \ldots, r_n\}$ is a collection of records (there is exactly one record for each individual in the dataset). Every record $r_i$ contains $n_a$ attributes $Att_1, \ldots, Att_{n_a}$ and each attribute $Att_j$ can take values $a_1^{(j)}, \ldots, a_{|Att_j|}^{(j)}$. An attribute value $a_i^{(j)}$ for attribute $Att_j$ can be represented as a vector using one-hot encoding. Specifically, let $e_i^{(j)}$ be a row vector of size $|Att_j|$ with a 1 in component $i$ and 0 everywhere else. In this way $e_i^{(j)}$ represents the attribute value $a_i^{(j)}$. A record $r$ with attributes $Att_1 = a_{i_1}^{(1)}$, $Att_2 = a_{i_2}^{(2)}, \ldots, Att_{n_a} = a_{i_{n_a}}^{(n_a)}$ can thus be represented as the Kronecker product $e_{i_1}^{(1)} \otimes e_{i_2}^{(2)} \otimes \cdots \otimes e_{i_{n_a}}^{(n_a)}$. This vector has a 1 in exactly one position and 0s everywhere else. The position of the 1 is the *index* of record $r$. With this notation, a dataset $\mathcal{D}$ can be represented as a vector $\mathbf{x}$ of integers. The value at index $i$ is the number of times the record associated with index $i$ appears in $\mathcal{D}$. The number of components in this vector is denoted as $d = \prod_{i=1}^{n_a} |Att_i|$. Given a subset $\mathbf{A}$ of attributes, a *marginal query* on $\mathbf{A}$ is a table of counts: for each combination of values for the attributes in $\mathbf{A}$, it provides the number of records in $\mathcal{D}$ having those attribute value combinations. The marginal query can be represented as a Kronecker product $\mathbf{Q_A} = \mathbf{V}_1 \otimes \cdots \otimes \mathbf{V}_{n_a}$ where $\mathbf{V}_i$ is the row vector of all ones (i.e, $\mathbf{1}_{|Att_i|}^T$) if $Att_i \notin \mathbf{A}$ and $\mathbf{V}_i$ is the identity matrix $\mathcal{I}_{|Att_i|}$ if $Att_i \in \mathbf{A}$. The answer to the marginal query is obtained by evaluating the matrix-vector product $\mathbf{Q_A x}$. For convenience, the notation introduced in this paper is summarized as a table in Section A in the supplementary material.

EXAMPLE 2.1. *As a running example, consider a dataset in which there are two attributes: $Att_1$ with values "yes" and "no", and $Att_2$ with values "low", "med", "high". The record (no, med) is represented by the kron product $\begin{bmatrix} 0 & 1 \end{bmatrix} \otimes \begin{bmatrix} 0 & 1 & 0 \end{bmatrix}$ and the marginal query on the set $\mathbf{A} = \{Att_1\}$ is represented as $\mathbf{Q}_{\{Att_1\}} = \begin{bmatrix} 1 & 0 \\ 0 & 1 \end{bmatrix} \otimes \begin{bmatrix} 1 & 1 & 1 \end{bmatrix}$. Similarly, the marginal on attribute $Att_2$ is represented as $\mathbf{Q}_{\{Att_2\}} = \begin{bmatrix} 1 & 1 \end{bmatrix} \otimes \begin{bmatrix} 1 & 0 & 0 \\ 0 & 1 & 0 \\ 0 & 0 & 1 \end{bmatrix}$. The query representing all one-way marginals is obtained by stacking them: $\mathbf{Q}^{1\text{-}way} = \begin{bmatrix} \mathbf{Q}_{\{Att_1\}} \\ \mathbf{Q}_{\{Att_2\}} \end{bmatrix}$ and $\mathbf{Q}^{1\text{-}way}\mathbf{x}$ consists of the five query answers (number of records with $Att_1 = yes$, number with $Att_1 = no$, number with $Att_2 = low$, etc.).*

### 2.1 Differential Privacy

A mechanism $\mathcal{M}$ is an algorithm whose input is a dataset and whose output provides privacy protections. Differential privacy is a family of privacy definitions that guide the behavior of mechanisms so that they can inject enough noise to mask the effects of any individual. There are many versions of differential privacy that support Gaussian noise, including approximate DP, zCDP, and Gaussian DP.

DEFINITION 2.2 (Differential Privacy). *Let $\mathcal{M}$ be a mechanism. For every pair of datasets $\mathcal{D}_1, \mathcal{D}_2$ that differ on the presence/absence of a single record and for all (measurable) sets $S \subseteq range(\mathcal{M})$,*

- *If $P(\mathcal{M}(\mathcal{D}_1) \in S) \leq e^\epsilon P(\mathcal{M}(\mathcal{D}_2) \in S) + \delta$ then $\mathcal{M}$ satisfies $(\epsilon, \delta)$-approximate differential privacy [17];*

- *If $\Phi^{-1}(P(\mathcal{M}(\mathcal{D}_1) \in S)) \leq \Phi^{-1}(P(\mathcal{M}(\mathcal{D}_2) \in S)) + \mu$, where $\Phi$ is the cdf of the standard Gaussian distribution, then $\mathcal{M}$ satisfies $\mu$-Gaussian DP [15].*

- *If the Rényi divergence $D_\alpha(\mathcal{M}(\mathcal{D}_1)\|\mathcal{M}(\mathcal{D}_2))$ between the output distributions of $\mathcal{M}(\mathcal{D}_1)$ and $\mathcal{M}(\mathcal{D}_2)$ satisfies $D_\alpha(\mathcal{M}(\mathcal{D}_1)\|\mathcal{M}(\mathcal{D}_2)) \leq \rho\alpha$ for all $\alpha > 1$, then $\mathcal{M}$ satisfies $\rho$-zCDP [7].*

Queries that are linear functions of the data vector $\mathbf{x}$ can be answered privately using the *linear Gaussian mechanism*, which adds correlated Gaussian noise to a linear function of $\mathbf{x}$, as follows.

DEFINITION 2.3 (Linear Gaussian Mechanism [46]). *Given a $m \times d$ matrix $\mathbf{B}$ and $m \times m$ covariance matrix $\mathbf{\Sigma}$, the (correlated) linear Gaussian mechanism $\mathcal{M}$ is defined as $\mathcal{M}(\mathbf{x}) = \mathbf{B}\mathbf{x} + N(\mathbf{0}, \mathbf{\Sigma})$. The privacy cost matrix of $\mathcal{M}$ is defined as $\mathbf{B}^T\mathbf{\Sigma}^{-1}\mathbf{B}$. The privacy cost of $\mathcal{M}$, denoted by $pcost(\mathcal{M})$, is the largest diagonal of the privacy cost matrix and is used to compute the privacy parameters: $\mathcal{M}$ satisfies $\rho$-zCDP with $\rho = pcost(\mathcal{M})/2$ [46], satisfies $(\epsilon, \delta)$-approximate DP with $\delta = \Phi(\sqrt{pcost(\mathcal{M})}/2 - \epsilon/\sqrt{pcost(\mathcal{M})}) - e^\epsilon\Phi(-\sqrt{pcost(\mathcal{M})}/2 - \epsilon/\sqrt{pcost(\mathcal{M})})$ (this is an increasing function of $pcost(\mathcal{M})$ [5]), and satisfies $\mu$-Gaussian DP with $\mu = \sqrt{pcost(\mathcal{M})}$ [15, 46].*

The use of a non-diagonal covariance matrix is crucial because it will help simplify the description of the optimal choices of $\mathbf{B}$ and $\mathbf{\Sigma}$. In particular, using non-diagonal covariance allows us to provide explicit formulas for the entries of the $\mathbf{B}$ matrices. We note that an algorithm $\mathcal{M}^*$ that releases the outputs of multiple linear Gaussian mechanisms $\mathcal{M}_1, \ldots, \mathcal{M}_k$ (with $\mathcal{M}_i(\mathbf{x}) = \mathbf{B}_i\mathbf{x} + N(\mathbf{0}, \mathbf{\Sigma}_i)$ ) is again a linear Gaussian mechanism. It is represented as $\mathcal{M}^*(\mathbf{x}) = \mathbf{B}^*\mathbf{x} + N(\mathbf{0}, \mathbf{\Sigma}^*)$ with the matrix $\mathbf{B}^*$ obtained by vertically stacking the $\mathbf{B}_i$ and with covariance $\mathbf{\Sigma}^*$ being a block-diagonal matrix where the blocks are the $\mathbf{\Sigma}_i$. Its privacy cost $pcost(\mathcal{M}^*) = pcost(\mathcal{M}_1, \ldots, \mathcal{M}_k)$ is the largest diagonal entry of $\sum_{i=1}^k \mathbf{B}_i^T\mathbf{\Sigma}_i^{-1}\mathbf{B}_i$.

## 2.2 Matrix Mechanism

The Matrix Mechanism [32, 52, 30, 53, 37, 38, 51, 46, 18, 42] is a framework for providing unbiased privacy-preserving answers to a workload of linear queries, represented by a matrix $\mathbf{W}$ (so that the true non-private answer to the workload queries is $\mathbf{W}\mathbf{x}$). The matrix mechanism framework consists of 3 steps: *select*, *measure*, and *reconstruct*. The purpose of the *select* phase is to determine *what* we add noise to and *how much* noise to use. More formally, when a user's preferred noise distribution is Gaussian, the select phase chooses a Gaussian linear mechanism $\mathcal{M}(\mathbf{x}) \equiv \mathbf{B}\mathbf{x} + N(\mathbf{0}, \mathbf{\Sigma})$ whose noisy output can be used to estimate the true query answer $\mathbf{W}\mathbf{x}$. Ideally, $\mathcal{M}$ uses the least amount of noise subject to privacy constraints (specified by a privacy definition and settings of its privacy parameters). The *measure* phase runs the mechanism on the data to produce (noisy) privacy-preserving outputs $\omega = \mathcal{M}(\mathbf{x})$. The *reconstruct* step uses $\omega$ to compute an unbiased estimate of $\mathbf{W}\mathbf{x}$. The unbiased estimate is typically $\mathbf{W}(\mathbf{B}^T\mathbf{\Sigma}^{-1}\mathbf{B})^\dagger\mathbf{B}^T\mathbf{\Sigma}^{-1}\omega$, where $\dagger$ represents the Moore-Penrose pseudo-inverse. This is the best linear unbiased estimate of $\mathbf{W}\mathbf{x}$ that can be obtained from $\omega$ [32]. This means that the goal of the select step is to optimize the choice of $\mathbf{B}$ and $\mathbf{\Sigma}$ so that the reconstructed answer is as accurate as possible, subject to privacy constraints. Ideally, a user would specify their accuracy requirements using a loss function, but existing matrix mechanisms do not allow this flexibility – they hard-code the loss function. In fact, adding support for new loss function used to require significant research and new optimization algorithms [53, 46, 18] because each new algorithm was customized to specific properties of a chosen loss function. On top of this, existing optimal matrix mechanism algorithms do not scale, while scalable matrix mechanisms are not guaranteed to produce optimal solutions [37]. Additionally, the reconstruction phase should also compute the variance of each workload answer. The variances are the diagonals of $\mathbf{W}(\mathbf{B}^T\mathbf{\Sigma}^{-1}\mathbf{B})^\dagger\mathbf{W}^T$ and making this computation scale is also challenging.

# 3 Additional Related Work

The marginal release mechanism by Barak et al. [6] predates the matrix mechanism [32, 52, 30, 53, 13, 43, 37, 51, 46, 18, 42, 38] and adds noise to the Fourier decomposition of marginals. We add noise to a *different* basis, resulting in the scalability and optimality properties. The SVD bound [31] is a lower bound on total matrix mechanism error when the loss function is the sum of variances. This lower bound is tight for marginals and we use it as a sanity check for our results and implementation (note ResidualPlanner provides optimal solutions even when the SVD bound is infeasible to compute).

Alternative approaches to the matrix mechanism can produce privacy preserving marginal query answers that reduce variance by adding bias. This is often done by generating differentially private synthetic data or other such data synopses from which marginals can be computed. State-of-the art approaches iteratively ask queries and fit synthetic data to the resulting answers [22, 34, 4, 19, 39, 35, 44, 56]. For such mechanisms, it is difficult to estimate error of a query answer but recently AIM [39] has made progress in upper bounding the error. PGM [41] provides a connection between the matrix mechanism and this line of work, as it can postprocess noisy marginals into synthetic data. It is a better alternative to sampling a synthetic dataset from models fit to carefully chosen marginals [54, 11, 55, 10]. Synthetic data for answering marginal queries can also be created from random projections [48], copulas [33, 3], and deep generative models [23, 1, 35].

With respect to the matrix mechanism, the reconstruction step is often one of the bottlenecks to scalability. While PGM [41] provides one solution, another proposal by McKenna et al. [40] is to further improve scalability by sacrificing some consistency (the answers to two different marginals may provide conflicting answers to submarginals they have in common). Work on differential privacy marginals has also seen extensions to hierarchical datasets, in which records form meaningful groups that need to be queried. That is, in addition to marginals on characteristics of people, marginals can be computed in different hierarchies such as geographic level (state, county, etc.) and marginals on household composition (or other groupings of people) [2, 28, 36].

# 4 ResidualPlanner

ResidualPlanner is our proposed matrix mechanism for optimizing the accuracy of marginal queries with Gaussian noise. It is optimal and more scalable than existing approaches. It supports optimizing the accuracy of marginals under a wide variety of loss functions and provides exact variances/covariances of the noisy marginals in closed-form. In this section, we first explain the loss functions it supports. We then describe the base mechanisms it uses to answer marginal queries. We next show how to reconstruct the marginal queries from the outputs of the base mechanisms and how to compute their variances in closed form. We then explain how to optimize these base mechanisms for different loss functions. The reason this selection step is presented last is because it depends on the closed form variance calculations. Then we analyze computational complexity. To aid in understanding, we added a complete numerical run-through of the steps in Section B of the supplementary material. All of our proofs also appear in the supplementary material.

## 4.1 Loss Functions Supported by ResidualPlanner

The loss functions we consider are generalizations of the sum of variances and max of variances used in prior work. Our more general class of loss functions is based on the following principle: different marginals can have different relative importance but within a marginal, its cells are equally important. That is, a loss function can express that the two-way marginal on the attribute set {Race, Marital Status} is more important (i.e., requires more accuracy) than the 1-way marginal on {EducationLevel}, but all cells within the {Race, MaritalStatus} marginal are equally important. This is a commonly accepted principle for answering differentially private marginal queries (e.g., [32, 52, 30, 53, 37, 51, 46, 18, 42, 39, 4, 34]) and is certainly true for the 2020 Census redistricting data [2].

Let $Wkload = \{\mathbf{A}_1, \ldots, \mathbf{A}_k\}$ be a workload of marginals, where each $\mathbf{A}_i$ is a subset of attributes and represents a marginal. E.g., $Wkload = \{\{\text{Race, MaritalStatus}\}, \{\text{EducationLevel}\}\}$ consists of 2 marginals, a two-way marginal on Race/MaritalStatus, and a one-way marginal on Education. Let $\mathcal{M}$ be a Gaussian linear mechanism whose output can be used to reconstruct unbiased answers to the marginals in $Wkload$. For each $\mathbf{A}_i \in Wkload$, let $Var(\mathbf{A}_i; \mathcal{M})$ be the function that returns the

variances of the reconstructed answers to the marginal on $\mathbf{A}_i$; the output of $Var(\mathbf{A}_i; \mathcal{M})$ is a vector $v_i$ with one component for each cell of the marginal on $\mathbf{A}_i$. A loss function $\mathcal{L}$ aggregates all of these vectors together: $\mathcal{L}(v_1, \ldots, v_k)$. We have the following regularity conditions on the loss function.

DEFINITION 4.1 (Regular Loss Function). *We say the loss function $\mathcal{L}$ is* regular *if: (1) $\mathcal{L}$ is convex and continuous; (2) $\mathcal{L}(v_1, \ldots, v_k)$ is minimized when all the $v_i$ are the 0 vectors; and (3) for any $i$, permuting just the components of $v_i$ does not affect the value of $\mathcal{L}(v_1, \ldots, v_k)$. This latter condition just says that cells within the* same *marginal are equally important.*

Loss functions used on prior work are all regular. For example, weighted sum of variances [32, 52, 30, 53, 37, 51] can be expressed as $\mathcal{L}(v_1, \ldots, v_k) = \sum_i c_i \mathbf{1}^T v_i$, where the $c_i$ are the nonnegative weights that indicate the relative importance of the different marginals. Another popular loss function is maximum (weighted) variance [46, 18, 42], expressed as $\mathcal{L}(v_1, \ldots, v_k) = \max \left\{ \frac{\max(v_1)}{c_1}, \ldots, \frac{\max(v_k)}{c_k} \right\}$. Thus, the optimization problem that the selection step needs to solve is either privacy constrained: minimize loss while keeping privacy cost (defined at the end of Section 2.1) below a threshold $\gamma$; or utility constrained: minimize privacy cost such that the loss is at most $\gamma$.

$$\text{Privacy constrained:} \quad \arg\min_{\mathcal{M}} \mathcal{L}(Var(A_1; \mathcal{M}), \ldots, Var(A_k; \mathcal{M})) \quad \textbf{s.t.} \quad pcost(\mathcal{M}) \leq \gamma \quad (1)$$

$$\text{Utility constrained:} \quad \arg\min_{\mathcal{M}} pcost(\mathcal{M}) \quad \textbf{s.t.} \quad \mathcal{L}(Var(A_1; \mathcal{M}), \ldots, Var(A_k; \mathcal{M})) \leq \gamma \quad (2)$$

We note that regular loss functions cover other interesting cases. For instance, suppose Alicia, Bob, and Carol wish to minimize the sum of variances on their own separate workloads. Taking the max over these three sum-of-variances as the loss function allows the data curator to minimize the largest unhappiness among the three data stakeholders.

## 4.2 Base Mechanisms used by ResidualPlanner

In the most common application setting, the user picks a privacy budget, a workload of marginals and a loss function $\mathcal{L}$. Based on these choices, a matrix mechanism must decide what linear queries to add noise to and how much noise to add to them. Then it uses those noisy measurements to reconstruct answers to the workload queries. In the case of ResidualPlanner, the linear queries that need noise are represented by *base mechanisms* that are described in this section. Each base mechanism has a scalar noise parameter that determines how much noise it uses (so optimizing the loss function $\mathcal{L}$ is equivalent to finding a good value for the noise parameter of each base mechanism). As long as the loss function $\mathcal{L}$ is regular, we prove that an optimal mechanism can be constructed from the set of base mechanisms that we describe here.

To begin, we define a *subtraction matrix* $\mathbf{Sub}_m$ to be an $(m-1) \times m$ matrix where the first column is filled with 1, entries of the form $(i, i+1)$ are -1, and all other entries are 0. For example, $\mathbf{Sub}_3 = \left[ \begin{smallmatrix} 1 & -1 & 0 \\ 1 & 0 & -1 \end{smallmatrix} \right]$ and $\mathbf{Sub}_2 = \left[ \begin{smallmatrix} 1 & -1 \end{smallmatrix} \right]$. We use these subtraction matrices to define special matrices called *residual matrices* that are important for our algorithm.

For any subset $\mathbf{A} \subseteq \{Att_1, \ldots, Att_{n_a}\}$ of attributes, we define the *residual matrix* $\mathbf{R_A}$ as the Kronecker product $\mathbf{R_A} = \mathbf{V}_1 \otimes \cdots \otimes \mathbf{V}_{n_a}$, where $\mathbf{V}_i = \mathbf{1}_{|Att_i|}^T$ if $Att_i \notin \mathbf{A}$ and $\mathbf{V}_i = \mathbf{Sub}_{|Att_i|}$ if $Att_i \in \mathbf{A}$. Continuing Example 2.1, we have $\mathbf{R}_\emptyset = \left[ \begin{smallmatrix} 1 & 1 \end{smallmatrix} \right] \otimes \left[ \begin{smallmatrix} 1 & 1 & 1 \end{smallmatrix} \right]$, and $\mathbf{R}_{\{Att_1\}} = \left[ \begin{smallmatrix} 1 & -1 \end{smallmatrix} \right] \otimes \left[ \begin{smallmatrix} 1 & 1 & 1 \end{smallmatrix} \right]$, and $\mathbf{R}_{\{Att_2\}} = \left[ \begin{smallmatrix} 1 & 1 \end{smallmatrix} \right] \otimes \left[ \begin{smallmatrix} 1 & -1 & 0 \\ 1 & 0 & -1 \end{smallmatrix} \right]$, and $\mathbf{R}_{\{Att_1, Att_2\}} = \left[ \begin{smallmatrix} 1 & -1 \end{smallmatrix} \right] \otimes \left[ \begin{smallmatrix} 1 & -1 & 0 \\ 1 & 0 & -1 \end{smallmatrix} \right]$.

Using subtraction matrices, we also define the matrix $\mathbf{\Sigma_A}$ as the Kronecker product $\bigotimes_{Att_i \in \mathbf{A}} (\mathbf{Sub}_{|Att_i|} \mathbf{Sub}_{|Att_i|}^T)$ and we note that it is proportional to $\mathbf{R_A} \mathbf{R_A}^T$. $\mathbf{\Sigma}_\emptyset$ is defined as 1.

Each subset $\mathbf{A}$ of attributes can be associated with a "base" mechanism $\mathcal{M}_\mathbf{A}$ that takes as input the data vector $\mathbf{x}$ and a scalar parameter $\sigma_\mathbf{A}^2$ for controlling how noisy the answer is. $\mathcal{M}_\mathbf{A}$ is defined as:

$$\mathcal{M}_\mathbf{A}(\mathbf{x}; \sigma_\mathbf{A}^2) \equiv \mathbf{R_A} \mathbf{x} + N(\mathbf{0}, \sigma_\mathbf{A}^2 \mathbf{\Sigma_A}) \quad (3)$$

The residual matrices $\mathbf{R_A}$ used by base mechanisms form a linearly independent basis that compactly represent marginals, as the next result shows.

THEOREM 4.2. *Let $\mathbf{A}$ be a set of attributes and let $\mathbf{Q_A}$ be the matrix representation of the marginal on $\mathbf{A}$. Then the rows of the matrices $\mathbf{R}_{\mathbf{A}'}$, for all $\mathbf{A}' \subseteq \mathbf{A}$, form a linearly independent basis of the row space of $\mathbf{Q_A}$. Furthermore, if $\mathbf{A}' \neq \mathbf{A}''$ then $\mathbf{R}_{\mathbf{A}'} \mathbf{R}_{\mathbf{A}''}^T = \mathbf{0}$ (they are mutually orthogonal).*

REMARK 4.3. *To build an intuitive understanding of residual matrices, consider again Example 2.1. Both $\mathbf{R}_\emptyset$ and $\mathbf{Q}_\emptyset$ are the sum query (marginal on no attributes). The rows of $\mathbf{R}_{\{Att_1\}}$ span the subspace of $\mathbf{Q}_{\{Att_1\}}$ that is orthogonal to $\mathbf{Q}_\emptyset$ (and similarly for $\mathbf{R}_{\{Att_2\}}$). The rows of $\mathbf{R}_{\{Att_1,Att_2\}}$ span the subspace of $\mathbf{Q}_{\{Att_1,Att_2\}}$ that is orthogonal to both $\mathbf{Q}_{\{Att_1\}}$ and $\mathbf{Q}_{\{Att_2\}}$. Hence a residual matrix spans the subspace of a marginal that is orthogonal to its sub-marginals.*

Theorem 4.2 has several important implications. If we define the downward closure of a marginal workload $Wkload = \{\mathbf{A}_1, \ldots, \mathbf{A}_k\}$ as the collection of all subsets of the sets in $Wkload$ (i.e., $\text{closure}(Wkload) = \{\mathbf{A}' : \mathbf{A}' \subseteq \mathbf{A} \text{ for some } \mathbf{A} \in Wkload\}$) then the theorem implies that the combined rows from $\{\mathbf{R}_{\mathbf{A}'} : \mathbf{A}' \in \text{closure}(Wkload)\}$ forms a linearly independent basis for the marginals in the workload. In other words, it is a linearly independent bases for the space spanned by the rows of the marginal query matrices $\mathbf{Q}_\mathbf{A}$ for $\mathbf{A} \in Wkload$. Thus, in order to provide privacy-preserving answers to all of the marginals represented in $Wkload$, we need all the mechanisms $\mathcal{M}_{\mathbf{A}'}$ for $\mathbf{A}' \in \text{closure}(Wkload)$ – any other matrix mechanism that provides fewer noisy outputs cannot reconstruct unbiased answers to the workload marginals. This is proved in Theorem 4.4, which also states that optimality is achieved by carefully setting the $\sigma_\mathbf{A}$ noise parameter for each $\mathcal{M}_\mathbf{A}$.

THEOREM 4.4. *Given a marginal workload $Wkload$ and a regular loss function $\mathcal{L}$, suppose the optimization problem (either Equation 1 or 2) is feasible. Then there exist nonnegative constants $\sigma_\mathbf{A}^2$ for each $\mathbf{A} \in \text{closure}(Wkload)$ (the constants do not depend on the data), such that the optimal linear Gaussian mechanism $\mathcal{M}_{opt}$ for loss function $\mathcal{L}$ releases $\mathcal{M}_\mathbf{A}(\mathbf{x}; \sigma_\mathbf{A}^2)$ for all $\mathbf{A} \in \text{closure}(Wkload)$. Furthermore, any matrix mechanism for this workload must produce at least this many noise measurements during its selection phase.*

---

**Algorithm 1:** Efficient implementation of $\mathcal{M}_\mathbf{A}(\mathbf{x}; \sigma_\mathbf{A}^2) \equiv \mathbf{R}_\mathbf{A}\mathbf{x} + N(\mathbf{0}, \sigma_\mathbf{A}^2 \mathbf{\Sigma}_\mathbf{A})$

---
1 $\mathbf{v} \leftarrow \mathbf{Q}_\mathbf{A}\mathbf{x}$// Evaluate the true marginal
2 $m \leftarrow \prod_{Att_i \in \mathbf{A}} |Att_i|$
3 $\mathbf{H} \leftarrow \bigotimes_{Att_i \in \mathbf{A}} \mathbf{Sub}_{|Att_i|}$// Use implicit representation, don't expand
4 $\mathbf{z} \leftarrow N(\mathbf{0}, \mathcal{I}_m)$// independent noise
5 **return** $\mathbf{H}\mathbf{v} + \sigma_\mathbf{A}\mathbf{H}\mathbf{z}$// use kron-product/vector multiplication from [37]

---

$\mathcal{M}_\mathbf{A}$ can be evaluated efficiently, directly from the marginal of $\mathbf{x}$ on attribute set $\mathbf{A}$, as shown in Algorithm 1. It uses the technique from [37] to perform fast multiplication between a Kronecker product and a vector (so that the Kronecker product does not need to be expanded). It also generates correlated noise from independent Gaussians. The privacy cost $pcost(\mathcal{M}_\mathbf{A})$ of each base mechanism $\mathcal{M}_\mathbf{A}$ is also easy to compute and is given by the following theorem.

THEOREM 4.5. *The privacy cost of $\mathcal{M}_\mathbf{A}$ with noise parameter $\sigma_\mathbf{A}^2$ is $\frac{1}{\sigma_\mathbf{A}^2} \prod_{Att_i \in \mathbf{A}} \frac{|Att_i|-1}{|Att_i|}$ and the evaluation of $\mathcal{M}_\mathbf{A}$ given in Algorithm 1 is correct – i.e., the output has the distribution $N(\mathbf{R}_\mathbf{A}\mathbf{x}, \sigma_\mathbf{A}^2 \mathbf{\Sigma}_\mathbf{A})$.*

### 4.3 Reconstruction

Next we explain how to reconstruct unbiased answers to marginal queries from the outputs of the base mechanisms and how to compute (co)variances of the reconstructed marginals efficiently, without any heavy matrix operations (inversion, pseudo-inverses, etc.). Then, given the closed form expressions for marginals and privacy cost (Theorem 4.5), we will be able to explain in Section 4.4 how to optimize the $\sigma_\mathbf{A}^2$ parameters of the base mechanisms $\mathcal{M}_\mathbf{A}$ to optimize regular loss functions $\mathcal{L}$.

Since the base mechanisms were built using a linearly independent basis, reconstruction is unique – just efficiently invert the basis. Hence, unlike PGM and its extensions [41, 40], our reconstruction algorithm does not need to solve an optimization problem and can reconstruct each marginal independently, thus allowing marginals to be reconstructed in parallel, or as needed by users. The reconstructed marginals are consistent with each other (any two reconstructed marginals agree on their sub-marginals). Just as the subtraction matrices $\mathbf{Sub}_k$ were useful in constructing the base mechanisms $\mathcal{M}_\mathbf{A}$, their pseudo-inverses $\mathbf{Sub}_k^\dagger$ are useful for reconstructing noisy marginals from the

noisy answers of $\mathcal{M}_\mathbf{A}$. The pseudo-inverses have a closed form. For example $\mathbf{Sub}_4 = \begin{bmatrix} 1 & -1 & 0 & 0 \\ 1 & 0 & -1 & 0 \\ 1 & 0 & 0 & -1 \end{bmatrix}$ and $\mathbf{Sub}_4^\dagger = \frac{1}{4}\begin{bmatrix} 1 & 1 & 1 \\ -3 & 1 & 1 \\ 1 & -3 & 1 \\ 1 & 1 & -3 \end{bmatrix}$. More generally, they are expressed as follows:

LEMMA 4.6. *For any* $Att_i$, *let* $\ell = |Att_i|$. *The matrix* $\mathbf{Sub}_\ell$ *has the following block matrix, with dimensions* $\ell \times (\ell-1)$, *as its pseudo-inverse (and right inverse):* $\mathbf{Sub}_\ell^\dagger = \frac{1}{\ell}\begin{bmatrix} \mathbf{1}_{\ell-1}^T \\ \mathbf{1}_{\ell-1}\mathbf{1}_{\ell-1}^T - \ell\mathcal{I}_{\ell-1} \end{bmatrix}$.

Each mechanism $\mathcal{M}_\mathbf{A}$, for $\mathbf{A} \in \text{closure}(Wkload)$, has a noise scale parameter $\sigma_\mathbf{A}^2$ and a noisy output that we denote by $\omega_\mathbf{A}$. After we have obtained the noisy outputs $\omega_\mathbf{A}$ for all $\mathbf{A} \in \text{closure}(Wkload)$, we can proceed with the reconstruction phase. The reconstruction of an unbiased noisy answer for any marginal on an attribute set $\mathbf{A} \in \text{closure}(Wkload)$ is obtained using Algorithm 2. We note that to reconstruct a marginal on attribute set $\mathbf{A}$, one only needs to use the noisy answers $\omega_{\mathbf{A}'}$ for $\mathbf{A}' \in \text{closure}(\mathbf{A})$. In other words, if we want to reconstruct a marginal on attribute set $\{Att_1, Att_2\}$, we only need the outputs of $\mathcal{M}_\emptyset$, $\mathcal{M}_{\{Att_1\}}$, $\mathcal{M}_{\{Att_2\}}$, and $\mathcal{M}_{\{Att_1, Att_2\}}$ no matter how many other attributes are in the data and no matter what other marginals are in the $Wkload$. We emphasize again, the reconstruction phase does not run the base mechanisms anymore, it is purely post-processing.

---

**Algorithm 2:** Reconstruct Unbiased Answers to the Marginal on $\mathbf{A}$

**Input:** Noise scale parameters $\sigma_{\mathbf{A}'}^2$ and noisy answer vector $\omega_{\mathbf{A}'}$ of mechanism $\mathcal{M}_{\mathbf{A}'}$ for every $\mathbf{A}' \in closure(\mathbf{A})$.
**Output:** $\mathbf{q}$ is output as an unbiased noisy estimate of $\mathbf{Q}_\mathbf{A}\mathbf{x}$.

1   $\mathbf{q} \leftarrow \mathbf{0}$
2   **for** *each* $\mathbf{A}' \in closure(\mathbf{A})$ **do**
3      $\mathbf{U} \leftarrow \mathbf{V}_1 \otimes \cdots \otimes \mathbf{V}_{n_a}$, where $\mathbf{V}_i = \begin{cases} \mathbf{Sub}_{|Att_i|}^\dagger & \text{if } Att_i \in \mathbf{A}' \\ \frac{1}{|Att_i|}\mathbf{1}_{|Att_i|} & \text{if } Att_i \in \mathbf{A}/\mathbf{A}' \\ [1] & \text{if } Att_i \notin \mathbf{A} \end{cases}$
4      $\mathbf{q} \leftarrow \mathbf{q} + \mathbf{U}\omega_{\mathbf{A}'}$ // use kron-product/vector multiplication from [37]
5   **return** $\mathbf{q}$

---

THEOREM 4.7. *Given a marginal workload* $Wkload$ *and positive numbers* $\sigma_\mathbf{A}^2$ *for each* $\mathbf{A} \in closure(Wkload)$, *let* $\mathcal{M}$ *be the mechanism that outputs* $\{\mathcal{M}_\mathbf{A}(\mathbf{x}; \sigma_\mathbf{A}^2) : \mathbf{A} \in closure(Wkload)\}$ *and let* $\{\omega_\mathbf{A} : \mathbf{A} \in closure(Wkload)\}$ *denote the privacy-preserving noisy answers (e.g.,* $\omega_\mathbf{A} = \mathcal{M}_\mathbf{A}(\mathbf{x}, \sigma^2)$). *Then for any marginal on an attribute set* $\mathbf{A} \in closure(Wkload)$, *Algorithm 2 returns the unique linear unbiased estimate of* $\mathbf{Q}_\mathbf{A}\mathbf{x}$ *(i.e., answers to the marginal query) that can be computed from the noisy differentially private answers.*

*The variances* $Var(\mathbf{A}; \mathcal{M})$ *of all the noisy cell counts of the marginal on* $\mathbf{A}$ *is the vector whose components are all equal to* $\sum_{\mathbf{A}' \subseteq \mathbf{A}} \left( \sigma_{\mathbf{A}'}^2 \prod_{Att_i \in \mathbf{A}'} \frac{|Att_i|-1}{|Att_i|} * \prod_{Att_j \in (\mathbf{A}/\mathbf{A}')} \frac{1}{|Att_j|^2} \right)$. *The covariance between any two noisy answers of the marginal on* $\mathbf{A}$ *is* $\sum_{\mathbf{A}' \subseteq \mathbf{A}} \left( \sigma_{\mathbf{A}'}^2 \prod_{Att_i \in \mathbf{A}'} \frac{-1}{|Att_i|} * \prod_{Att_j \in (\mathbf{A}/\mathbf{A}')} \frac{1}{|Att_j|^2} \right)$.

To see an example of how the choices of the $\sigma_\mathbf{A}^2$ affect the variance of different marginals, see Section C in the supplementary material.

## 4.4   Optimizing the Base Mechanism Selection

We now consider how to find the optimal Gaussian linear mechanism $\mathcal{M}^*$ that solves the optimization problems in Equations 1 or 2. Given a workload on marginals $Wkload$, the optimization involves $Var(\mathbf{A}; \mathcal{M}^*)$ for $\mathbf{A} \in Wkload$ (the variance of the marginal answers reconstructed from the output of $\mathcal{M}^*$) and $pcost(\mathcal{M}^*)$, from which the privacy parameters of different flavors of differential privacy can be computed.

Theorem 4.4 says that $\mathcal{M}^*$ works by releasing $\mathcal{M}_\mathbf{A}(\mathbf{x}; \sigma_\mathbf{A}^2)$ for each $\mathbf{A} \in \text{closure}(Wkload)$ for appropriately chosen values of $\sigma_\mathbf{A}^2$. The privacy cost $pcost(\mathcal{M}^*)$ is the sum of the privacy costs of

the $\mathcal{M}_{\mathbf{A}}$. Theorem 4.5 therefore shows that $pcost(\mathcal{M}^*)$ is a positive linear combination of the values $1/\sigma_{\mathbf{A}}^2$ for $\mathbf{A} \in \text{closure}(Wkload)$ and is therefore convex in the $\sigma_{\mathbf{A}}^2$ values. Meanwhile, Theorem 4.7 shows how to represent, for each $\mathbf{A} \in \text{closure}(Wkload)$, the quantity $Var(\mathbf{A}; \mathcal{M}^*)$ as a positive linear combination of $\sigma_{\mathbf{A}'}^2$, for $\mathbf{A}' \in \text{closure}(\mathbf{A}) \subseteq \text{closure}(Wkload)$. Therefore, the loss function $\mathcal{L}$ is also convex in the $\sigma_{\mathbf{A}}^2$ values.

Thus the optimization problems in Equations 1 and 2 can be written as minimizing a convex function of the $\sigma_{\mathbf{A}}^2$ subject to convex constraints. In fact, in Equation 2, the constraints are linear when the optimization variables represent the $\sigma_{\mathbf{A}}^2$ and in Equation 1 the constraints are linear when the optimization variables represent the $1/\sigma_{\mathbf{A}}^2$. Furthermore, when the loss function is the weighted sum of variances of the marginal cells, the solution can be obtained in closed form (see supplementary material). Otherwise, we use CVXPY/ECOS [12, 14] for solving these convex optimization problems.

### 4.5 Computational Complexity

Although the universe size $|Att_1| \times \cdots \times |Att_{n_a}|$ grows exponentially with the number of attributes, the following theorem shows that the time complexity of ResidualPlanner depends directly on quantities that typically grow polynomially, such as the number of desired marginals and total number of cells in those marginals.

THEOREM 4.8. *Let $n_a$ be the total number of attributes. Let $\#cells(\mathbf{A})$ denote the number of cells in the marginal on attribute set $\mathbf{A}$. Then:*

1. *Expressing the privacy cost of the optimal mechanism $\mathcal{M}^*$ as a linear combination of the $1/\sigma_{\mathbf{A}}^2$ values takes $O(\sum_{\mathbf{A} \in Wkload} \#cells(\mathbf{A}))$ total time.*

2. *Expressing all of the $Var(\mathbf{A}; \mathcal{M}^*)$, for $\mathbf{A} \in Wkload$, as a linear combinations of the $\sigma_{\mathbf{A}}^2$ values can be done in $O(\sum_{\mathbf{A} \in Wkload} \#cells(\mathbf{A}))$ total time.*

3. *Computing all the noisy outputs of the optimal mechanism (i.e., $\mathcal{M}_{\mathbf{A}}(\mathbf{x}; \sigma_{\mathbf{A}}^2)$ for $\mathbf{A} \in \text{closure}(Wkload)$) takes $O\left(n_a \sum_{\mathbf{A} \in Wkload} \prod_{Att_i \in \mathbf{A}}(|Att_i| + 1)\right)$ total time after the true answers have been precomputed (Line 1 in Algorithm 1). Note that the total number of cells on marginals in $Wkload$ is $O\left(\sum_{\mathbf{A} \in Wkload} \prod_{Att_i \in \mathbf{A}} |Att_i|\right)$.*

4. *Reconstructing marginals for all $\mathbf{A} \in Wkload$ takes $O(\sum_{\mathbf{A} \in Wkload} |\mathbf{A}| \#cells(\mathbf{A})^2)$ total time.*

5. *Computing the variance of the cells for all of the marginals for $\mathbf{A} \in Wkload$ can be done in $O(\sum_{\mathbf{A} \in Wkload} \#cells(\mathbf{A}))$ total time.*

To get a sense of these numbers, consider a dataset with 20 attributes, each having 3 possible values. If the workload consists of all 3-way marginals, there are 1,140 marginals each with 27 cells so $n_{cells} = 30,780$. The quantity inside the big-O for the selection step is $1,459,200$ (roughly the number of scalar multiplications needed). These are all easily manageable on modern computers even without GPUs. Our experiments, under more challenging conditions, run in seconds.

## 5 Experiments

We next compare the accuracy and scalability of ResidualPlanner against HDMM [38], including variations of HDMM with faster reconstruction phases [41]. The hardware used was an Ubuntu 22.04.2 server with 12 Intel(R) Core(TM) i7-8700 CPU @ 3.20GHz processors and 32GB of DDR4 RAM. We use 3 real datasets to evaluate accuracy and 1 synthetic dataset to evaluate scalability. The real datasets are (1) the Adult dataset [16] with 14 attributes, each having domain sizes $100, 100, 100, 99, 85, 42, 16, 15, 9, 7, 6, 5, 2, 2$, respectively, resulting in a record domain size of $6.41 * 10^{17}$; (2) the CPS dataset [9] with 5 attributes, each having domain size $100, 50, 7, 4, 2$, respectively, resulting in a record domain size of $2.8 * 10^5$; (3) the Loans dataset [24] with 12 attributes, each having domain size $101, 101, 101, 101, 3, 8, 36, 6, 51, 4, 5, 15$, respectively, resulting in a record domain size of $8.25 * 10^{15}$. The synthetic dataset is called Synth-$n^d$. Here $d$ refers to the number of attributes (we experiment from $d = 2$ to $d = 100$) and $n$ is the domain size of each attribute. The running times of the algorithms only depend on $n$ and $d$ and not on the records in the synthetic data. For all experiments, we set the privacy cost $pcost$ to 1, so all mechanisms being compared satisfy 0.5-zCDP and 1-Gaussian DP.

## 5.1 Scalability of the Selection Phase

We first consider how long each method takes to perform the selection phase (i.e., determine what needs noisy answers and how much noise to use). HDMM can only optimize total variance, which is equivalent to root mean squared error. For ResidualPlanner we consider both RMSE and max variance as objectives (the latter is a harder to solve problem). Each algorithm is run 5 times and the average is taken. Table 1 shows running time results; accuracy results will be presented later.

Table 1: **Time for Selection Step in seconds** on Synth$-n^d$ dataset. $n = 10$ and the number of attributes $d$ varies. The workload consists of all marginals on $\leq 3$ attributes each. Times for HDMM are reported with $\pm 2$ standard deviations.

| $d$ | HDMM RMSE Objective | ResidualPlanner RMSE Objective | ResidualPlanner Max Variance Objective |
|---|---|---|---|
| 2 | $0.013 \pm 0.003$ sec | $0.001 \pm 0.0008$ sec | $0.007 \pm 0.001$ sec |
| 6 | $0.065 \pm 0.012$ sec | $0.002 \pm 0.0008$ sec | $0.009 \pm 0.001$ sec |
| 10 | $0.639 \pm 0.059$ sec | $0.009 \pm 0.001$ sec | $0.018 \pm 0.001$ sec |
| 12 | $4.702 \pm 0.315$ sec | $0.015 \pm 0.001$ sec | $0.028 \pm 0.001$ sec |
| 14 | $46.054 \pm 12.735$ sec | $0.025 \pm 0.002$ sec | $0.041 \pm 0.001$ sec |
| 15 | $201.485 \pm 13.697$ sec | $0.030 \pm 0.017$ sec | $0.050 \pm 0.001$ sec |
| 20 | Out of memory | $0.079 \pm 0.017$ sec | $0.123 \pm 0.023$ sec |
| 30 | Out of memory | $0.247 \pm 0.019$ sec | $0.461 \pm 0.024$ sec |
| 50 | Out of memory | $1.207 \pm 0.047$ sec | $4.011 \pm 0.112$ sec |
| 100 | Out of memory | $9.913 \pm 0.246$ sec | $121.224 \pm 3.008$ sec |

As we can see, optimizing for max variance is more difficult than for RMSE, but ResidualPlanner does it quickly even for data settings too big for HDMM. The runtime of HDMM increases rapidly, while even for the extreme end of our experiments, ResidualPlanner needs just a few minutes.

## 5.2 Scalability of the Reconstruction Phase

We next evaluate the scalability of the reconstruction phase under the same settings. The reconstruction speed for ResidualPlanner does not depend on the objective of the selection phase. Here we compare against HDMM [38] and a version of HDMM with improved reconstruction scalability called HDMM+PGM [38, 41] (the PGM settings used 50 iterations of its Local-Inference estimator, as the default 1000 was too slow). Since HDMM cannot perform the selection phase after a certain point, reconstruction results also become unavailable. Table 2 shows ResidualPlanner is clearly faster.

Table 2: **Time for Reconstruction Step in seconds** on Synth$-n^d$ dataset. $n = 10$ and the number of attributes $d$ varies. The workload consists of all marginals on $\leq 3$ attributes each. Times are reported with $\pm 2$ standard deviations. Reconstruction can only be performed if the select step completed.

| $d$ | HDMM | HDMM + PGM | ResidualPlanner |
|---|---|---|---|
| 2 | $0.003 \pm 0.0006$ sec | $0.155 \pm 0.011$ sec | $0.005 \pm 0.003$ sec |
| 6 | $0.173 \pm 0.011$ sec | $4.088 \pm 0.233$ sec | $0.023 \pm 0.004$ sec |
| 10 | Out of memory in reconstruction | $20.340 \pm 2.264$ sec | $0.125 \pm 0.032$ sec |
| 12 | Out of memory in reconstruction | $39.162 \pm 1.739$ sec | $0.207 \pm 0.004$ sec |
| 14 | Out of memory in reconstruction | $69.975 \pm 4.037$ sec | $0.330 \pm 0.006$ sec |
| 15 | Out of memory in reconstruction | $91.101 \pm 7.621$ sec | $0.413 \pm 0.006$ sec |
| 20 | N/A (select step failed) | N/A (select step failed) | $1.021 \pm 0.011$ sec |
| 30 | N/A (select step failed) | N/A (select step failed) | $3.587 \pm 0.053$ sec |
| 50 | N/A (select step failed) | N/A (select step failed) | $17.029 \pm 0.212$ sec |
| 100 | N/A (select step failed) | N/A (select step failed) | $154.538 \pm 15.045$ sec |

## 5.3 Accuracy Comparisons

Since ResidualPlanner is optimal, the purpose of the accuracy comparisons is a sanity check. For RMSE, comparisons of the quality of ResidualPlanner to the theoretically optimal lower bound, known as the SVD bound [31], can be found in the supplementary material in Section I (ResidualPlanner

matches the lower bound). We note ResidualPlanner can provide solutions even when the SVD bound is infeasible to compute.

We also compare ResidualPlanner to HDMM when the user is interested in the maximum variance objective. This just shows that it is important to optimize for the user's objective function and that the optimal solution for RMSE (the only objective HDMM can optimize) is not a good general-purpose approximation for other objectives (as shown in Table 3). Additional comparisons are provided in the supplementary material.

Table 3: Max Variance Comparisons with ResidualPlanner and HDMM (showing that being restricted to optimizing only RMSE is not a good approximation of Max Variance optimization).

| Workload | Adult Dataset | | CPS Dataset | | Loans Dataset | |
|---|---|---|---|---|---|---|
| | ResPlan | HDMM | ResPlan | HDMM | ResPlan | HDMM |
| 1-way Marginals | 12.047 | 41.772 | 4.346 | 13.672 | 10.640 | 33.256 |
| 2-way Marginals | 67.802 | 599.843 | 7.897 | 47.741 | 52.217 | 437.478 |
| 3-way Marginals | 236.843 | 5675.238 | 7.706 | 71.549 | 156.638 | 3095.997 |
| $\leq$ 3-way Marginals | 253.605 | 6677.253 | 13.216 | 415.073 | 180.817 | 4317.709 |

# 6 Limitations, Conclusion, and Future Work.

In this paper, we introduced ResidualPlanner, a matrix mechanism that is scalable and optimal for marginals under Gaussian noise, for a large class of convex objective functions. While these are important improvements to the state of the art, there are limitations.

First, for some attributes, a user might not want marginals. For example, they might want range queries or queries with hierarchies (e.g., how many people drive sedans vs. vans; out of the sedans, how many are red vs. green, etc) [2, 28, 36]. In some cases, an attribute might have an infinite domain (e.g., a URL) and need to be handled differently [27, 45]. In other cases, the user may want other noise distributions, like the Laplace. These types of queries do not have the same type of symmetry as marginals that was crucial to proving the optimality of ResidualPlanner. For these situations, one of the key ideas of ResidualPlanner can be used – find a linear basis that compactly represents both the queries and "residual" (information provided by a query that is not contained in the other queries). Such a feature would result in scalability. It is future work to determine how to extend both scalability and optimality to such situations. Another limitation is that this work considers the setting where an individual can contribute exactly one (rather than arbitrarily many) records to the dataset.

# 7 Acknowledgments

This work was supported by NSF grant CNS-1931686 and a gift from Facebook.

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

# Contents

## A   Table of Notation

## B   A Run-through of Residual Planner

In this section, we provide a complete runthrough of ResidualPlanner using a small toy dataset.

### B.1   A Small Dataset and its Vectorized Representation

In our example, we have a dataset with 3 attributes, so $n_a = 3$. $Att_1$ takes values 'a' or 'b'; $Att_2$ takes values 'y' or 'n'; $Att_3$ takes values 1 or 2 or 3.

In this dataset, there are 5 people, and the tabular representation is shown in Table 5. For each attribute, we can one-hot encode its attribute values as row vectors. So, for $Att_1$, the attribute value 'a' is encoded as $[1, 0]$ and 'b' is encoded as $[0, 1]$. For $Att_2$, the attribute value 'y' is encoded as $[1, 0]$ and 'n' is encoded as $[0, 1]$. For attribute $Att_3$, the attribute value '1' is encoded as $[1, 0, 0]$, the value '2' is encoded as $[0, 1, 0]$ and '3' is encoded as $[0, 0, 1]$.

The kronecker product representation of a record is the kronecker product of the one-hot encoding of each attributes. So, for example, the record 'an2' is encoded as the kronecker product $[1, 0] \otimes [0, 1] \otimes [0, 1, 0]$. When this kronecker product is expanded, it has 12 components. One of the contains a 1 and the rest contain a 0. Thus the expanded kronecker product can be thought of as a one-hot encoding of the entire record.

Indeed, in the expanded kronecker product, each dimension of the resulting vector is associated with a record. In table 6, we show the kronecker product representation of each record from Table 5. The left column of Table 6 shows the record and its kronecker representation. The next 12 columns show the resulting expansion. Each record becomes as 12-dimensional vector and the column labels in Table 6 show which record is associated with which index in the 12-dimensional vector.

The sum of the kron representations of all the records is the data vector $\mathbf{x}$. It is again a 12-dimensional vector. At each index $i$, $\mathbf{x}[i]$ is the number of people whose record is associated index $i$. For example, the 5th component is associated with the record 'an2' and there are 2 people with that record. For mathematical convenience, $\mathbf{x}$ is treated as a column vector, but for display purposes, in Table 6 it is written as a row vector.

### B.2   The Marginal Workload and its Representation as a Query Matrix.

For this example, we set the marginal workload to consist of 3 marginals $Wkload = \{\{Att_1\}, \{Att_1, Att_2\}, \{Att_2, Att_3\}\}$.

The marginal on attribute set $\mathbf{A} = \{Att_1\}$ has only two cells, which correspond to the number of people with $Att_1 = a$ (i.e., 3) and the number with $Att_1 = b$ (i.e., 3). This is called a one-way marginal. The other marginals are two-way marginals because they involve two attributes. For example, the marginal on $\mathbf{A} = \{Att_2, Att_3\}$ has 6 cells. It represents the number of people for each combination of values for $Att_2$ and $Att_3$. For example, there are 2 people with $Att_2 = y$ and $Att_3 = 3$.

For each set $\mathbf{A}$, the marginal on those attributes can be represented as a matrix $\mathbf{Q_A}$ such that calculating the marginal is equivalent to the matrix-vector multiplication $\mathbf{Q_A x}$. The construction of

Table 4: Table of Notation

| | |
|---|---|
| $\mathcal{D}$: | Dataset |
| $r_i$: | $i^{\text{th}}$ record in $\mathcal{D}$ |
| $n_a$ : | number of attributes each record has |
| $Att_j$: | $j^{\text{th}}$ attribute. |
| $\lvert Att_j \rvert$: | size of the domain of attribute $Att_j$. |
| $a_1^{(j)}, \ldots, a_{\lvert Att_j \rvert}^{(j)}$: | possible values (domain) of $Att_j$. |
| $d$: | Number of possible records: $d = \prod\limits_{j=1}^{n_a} \lvert Att_j \rvert$ |
| $\mathbf{x}$: | Representation of $\mathcal{D}$ as a $d$-dimensional vector of counts (e.g., histogram) |
| $\mathbf{A}$: | (Sub)set of attributes |
| $\mathbf{Q_A}$: | Matrix representation of the marginal on $\mathbf{A}$. The counts in the marginal are the result of matrix-vector product $\mathbf{Q_A x}$. |
| $\#cells(\mathbf{A})$: | Number of cells in the marginal on $\mathbf{A}$. Equals $\prod_{Att_i \in \mathbf{A}} \lvert Att_i \rvert$ |
| $e_i$: | one-hot encoding vector with entry $i$ being 1 and the rest 0 |
| $e_{i,j}$: | equal to $e_i - e_j$ |
| $\mathbf{1}_k$: | the $k$-dimensional vector whose entries are all 1. |
| $\mathcal{I}_k$: | the $k \times k$ identity matrix |
| $\mathcal{M}$: | A privacy mechanism. |
| $\omega$: | Output of a mechanism. |
| $\mathbf{B}$: | Query matrix of a Gaussian linear query mechanism: $\mathcal{M}(\mathbf{x}) \equiv \mathbf{Bx} + N(\mathbf{0}, \mathbf{\Sigma})$ |
| $\mathbf{\Sigma}$: | Covariance matrix. |
| $pcost(\mathcal{M})$: | Privacy cost of a Gaussian linear mechanism $\mathcal{M}(\mathbf{x}) \equiv \mathbf{Bx} + N(\mathbf{0}, \mathbf{\Sigma})$. It is defined as the largest diagonal of $\mathbf{B}^T \mathbf{\Sigma}^{-1} \mathbf{B}$. Differential privacy parameters can be computed from $pcost(\mathcal{M})$. |
| $Wkload$: | A workload of marginals. Each element of $Wkload$ is a set of attributes (representing the marginal on those attributes). |
| $n_{cells}$: | Total number of cells in the marginals in the marginal workload (i.e., the output size). |
| closure($Wkload$): | The set of all subsets of $Wkload$. Formally defined as $\{\mathbf{A}' \ : \ \mathbf{A}' \subseteq \mathbf{A} \text{ for some } \mathbf{A} \in Wkload\}$. |
| $Var(\mathbf{A}, \mathcal{M})$: | When the output of $\mathcal{M}$ is used to reconstruct answers to the marginal on $\mathbf{A}$, then $Var$ returns the vector of variances of the marginal cells. |
| $\mathcal{L}$: | The loss function |
| $\dagger$: | The operator that gives the pseudo-inverse of a matrix |
| $\mathbf{Sub}_m$: | An $(m-1) \times m$ subtraction matrix. The first column is filled with 1, entries of the form $(i, i+1)$ are -1, and all other entries are 0. |
| $\mathbf{R_A}$: | Residual matrix. Given a set $\mathbf{A} \subset \{Att_1, \ldots, Att_{n_a}\}$ of attributes, $\mathbf{R_A} = \mathbf{V}_1 \otimes \cdots \otimes \mathbf{V}_{n_a}$, where $\mathbf{V}_i = \mathbf{1}_{\lvert Att_i \rvert}$ if $Att_i \notin \mathbf{A}$ and $\mathbf{V}_i = \mathbf{Sub}_{\lvert Att_i \rvert}$ if $Att_i \in \mathbf{A}$. |
| $\mathbf{\Sigma_A}$: | The covariance matrix used by the base mechanisms, formed as the kronecker product $\bigotimes\limits_{Att_i \in \mathbf{A}} (\mathbf{Sub}_{\lvert Att_i \rvert} \mathbf{Sub}_{\lvert Att_i \rvert}^T)$. Also $\mathbf{\Sigma}_\emptyset = 1$. |
| $\sigma_\mathbf{A}$: | Data-independent noise scale parameter |
| $\mathcal{M}_\mathbf{A}$: | The base mechanism defined as $\mathcal{M}_\mathbf{A}(\mathbf{x}) \equiv \mathbf{R_A x} + N(\mathbf{0}, \sigma_\mathbf{A}^2 \mathbf{\Sigma_A})$. It uses a data-independent noise parameter $\sigma_\mathbf{A}^2$ |
| $\omega_\mathbf{A}$: | noisy output of mechanism $\mathcal{M}_\mathbf{A}$ |

the matrix $\mathbf{Q_A}$ is straightforward. It is a kronecker product of 3 matrices. Each matrix corresponds to an attribute. If the attribute is in $\mathbf{A}$ then the corresponding term is the identity matrix, otherwise is is the row vector full of ones. For example, $\mathbf{Q}_{\{Att_1\}}$ is a kron product of 3 matrices: the first matrix corresponds to $Att_1$ and is the $2 \times 2$ identity matrix. The second matrix is actually the vector full of ones because $Att_2$ is not part of the marginal. This vector has 2 components because $Att_2$ has 2 possible values. Similarly, the third matrix is the vector full of ones with 3 components.

| $Att_1$ | $Att_2$ | $Att_3$ |
|---|---|---|
| a | n | 2 |
| b | n | 3 |
| b | y | 3 |
| a | n | 2 |
| b | y | 3 |

Table 5: A Toy Dataset $\mathcal{D}$

| | $ay1$ | $ay2$ | $ay3$ | $an1$ | $an2$ | $an3$ | $by1$ | $by2$ | $by3$ | $bn1$ | $bn2$ | $bn3$ |
|---|---|---|---|---|---|---|---|---|---|---|---|---|
| an2:$[1,0]\otimes[0,1]\otimes[0,1,0]$ | 0 | 0 | 0 | 0 | **1** | 0 | 0 | 0 | 0 | 0 | 0 | 0 |
| bn3: $[0,1]\otimes[0,1]\otimes[0,0,1]$ | 0 | 0 | 0 | 0 | 0 | 0 | 0 | 0 | 0 | 0 | 0 | **1** |
| by3: $[0,1]\otimes[1,0]\otimes[0,0,3]$ | 0 | 0 | 0 | 0 | 0 | 0 | 0 | 0 | **1** | 0 | 0 | 0 |
| an2: $[1,0]\otimes[0,1]\otimes[0,1,0]$ | 0 | 0 | 0 | 0 | **1** | 0 | 0 | 0 | 0 | 0 | 0 | 0 |
| by3: $[0,1]\otimes[1,0]\otimes[0,0,3]$ | 0 | 0 | 0 | 0 | 0 | 0 | 0 | 0 | **1** | 0 | 0 | 0 |
| | | | | | | | | | | | | |
| Vector of counts $\mathbf{x}$: | 0 | 0 | 0 | 0 | **2** | 0 | 0 | 0 | **2** | 0 | 0 | **1** |

Table 6: Kron product representations or each record and the whole dataset $\mathbf{x}$. Nonzero components are shown in bold red.

For the marginals in $Wkload$, these are the the corresponding matrices:

$$\mathbf{Q}_{\{Att_1\}} = \begin{bmatrix} 1 & 0 \\ 0 & 1 \end{bmatrix} \otimes \begin{bmatrix} 1 & 1 \end{bmatrix} \otimes \begin{bmatrix} 1 & 1 & 1 \end{bmatrix}$$

$$= \begin{bmatrix} 1 & 1 & 1 & 1 & 1 & 1 & 0 & 0 & 0 & 0 & 0 & 0 \\ 0 & 0 & 0 & 0 & 0 & 0 & 1 & 1 & 1 & 1 & 1 & 1 \end{bmatrix}$$

$$\mathbf{Q}_{\{Att_1,Att_2\}} = \begin{bmatrix} 1 & 0 \\ 0 & 1 \end{bmatrix} \otimes \begin{bmatrix} 1 & 0 \\ 0 & 1 \end{bmatrix} \otimes \begin{bmatrix} 1 & 1 & 1 \end{bmatrix}$$

$$= \begin{bmatrix} 1 & 1 & 1 & 0 & 0 & 0 & 0 & 0 & 0 & 0 & 0 & 0 \\ 0 & 0 & 0 & 1 & 1 & 1 & 0 & 0 & 0 & 0 & 0 & 0 \\ 0 & 0 & 0 & 0 & 0 & 0 & 1 & 1 & 1 & 0 & 0 & 0 \\ 0 & 0 & 0 & 0 & 0 & 0 & 0 & 0 & 0 & 1 & 1 & 1 \end{bmatrix}$$

$$\mathbf{Q}_{\{Att_2,Att_3\}} = \begin{bmatrix} 1 & 1 \end{bmatrix} \otimes \begin{bmatrix} 1 & 0 \\ 0 & 1 \end{bmatrix} \otimes \begin{bmatrix} 1 & 0 & 0 \\ 0 & 1 & 0 \\ 0 & 0 & 1 \end{bmatrix}$$

$$= \begin{bmatrix} 1 & 0 & 0 & 0 & 0 & 0 & 1 & 0 & 0 & 0 & 0 & 0 \\ 0 & 1 & 0 & 0 & 0 & 0 & 0 & 1 & 0 & 0 & 0 & 0 \\ 0 & 0 & 1 & 0 & 0 & 0 & 0 & 0 & 1 & 0 & 0 & 0 \\ 0 & 0 & 0 & 1 & 0 & 0 & 0 & 0 & 0 & 1 & 0 & 0 \\ 0 & 0 & 0 & 0 & 1 & 0 & 0 & 0 & 0 & 0 & 1 & 0 \\ 0 & 0 & 0 & 0 & 0 & 1 & 0 & 0 & 0 & 0 & 0 & 1 \end{bmatrix}$$

If we multiply $\mathbf{Q}_{\{attr_2,Att_3\}}$ by the data vector $\mathbf{x}$ from Table 6, we get:

$$\mathbf{Q}_{\{attr_2,Att_3\}}\mathbf{x} = \begin{bmatrix} 0 \\ 0 \\ 2 \\ 0 \\ 2 \\ 1 \end{bmatrix}$$

Comparing it to the marginals shown in Table 7 we see that it is the flattened version of the marginal. That is, we take the first column of the $\{Att_2, Att_3\}$ marginal of Table 7, then we put the next column below it, and the third column is placed at the bottom.

|   | $\mathbf{A} = \{Att_1\}$ |
|---|---|
| **a** | 2 |
| **b** | 3 |

| $\mathbf{A} = \{Att_1, Att_2\}$ | **y** | **n** |
|---|---|---|
| **a** | 0 | 2 |
| **b** | 2 | 1 |

| $\mathbf{A} = \{Att_2, Att_3\}$ | **1** | **2** | **3** |
|---|---|---|---|
| **y** | 0 | 0 | 2 |
| **n** | 0 | 2 | 1 |

Table 7: True answers to the marginal queries in the marginal workload $Wkload = \{\{Att_1\}, \{Att_1, Att_2\}, \{Att_2, Att_3\}\}$.

## B.3   The Base Mechanisms

Recall that our workload, the marginals we want privacy-preserving answers to, is $Wkload = \{\{Att_1\}, \{Att_1, Att_2\}, \{Att_2, Att_3\}\}$. Its closure, denoted by closure($Wkload$) is all of its subsets. So,

$$\text{closure}(Wkload) = \{\ \emptyset,\ \{Att_1\},\ \{Att_2\},\ \{Att_3\},\ \{Att_1, Att_2\},\ \{Att_2, Att_3\}\ \}$$

For each $\mathbf{A} \in$ closure($Wkload$) we need to form a base mechanism $\mathcal{M}_\mathbf{A}$. Each $\mathcal{M}_\mathbf{A}$ has a free parameter $\sigma_\mathbf{A}^2$ that we are free to choose. Each mechanism $\mathcal{M}_\mathbf{A}$ has the form $\mathcal{M}_\mathbf{A}(\mathbf{x}; \sigma_\mathbf{A}^2) = \mathbf{R}_\mathbf{A}\mathbf{x} + N(\mathbf{0}, \mathbf{\Sigma}_\mathbf{A})$. That is, on input $\mathbf{x}$, the mechanism multiplies it by a special "residual" matrix $\mathbf{R}_\mathbf{A}$ and then adds correlated Gaussian noise, with zero mean and with covariance matrix $\sigma_\mathbf{A}^2\mathbf{\Sigma}_\mathbf{A}$. The residual and covariance matrices for each base mechanism are shown below.

$$\mathcal{M}_\emptyset: \quad \mathbf{R}_\emptyset = [\ 1 \quad 1\ ] \otimes [\ 1 \quad 1\ ] \otimes [\ 1 \quad 1 \quad 1\ ]$$
$$\mathbf{\Sigma}_\emptyset = [1]$$

$$\mathcal{M}_{\{Att_1\}}: \quad \mathbf{R}_{\{Att_1\}} = [\ 1 \quad -1\ ] \otimes [\ 1 \quad 1\ ] \otimes [\ 1 \quad 1 \quad 1\ ]$$
$$\mathbf{\Sigma}_{\{Att_1\}} = [\ 1 \quad -1\ ]([\ 1 \quad -1\ ])^T = [2]$$

$$\mathcal{M}_{\{Att_2\}}: \quad \mathbf{R}_{\{Att_2\}} = [\ 1 \quad 1\ ] \otimes [\ 1 \quad -1\ ] \otimes [\ 1 \quad 1 \quad 1\ ]$$
$$\mathbf{\Sigma}_{\{Att_2\}} = [\ 1 \quad -1\ ]([\ 1 \quad -1\ ])^T = [2]$$

$$\mathcal{M}_{\{Att_3\}}: \quad \mathbf{R}_{\{Att_3\}} = [\ 1 \quad 1\ ] \otimes [\ 1 \quad 1\ ] \otimes \begin{bmatrix} 1 & -1 & 0 \\ 1 & 0 & -1 \end{bmatrix}$$
$$\mathbf{\Sigma}_{\{Att_3\}} = \begin{bmatrix} 1 & -1 & 0 \\ 1 & 0 & -1 \end{bmatrix} \left( \begin{bmatrix} 1 & -1 & 0 \\ 1 & 0 & -1 \end{bmatrix} \right)^T = \begin{bmatrix} 2 & 1 \\ 1 & 2 \end{bmatrix}$$

$$\mathcal{M}_{\{Att_1, Att_2\}}: \quad \mathbf{R}_{\{Att_1, Att_2\}} = [\ 1 \quad -1\ ] \otimes [\ 1 \quad -1\ ] \otimes [\ 1 \quad 1 \quad 1\ ]$$
$$\mathbf{\Sigma}_{\{Att_1, Att_2\}} = \left( [\ 1 \quad -1\ ] \otimes [\ 1 \quad -1\ ] \right) \left( [\ 1 \quad -1\ ] \otimes [\ 1 \quad -1\ ] \right)^T = [4]$$

$$\mathcal{M}_{\{Att_2, Att_3\}}: \quad \mathbf{R}_{\{Att_2, Att_3\}} = [\ 1 \quad 1\ ] \otimes [\ 1 \quad -1\ ] \otimes \begin{bmatrix} 1 & -1 & 0 \\ 1 & 0 & -1 \end{bmatrix}$$
$$\mathbf{\Sigma}_{\{Att_2, Att_3\}} = \left( [\ 1 \quad -1\ ] \otimes \begin{bmatrix} 1 & -1 & 0 \\ 1 & 0 & -1 \end{bmatrix} \right) \left( [\ 1 \quad -1\ ] \otimes \begin{bmatrix} 1 & -1 & 0 \\ 1 & 0 & -1 \end{bmatrix} \right)^T$$
$$= \begin{bmatrix} 4 & 2 \\ 2 & 4 \end{bmatrix}$$

Note that for any $\mathbf{A}$, the residual matrix $\mathbf{R}_\mathbf{A}$ has a similar structure to $\mathbf{Q}_\mathbf{A}$ except that where $\mathbf{Q}_\mathbf{A}$ has an identity matrix in its kron product, $\mathbf{R}_\mathbf{A}$ has a subtraction matrix (e.g. $[\,1\ -1\,]$ or $\left[\begin{smallmatrix} 1 & -1 & 0 \\ 1 & 0 & -1 \end{smallmatrix}\right]$). Meanwhile the covariance matrix $\mathbf{\Sigma}_\mathbf{A}$ looks like $\mathbf{R}_\mathbf{A}\mathbf{R}_\mathbf{A}^T$ except that the vectors full of 1s have been first removed.

How do we interpret the residual matrices? Well, $\mathbf{R}_\emptyset$ is the sum query. In fact the matrix vector multiplication $\mathbf{R}_\emptyset \mathbf{x}$ gives us the total number of people in the data.

Next, $\mathbf{R}_{\{Att_1\}}$ tells us the information contained in the marginal on $\{Att_1\}$ that is not contained in the sum query. If we know the total number of people in the data, then the only new information the marginal gives us is the difference between the number of people with $Att_1 = a$ and the number of people with $Att_1 = b$. In other words, $\mathbf{R}_{\{Att_1\}}\mathbf{x}$ is this difference. Given this difference, and the total, once can recover the marginal on attribute $Att_1$.

Similarly, $\mathbf{R}_{\{Att_2\}}$ contains the information in the marginal on $\{Att_2\}$ that is not provided by the sum query. Finally $\mathbf{R}_{\{Att_3\}}$ contains the information in the marginal on $\{Att_3\}$ not provided in the sum query, which is the number of people with $Att_3 = 1$ minus the number with $Att_3 = 2$, and also the number of people with $Att_3 = 1$ minus the number with $Att_3 = 3$. The product $\mathbf{R}_{\{Att_3\}}\mathbf{x}$ returns those two differences as a vector with two components.

Now, $\mathbf{R}_{\{Att_1, Att_2\}}$ and $\mathbf{R}_{\{Att_2, Att_3\}}$ are more complicated, but have the same idea. For example, $\mathbf{R}_{\{Att_1, Att_2\}}$ represents new information that the marginal on $\{Att_1, Att_2\}$ provides that is not captures by the sub-marginals (the marginal on $\{Att_1\}$ and the marginal on $\{Att_2\}$.

In general, the matrix $\mathbf{R}_\mathbf{A}$ represents the new information on that the marginal on $\mathbf{A}$ provides, which is not captured by the marginals on $\mathbf{A}'$, for $\mathbf{A}' \subset \mathbf{A}$ (strict subsets).

Now, Theorem 4.2 tells us that if we take all of the rows of all of the residual matrices, they will be linearly independent. Furthermore, given an attribute set $\mathbf{A}$, the total number of rows of $\mathbf{R}_{\mathbf{A}'}$ for all $\mathbf{A}' \subseteq \mathbf{A}$ is the number of rows in $\mathbf{Q}_\mathbf{A}$. Furthermore, the space spanned by those rows is the same as the space spanned by the rows of $\mathbf{Q}_\mathbf{A}$.

This also means that if we know $\mathbf{R}_{\mathbf{A}'}\mathbf{x}$ for all $\mathbf{A}' \subseteq \mathbf{A}$ then we can figure out $\mathbf{Q}_\mathbf{A}\mathbf{x}$ (and vice versa).

Now, we want to get privacy-preserving (noisy) answers to the marginal queries in $Wkload = \{\{Att_1\}, \{Att_1, Att_2\}, \{Att_2, Att_3\}\}$ that are as accurate as possible subject to privacy constraints. We quantify accuracy using a regular (Definition 4.1) loss function (e.g., sum of the variances of the answers to the marginals) and we quantify privacy by setting privacy parameters for either $(\epsilon, \delta)$-differential privacy, $\rho$-zCDP, or $\mu$-Gaussian differential privacy.

Theorem 4.4 says that to maximize the accuracy subject to privacy constraints, we need to take the closure of the workload, $\mathrm{closure}(Wkload) = \{\ \emptyset,\ \{Att_1\},\ \{Att_2\},\ \{Att_3\}, \{Att_1, Att_2\}, \{Att_2, Att_3\}\ \}$ and carefully choose positive numbers $\sigma_\mathbf{A}^2$ for each $\mathbf{A} \in \mathrm{closure}(Wkload)$ – so that is 6 numbers total. These numbers are chosen without looking at the data (we explain how in Section B.6). Once we have these numbers, we run the mechanisms $\mathcal{M}_\mathbf{A}(\mathbf{x}; \sigma_\mathbf{A}^2)$ and return their outputs. In other words, we must release the outputs of:

- $\mathcal{M}_\emptyset(\mathbf{x}; \sigma_\emptyset^2)$ – produces 1 number (a vector with just one component)
- $\mathcal{M}_{\{Att_1\}}(\mathbf{x}; \sigma_{\{Att_2\}}^2)$ – produces 1 number (a vector with just one component)
- $\mathcal{M}_{\{Att_2\}}(\mathbf{x}; \sigma_{\{Att_2\}}^2)$ – produces 1 number (a vector with just one component)
- $\mathcal{M}_{\{Att_3\}}(\mathbf{x}; \sigma_{\{Att_3\}}^2)$ – produces 2 numbers (a vector with 2 components)
- $\mathcal{M}_{\{Att_1, Att_2\}}(\mathbf{x}; \sigma_{\{Att_1, Att_2\}}^2)$ – produces 1 number (a vector with 1 component)
- $\mathcal{M}_{\{Att_2, Att_3\}}(\mathbf{x}; \sigma_{\{Att_2, Att_3\}}^2)$ – produces 2 numbers (a vector with 2 components)

Which gives us 8 total (noisy) numbers. In fact, any matrix mechanism for this workload must return at least 8 noisy numbers, by Theorem 4.4.

From these outputs, one can reconstruct noisy answers to the marginals in $Wkload$ (actually one can reconstruct noisy answers to any marginal in $\mathrm{closure}(Wkload)$). We show how to do this in Section B.4. Then we show how to compute the privacy cost and variances of the algorithm in Section B.5.

## B.4 Reconstruction

Let $\omega_\mathbf{A}$ denote the output of $\mathcal{M}_\mathbf{A}$. Thus, after running

- $\mathcal{M}_\emptyset(\mathbf{x}; \sigma_\emptyset^2)$

- $\mathcal{M}_{\{Att_1\}}(\mathbf{x}; \sigma^2_{\{Att_1\}})$

- $\mathcal{M}_{\{Att_2\}}(\mathbf{x}; \sigma^2_{\{Att_2\}})$

- $\mathcal{M}_{\{Att_3\}}(\mathbf{x}; \sigma^2_{\{Att_3\}})$

- $\mathcal{M}_{\{Att_1, Att_2\}}(\mathbf{x}; \sigma^2_{\{Att_1, Att_2\}})$

- and $\mathcal{M}_{\{Att_2, Att_3\}}(\mathbf{x}; \sigma^2_{\{Att_2, Att_3\}})$

we have the noisy answers

$$\omega_\emptyset, \quad \omega_{\{Att_1\}}, \quad \omega_{\{Att_2\}}, \quad \omega_{\{Att_3\}}, \quad \omega_{\{Att_1, Att_2\}}, \quad \omega_{\{Att_2, Att_3\}}$$

From these noisy answers we can produce noisy answers for any marginal in $Wkload$ or even closure($Wkload$). To reconstruct a marginal on $\mathbf{A}$, we need $\omega_{\mathbf{A'}}$ for all $\mathbf{A'} \subseteq \mathbf{A}$ – this is not a lot as these vectors represent as many noisy numbers as there are cells in the desired histogram. So, for example, if we want to get noisy answers for the marginal on $\{Att_2, Att_3\}$ (which has 6 cells), we need to use $\omega_\emptyset, \omega_{\{Att_2\}}, \omega_{\{Att_3\}}$, and $\omega_{\{Att_2, Att_3\}}$ (together these $\omega$ vectors represent a total of 6 noisy numbers).

In order to reconstruct the marginal on $\mathbf{A}$, Algorithm 2 multiplies each $\omega_{\mathbf{A'}}$ by a matrix that depends on both $\mathbf{A}$ and $\mathbf{A'}$. The algorithm calls this matrix $\mathbf{U}$, but to make the notation precise for this runthrough, we will call it $\mathbf{U}_{\mathbf{A} \leftarrow \mathbf{A'}}$ (the $\mathbf{U}$ matrix that multiplies $\omega_{\mathbf{A'}}$ when reconstructing $\mathbf{A}$). It turns out that:

$$\mathbf{Q_A x} = \sum_{\mathbf{A'} \subseteq \mathbf{A}} \mathbf{U}_{\mathbf{A} \leftarrow \mathbf{A'}} \mathbf{R_{A'} x}$$

which means that the marginal on $\mathbf{A}$ could be recreated if we know the quantities $\mathbf{R_{A'} x}$ (recall $\mathbf{R_{A'}}$ are the matrices used to define our base mechanisms). Now, since $\omega_{\mathbf{A'}}$ is a noisy version of $\mathbf{R_{A'} x}$, we can get noisy marginal answers by substituting in these noisy values into the above equation.

For example, to reconstruct a noisy answer to the marginal on $\{Att_2, Att_3\}$, we do the following:

$$
\begin{aligned}
\text{Noisy Marginal on } \{Att_2, Att_3\} = {} & (\mathbf{U}_{\{Att_2, Att_3\} \leftarrow \emptyset}) \omega_\emptyset \\
& + (\mathbf{U}_{\{Att_2, Att_3\} \leftarrow \{Att_2\}}) \omega_{\{Att_2\}} \\
& + (\mathbf{U}_{\{Att_2, Att_3\} \leftarrow \{Att_3\}}) \omega_{\{Att_3\}} \\
& + (\mathbf{U}_{\{Att_2, Att_3\} \leftarrow \{Att_2, Att_3\}}) \omega_{\{Att_2, Att_3\}}
\end{aligned}
$$

where

$$\mathbf{U}_{\{Att_2, Att_3\} \leftarrow \emptyset} = \left(\frac{1}{2}\mathbf{1}_2\right) \otimes \left(\frac{1}{3}\mathbf{1}_3\right) = \begin{bmatrix} 1/2 \\ 1/2 \end{bmatrix} \otimes \begin{bmatrix} 1/3 \\ 1/3 \\ 1/3 \end{bmatrix}$$

$$\mathbf{U}_{\{Att_2, Att_3\} \leftarrow \{Att_2\}} = \left(\mathbf{Sub}_2^\dagger\right) \otimes \left(\frac{1}{3}\mathbf{1}_3\right) = \begin{bmatrix} 1/2 \\ -1/2 \end{bmatrix} \otimes \begin{bmatrix} 1/3 \\ 1/3 \\ 1/3 \end{bmatrix}$$

$$\mathbf{U}_{\{Att_2, Att_3\} \leftarrow \{Att_3\}} = \left(\frac{1}{2}\mathbf{1}_2\right) \otimes \left(\mathbf{Sub}_3^\dagger\right) = \begin{bmatrix} 1/2 \\ 1/2 \end{bmatrix} \otimes \begin{bmatrix} 1/3 & 1/3 \\ -2/3 & 1/3 \\ 1/3 & -2/3 \end{bmatrix}$$

$$\mathbf{U}_{\{Att_2, Att_3\} \leftarrow \{Att_2, Att_3\}}) \omega_{\{Att_2, Att_3\}} = \left(\mathbf{Sub}_2^\dagger\right) \otimes \left(\mathbf{Sub}_3^\dagger\right) = \begin{bmatrix} 1/2 \\ -1/2 \end{bmatrix} \otimes \begin{bmatrix} 1/3 & 1/3 \\ -2/3 & 1/3 \\ 1/3 & -2/3 \end{bmatrix}$$

Note $\mathbf{Sub}_2^\dagger$ and $\mathbf{Sub}_3^\dagger$ are defined in Lemma 4.6.

## B.5 Privacy Cost and Marginal Variances

Recall that for a marginal workload $Wkload$, we need to run a mechanism $\mathcal{M}_\mathbf{A}$ for each $\mathbf{A} \in$ closure($Wkload$). Theorem 4.5 shows how to compute the privacy cost $pcost$ of each. In our running example, this means:

- $pcost(\mathcal{M}_\emptyset(\mathbf{x}; \sigma^2_\emptyset)) = \frac{1}{\sigma^2_\emptyset}$

- $pcost(\mathcal{M}_{\{Att_1\}}(\mathbf{x}; \sigma^2_{\{Att_1\}})) = \frac{1}{\sigma^2_{\{Att_1\}}} * \frac{1}{2}$

- $pcost(\mathcal{M}_{\{Att_2\}}(\mathbf{x}; \sigma^2_{\{Att_2\}})) = \frac{1}{\sigma^2_{\{Att_2\}}} * \frac{1}{2}$

- $pcost(\mathcal{M}_{\{Att_3\}}(\mathbf{x}; \sigma^2_{\{Att_3\}})) = \frac{1}{\sigma^2_{\{Att_3\}}} * \frac{2}{3}$

- $pcost(\mathcal{M}_{\{Att_1, Att_2\}}(\mathbf{x}; \sigma^2_{\{Att_1, Att_2\}})) = \frac{1}{\sigma^2_{\{Att_1, Att_2\}}} * \frac{1}{2} * \frac{1}{2}$

- and $pcost(\mathcal{M}_{\{Att_2, Att_3\}}(\mathbf{x}; \sigma^2_{\{Att_2, Att_3\}})) = \frac{1}{\sigma^2_{\{Att_2, Att_3\}}} * \frac{1}{2} * \frac{2}{3}$

The total privacy cost is,

$$\frac{1}{\sigma^2_{\emptyset}} + \frac{1}{2}\frac{1}{\sigma^2_{\{Att_1\}}} + \frac{1}{2}\frac{1}{\sigma^2_{\{Att_2\}}} + \frac{2}{3}\frac{1}{\sigma^2_{\{Att_3\}}} + \frac{1}{4}\frac{1}{\sigma^2_{\{Att_1, Att_2\}}} + \frac{1}{3}\frac{1}{\sigma^2_{\{Att_2, Att_3\}}}$$

Thus this is a symbolic expression in terms of the (currently unknown) noise scale parameters $\sigma^2_{\mathbf{A}}$. According to Definition 2.3, we can convert the privacy cost to the $\rho$ in $\rho$-zCDP by dividing by 2 and we can convert it to the $\mu$ from $\mu$-Gaussian DP by taking the square root.

For our running example, $Wkload = \{\{Att_1\}, \{Att_1, Att_2\}, \{Att_2, Att_3\}\}$ and we can express the variance of these marginals (after reconstruction from the noisy $\omega_{\mathbf{A}}$ answers) also in terms of the noise scale parameters. We do this with the help of Theorem 4.7.

- Marginal on $\{Att_1\}$. This marginal is reconstructed from the noisy answers $\omega_{\emptyset}$ and $\omega_{\{Att_1\}}$ and so the variance of its cells depends only on $\sigma^2_{\emptyset}$ and $\sigma^2_{\{Att_1\}}$. Applying Theorem 4.7, get that the variance in each cell of this marginal is the same and equals.

$$\left(\sigma^2_{\emptyset} * \frac{1}{2^2}\right) + \left(\sigma^2_{\{Att_1\}} * \frac{1}{2}\right)$$

- Marginal on $\{Att_1, Att_2\}$. This marginal is reconstructed from $\omega_{\emptyset}, \omega_{\{Att_1\}}, \omega_{\{Att_2\}}$, and $\omega_{\{Att_1, Att_2\}}$ and hence the variance of the cells in the marginal depend on the corresponding 4 noise scale parameters. The cell variance is

$$\left(\sigma^2_{\emptyset} * \frac{1}{2^2} * \frac{1}{2^2}\right) + \left(\sigma^2_{\{Att_1\}} * \frac{1}{2} * \frac{1}{2^2}\right) + \left(\sigma^2_{\{Att_2\}} * \frac{1}{2} * \frac{1}{2^2}\right) + \left(\sigma^2_{\{Att_1, Att_2\}} * \frac{1}{2} * \frac{1}{2}\right)$$

- Marginal on $\{Att_2, Att_3\}$. Similarly, this marginal also depends on 4 noise scale parameters as follows:

$$\left(\sigma^2_{\emptyset} * \frac{1}{2^2} * \frac{1}{3^2}\right) + \left(\sigma^2_{\{Att_2\}} * \frac{1}{2} * \frac{1}{3^2}\right) + \left(\sigma^2_{\{Att_3\}} * \frac{2}{3} * \frac{1}{2^2}\right) + \left(\sigma^2_{\{Att_2, Att_3\}} * \frac{1}{2} * \frac{2}{3}\right)$$

### B.6  The Sum-of-Variances Loss Function

Now we can express the overall privacy cost symbolically in terms of the noise scale parameters. We can also express the variance of each marginal cell symbolically. We can use these symbolic expressions to set up any regular loss function and then run it through a convex optimizer to solve it.

In this section, we give an example for the weighted sum of variances, which is one of the most popular loss functions for the matrix mechanism in research settings (mostly because this loss function is easiest to work with).

Each marginal has a weight, which we set to be 1 to avoid introducing more symbols, and the objective function is computed by adding up the cell variances in a marginal, multiplying by the weight, and adding up over the workload marginals. The marginal on $\{Att_1\}$ has two cells (so we multiply the cell variance for this marginal, computed in the previous section, by 2). The marginal on $\{Att_1, Att_2\}$ has 4 cells, and the marginal on $\{Att_2, Att_3\}$ has 6 cells. Thus, after the dust clears, the sum of the cell variances across the workload marginals is:

$$= \frac{11}{12}\sigma^2_{\emptyset} + \frac{3}{2}\sigma^2_{\{Att_1\}} + \frac{5}{6}\sigma^2_{\{Att_2\}} + \sigma^2_{\{Att_3\}} + \sigma^2_{\{Att_1, Att_2\}} + 2\sigma^2_{\{Att_2, Att_3\}}$$

Thus, we can set up the optimization problem: minimize the sum of variances subject to the privacy cost (computed in Section B.5) being less than some constant $c$:

$$\arg \min_{\substack{\sigma^2_\emptyset,\, \sigma^2_{\{Att_1\}} \\ \sigma^2_{\{Att_2\}},\, \sigma^2_{\{Att_3\}} \\ \sigma^2_{\{Att_1,Att_2\}},\, \sigma^2_{\{Att_2,Att_3\}}}} \frac{11}{12}\sigma^2_\emptyset + \frac{3}{2}\sigma^2_{\{Att_1\}} + \frac{5}{6}\sigma^2_{\{Att_2\}} + \sigma^2_{\{Att_3\}} + \sigma^2_{\{Att_1,Att_2\}} + 2\sigma^2_{\{Att_2,Att_3\}}$$

$$\text{such that } \frac{1}{\sigma^2_\emptyset} + \frac{1}{2}\frac{1}{\sigma^2_{\{Att_1\}}} + \frac{1}{2}\frac{1}{\sigma^2_{\{Att_2\}}} + \frac{2}{3}\frac{1}{\sigma^2_{\{Att_3\}}} + \frac{1}{4}\frac{1}{\sigma^2_{\{Att_1,Att_2\}}} + \frac{1}{3}\frac{1}{\sigma^2_{\{Att_2,Att_3\}}} \leq c$$

If we let the coefficient of $\sigma_\mathbf{A}$ be denoted by $v_\mathbf{A}$ and the coefficient of $1/\sigma^2_\mathbf{A}$ be denoted by $p_\mathbf{A}$, then this optimization problem can be written as:

$$\arg\min_{\sigma^2_\mathbf{A}:\ \mathbf{A}\in\text{closure}(Wkload)} \sum_{\mathbf{A}\in\text{closure}(Wkload)} v_\mathbf{A}\sigma^2_\mathbf{A}$$

$$s.t. \sum_{\mathbf{A}\in\text{closure}(Wkload)} \frac{p_\mathbf{A}}{\sigma^2_\mathbf{A}} \leq c$$

Lemma H.1 in Section H shows that the optimal solution is obtained by computing:

$$T = \left(\sum_\mathbf{A} \sqrt{v_\mathbf{A}p_\mathbf{A}}\right)^2 / c = \left(\sqrt{\frac{11}{12}*1} + \sqrt{\frac{3}{2}*\frac{1}{2}} + \sqrt{\frac{5}{6}*\frac{1}{2}} + \sqrt{2/3} + \sqrt{1/4} + \sqrt{2/3}\right)^2 / c$$

$$\approx 21.18/c$$

$$\sigma^2_\mathbf{A} = \sqrt{Tp_\mathbf{A}/(cv_\mathbf{A})} \approx \sqrt{21.18p_\mathbf{A}/v_\mathbf{A}}/c$$

$$\sigma^2_\emptyset \approx \sqrt{21.18*12/11}/c \approx 4.8/c$$

etc.

# C   Example Variance Calculations

We next illustrate how the $\sigma^2_\mathbf{A}$ parameters affect the variance of different marginals based on Theorem 4.7. We illustrate it with a toy dataset that has 5 attributes, each with 3 possible values. In this discussion, the variance of a marginal is the largest variance of its cells (all cells within the same marginal have the same variance, so the variance of a marginal is basically the variance of any cell).

EXAMPLE C.1. *The objective function is to minimize the max variance among all marginals while satisfying $\mu$-Gaussian DP with $\mu$=1. In this case all marginals end up with the same variance of 7.594 and the $\sigma^2_\mathbf{A}$ parameters are:*

- $\sigma^2_\emptyset = 7.594$

- $\sigma^2_A = 10.125$ *when $\mathbf{A}$ contains 1 attribute.*

- $\sigma^2_A = 13.5$ *when $\mathbf{A}$ contains 2 attributes.*

- $\sigma^2_A = 18$ *when $\mathbf{A}$ contains 3 attributes.*

- $\sigma^2_A = 24$ *when $\mathbf{A}$ contains 4 attributes.*

- $\sigma^2_A = 32$ *when $\mathbf{A}$ contains all of the attributes.*

EXAMPLE C.2. *The objective function is to minimize the weighted max variance among all marginals (i.e., minimize $\max_m weight_m * var(m)$). We set the weight of a marginal to be 3 if it is the 5-way marginal and 1 otherwise. This objective function basically says we want more accuracy on the 5-way marginal (it has higher weight). Again the privacy constraint is 1-Gaussian DP. In the optimal solution, each cell in the 5-way marginal has variance 2.718, the 4-way marginals have variance 5.528, and the other marginals have variance 8.154. The $\sigma^2_\mathbf{A}$ parameters are:*

- $\sigma_\emptyset^2 = 8.154$

- $\sigma_A^2 = 10.871$ *when* $\mathbf{A}$ *contains 1 attribute.*

- $\sigma_A^2 = 14.495$ *when* $\mathbf{A}$ *contains 2 attributes.*

- $\sigma_A^2 = 19.327$ *when* $\mathbf{A}$ *contains 2 attributes.*

- $\sigma_A^2 = 12.477$ *when* $\mathbf{A}$ *contains 2 attributes.*

- $\sigma_A^2 = 4.159$ *when* $\mathbf{A}$ *contains all of the attributes.*

For the variance calculation, we show this for Example C.2.

We start with the variance of a one-way marginal, say the marginal on attribute 1 (i.e., $\mathbf{A} = \{1\}$). The calculation involves $\sigma_{\{1\}}^2$ and $\sigma_\emptyset^2$ (because $\emptyset \subseteq \{1\}$) and the summation has two terms. The term involving $\sigma_\emptyset^2$ is multiplied by 1 (the first product in the variance expression for Theorem 4.7, since an empty product is 1) and that is multiplied by 1/9 (the second product, since each attribute has 3 possible values). Meanwhile, the term involving $\sigma_{\{1\}}^2$ would be multiplied by 2/3 (first product) and 1 (since the second product is empty). Thus the variance for the marginal on attribute 1 is: $8.154 * 1 * 1/9 + 10.871 * 2/3 * 1 \approx 8.154$.

For the variance on the 5-way marginal, it consists of 1 term for $\mathbf{A}' = \emptyset$, 5 terms for when $\mathbf{A}'$ has 1 attribute since there are 5 such $\mathbf{A}'$ (but all the terms have the same value), 10 terms for when $\mathbf{A}'$ has 2 attributes (all have the same value), 10 terms for when $\mathbf{A}'$ has 3 attributes (again these terms all are equal), 5 terms when $\mathbf{A}'$ has 4 attributes, and 1 term for when $\mathbf{A}' = \{1, 2, 3, 4, 5\}$. First, we note that the total number of terms is 32, which is not bad because the number of cells in the 5-way marginal is $3^5$ (i.e., much larger). The expression for the variance is:

$$
\begin{aligned}
&1 * (8.154 * (2/3)^0 * (1/9)^5) && \text{(the term involving } \sigma_\emptyset^2) \\
&+ 5 * (10.871 * (2/3)^1 * (1/9)^4) && \text{(the sum of the 5 terms having } \mathbf{A}' \text{ with 1 attribute)} \\
&+ 10 * (14.495 * (2/3)^2 * (1/9)^3) && \text{(the sum of the 10 terms having } \mathbf{A}' \text{ with 2 attributes, etc.)} \\
&+ 10 * (19.327 * (2/3)^3 * (1/9)^2) \\
&+ 5 * (12.477 * (2/3)^4 * (1/9)^1) \\
&+ 1 * (4.159 * (2/3)^5 * (1/9)^0) \\
&\approx 2.718
\end{aligned}
$$

So the main takeaway here is that when evaluating the variance of the 5-way marginal, the 1/9 terms in the products reduce the influence of any $\sigma_{\mathbf{A}'}^2$ for which $\mathbf{A}'$ has a small number of attributes. Hence the most important value is $\sigma_{\{1,2,3,4,5\}}^2$ followed by the $\sigma_{\mathbf{A}'}^2$ for the 5 sets $\mathbf{A}'$ that contain 4 attributes, etc.

# D  Optimality Proof of ResidualPlanner

In this section, we prove the optimality of ResidualPlanner. It takes advantage of the symmetry inherent in marginals and regular loss functions.

The proof sketch is the following. Given one optimal mechanism $\mathcal{M}$, we can create a variation $\widetilde{\mathcal{M}}$ of that does the following. (1) $\widetilde{\mathcal{M}}$ modifies each input record by applying some invertible function $f_i$ to each attribute $Att_i$ (for example, if $Att_i$ is a tertiary attribute, we can modify the value of $Att_i$ for each record using a function $f_i$ where $f_i(1) = 3$, $f_i(2) = 1$, $f_i(3) = 2$). This step can be viewed as simply renaming the attribute values within an attribute. (2) Then $\widetilde{\mathcal{M}}$ runs $\mathcal{M}$ on the resulting dataset. Note that marginals can be reconstructed from the output of $\widetilde{\mathcal{M}}$ by first running the reconstruction one would do for $\mathcal{M}$ and then inverting the $f_i$ functions on the resulting marginals (i.e., rearranging the cells in each marginal to undo the within-attribute renaming caused by the $f_i$). This variation $\widetilde{\mathcal{M}}$ has the same privacy properties as $\mathcal{M}$ and the same loss (due to the regularity condition on the loss).

Hence $\widetilde{\mathcal{M}}$ is also optimal. Then we create yet another optimal privacy mechanism $\mathcal{M}^*$ that splits the privacy budget across all variations of $\mathcal{M}$ and returns their outputs. It turns out that the privacy cost matrix of $\mathcal{M}^*$ has eigenvectors that are equal to the rows of the residual matrices $\mathbf{R_A}$ used by ResidualPlanner. Rewriting the privacy cost matrix of $\mathcal{M}^*$ using this eigendecomposition, we create another mechanism (the mechanism that runs the base mechanisms of ResidualPlanner) that has the same privacy cost matrix and the same value for the loss and hence is optimal.

The rest of this section explains these steps in details with formal proofs and running commentary that helps to better understand the notation and constructs in the proof.

## D.1 Notation Review

We first start with a review of key notation. Recall that a dataset $\mathcal{D} = \{r_1, \ldots, r_n\}$ is a collection of records. Each record $r_i$ contains attributes $Att_1, \ldots, Att_{n_a}$ and each attribute $Att_j$ can take values $a_1^{(j)}, \ldots, a_{|Att_j|}^{(j)}$.

An attribute value $a_i^{(j)}$ for attribute $Att_j$ can be represented as a vector using one-hot encoding. Specifically, let $e_i^{(j)}$ be a row vector of size $|Att_j|$ with a one in component $i$ and 0 everywhere else. In this way, $e_i^{(j)}$ is a representation of $a_i^{(j)}$.

A record $r$ with attributes $Att_1 = a_{i_1}^{(1)}$, $Att_2 = a_{i_2}^{(2)}, \ldots, Att_{n_a} = a_{i_{n_a}}^{(n_a)}$ can thus be represented as the kron product $e_{i_1}^{(1)} \otimes e_{i_2}^{(2)} \otimes \cdots \otimes e_{i_{n_a}}^{(n_a)}$. This vector has a 1 in exactly one position and 0s everywhere else. The position of the 1 is the *index* of record $r$.

Thus, a data vector $\mathbf{x}$ is a vector of integers. The value at index $i$ is the number of times the record associated with index $i$ appears in $\mathcal{D}$.

## D.2 Permutations

For each attribute $Att_i$, let $\Pi^{(i)}$ be the set of permutations on the numbers $1, \ldots, |Att_i|$, so that each $\pi \in \Pi^{(i)}$ can be interpreted as a permutation (or renaming) of the attributes values of $Att_i$. We can also view $\pi$ as a function on vectors of size $|Att_i|$ that permutes their coordinates. That is, the $i^{\text{th}}$ coordinate of a vector $\mathbf{y}$ is the $\pi(i)^{\text{th}}$ coordinate of $\pi(\mathbf{y})$.

One can select a permutation for each attribute $\pi^{(1)} \in \Pi^{(1)}, \ldots, \pi^{(n_a)} \in \Pi^{(n_a)}$ and use it to define a permutation over records. This permutation maps a record represented by the kron product $e_{i_1}^{(1)} \otimes e_{i_2}^{(2)} \otimes \cdots \otimes e_{i_{n_a}}^{(n_a)}$ into $\pi^{(1)}(e_{i_1}^{(1)}) \otimes \pi^{(2)}(e_{i_2}^{(2)}) \otimes \cdots \otimes \pi^{(n_a)}(e_{i_{n_a}}^{(n_a)})$. We can think of this permutation $\pi = (\pi^{(1)}, \ldots, \pi^{(n_a)})$ as a function that independently renames each attribute value in a record. Thus this permutation can be extended to datavectors $\mathbf{x}$. The value of $\mathbf{x}$ at the index associated with record $r$ is the value of $\pi(\mathbf{x})$ at the index associated with record $\pi(r)$. Another way to look at it is that $\pi(\mathbf{x})$ is the histogram associated with the dataset $\{\pi(r_1), \pi(r_2), \ldots, \pi(r_n)\}$. This permutation can be represented as a permutation matrix $\mathbf{W}_\pi$ such that $\mathbf{W}_\pi \mathbf{x} = \pi(\mathbf{x})$.

We let $\Pi = \Pi^{(1)} \times \cdots \times \Pi^{(n_a)}$ be the set of all such permutations. We call this the space of *renaming* permutations since each $\pi \in \Pi$ renames the values of each attribute separately.

Our first result is that permutation does not affect the privacy parameters of a mechanism.

LEMMA D.1. *Let $\mathcal{M}(\mathbf{x}) \equiv \mathbf{Bx} + N(\mathbf{0}, \boldsymbol{\Sigma})$ be a mechanism that satisfies $\rho$-zCDP, $(\epsilon, \delta)$-approximate DP, and $\mu$-Gaussian DP. Let $\pi$ be a permutation of the indices of $\mathbf{x}$ and $\mathbf{W}_\pi$ the corresponding permutation matrix. Then $\mathcal{M}_\pi(\mathbf{x}) \equiv \mathbf{B}\mathbf{W}_\pi \mathbf{x} + N(\mathbf{0}, \boldsymbol{\Sigma})$ satisfies $\rho$-zCDP, $(\epsilon, \delta)$-approximate DP, and $\mu$-Gaussian DP (i.e., with the same privacy parameters).*

*Proof.* The privacy cost $pcost(\mathcal{M})$ of $\mathcal{M}$ is the largest diagonal of $\mathbf{B}^T \boldsymbol{\Sigma}^{-1} \mathbf{B}$. The privacy cost $pcost(\mathcal{M}_\pi)$ of $\mathcal{M}_\pi$ is the largest diagonal of $\mathbf{W}_\pi^T \mathbf{B}^T \boldsymbol{\Sigma}^{-1} \mathbf{B} \mathbf{W}_\pi$. The effect of $\mathbf{W}_\pi$ on both sides is to permute the rows and columns of $\mathbf{B}^T \boldsymbol{\Sigma}^{-1} \mathbf{B}$ in the same way. Thus the diagonals of $\mathbf{B}^T \boldsymbol{\Sigma}^{-1} \mathbf{B}$ and $\mathbf{W}_\pi^T \mathbf{B}^T \boldsymbol{\Sigma}^{-1} \mathbf{B} \mathbf{W}_\pi$ are the same up to permutation and hence $\mathcal{M}$ and $\mathcal{M}_\pi$ have the same privacy cost and therefore the same privacy parameters. $\square$

The next result is that a renaming permutation preserves the accuracy of a marginal derived from the answer to a mechanism.

LEMMA D.2. *Let $Wkload = \{\mathbf{A}_1, \ldots, \mathbf{A}_k\}$ be a workload on marginals. Let $\mathcal{M}(\mathbf{x}) \equiv \mathbf{Bx} + N(\mathbf{0}, \mathbf{\Sigma})$ be a mechanism whose output can be used to provide unbiased estimates of those marginals. Let $\pi \in \Pi$ be a renaming permutation and $\mathbf{W}_\pi$ the corresponding permutation matrix. Define $\mathcal{M}_\pi(\mathbf{x}) \equiv \mathbf{BW}_\pi\mathbf{x} + N(\mathbf{0}, \mathbf{\Sigma})$. Then unbiased answers to $Wkload$ can be obtained from the output of $\mathcal{M}_\pi$ and for any regular loss function $\mathcal{L}$ (Definition 4.1), $\mathcal{L}(Var(\mathbf{A}_1; \mathcal{M}), \ldots, Var(\mathbf{A}_k; \mathcal{M})) = \mathcal{L}(Var(\mathbf{A}_1; \mathcal{M}_\pi), \ldots, Var(\mathbf{A}_k; \mathcal{M}_\pi))$*

*Proof.* For each set of attributes $\mathbf{A}_i \in Wkload$, let $\mathbf{Q}_{\mathbf{A}_i}$ be the query matrix of the marginal (i.e., the true marginal is computed as $\mathbf{Q}_{\mathbf{A}_i}\mathbf{x}$). Then the best linear unbiased estimate of the marginal on $\mathbf{A}_i$ from the output $\omega$ of $\mathcal{M}$ is $\mathbf{Q}_{\mathbf{A}_i}(\mathbf{B}^T\mathbf{\Sigma}^{-1}\mathbf{B})^\dagger\mathbf{B}^T\mathbf{\Sigma}^{-1}\omega$ and $Var(\mathbf{A}_i; \mathcal{M})$ is the diagonal of the covariance matrix of this estimate, which is $\mathbf{Q}_{\mathbf{A}_i}(\mathbf{B}^T\mathbf{\Sigma}^{-1}\mathbf{B})^\dagger\mathbf{Q}_{\mathbf{A}_i}^T$. Meanwhile, the best linear unbiased estimate of the marginal on $\mathbf{A}_i$ from the output $\omega'$ of $\mathcal{M}_\pi$ is is $\mathbf{Q}_{\mathbf{A}_i}(\mathbf{W}_\pi^T\mathbf{B}^T\mathbf{\Sigma}^{-1}\mathbf{BW}_\pi)^\dagger\mathbf{W}_\pi^T\mathbf{B}^T\mathbf{\Sigma}^{-1}\omega'$ and $Var(\mathbf{A}_i; \mathcal{M})$ is the diagonal of $\mathbf{Q}_{\mathbf{A}_i}(\mathbf{W}_\pi^T\mathbf{B}^T\mathbf{\Sigma}^{-1}\mathbf{BW}_\pi)^\dagger\mathbf{Q}_{\mathbf{A}_i}^T = \mathbf{Q}_{\mathbf{A}_i}\mathbf{W}_\pi^T(\mathbf{B}^T\mathbf{\Sigma}^{-1}\mathbf{B})^\dagger\mathbf{W}_\pi\mathbf{Q}_{\mathbf{A}_i}^T$.

We note that $\mathbf{Q}_{\mathbf{A}_i}\mathbf{W}_\pi^T$ is a permutation of the rows of $\mathbf{Q}_{\mathbf{A}_i}$ (computing a marginal on a dataset in which attribute values within the same attribute are renamed is the same as computing the marginal on the original dataset and then renaming the marginal cells, which is permutation of the output of the marginal computation).

Therefore the diagonals of $\mathbf{Q}_{\mathbf{A}_i}(\mathbf{B}^T\mathbf{\Sigma}^{-1}\mathbf{B})^\dagger\mathbf{Q}_{\mathbf{A}_i}^T$ and $\mathbf{Q}_{\mathbf{A}_i}(\mathbf{W}_\pi^T\mathbf{B}^T\mathbf{\Sigma}^{-1}\mathbf{BW}_\pi)^\dagger\mathbf{Q}_{\mathbf{A}_i}^T$ are the same up to permutation. Hence the vector $Var(\mathbf{A}_i; \mathcal{M})$ is the same as the vector $Var(\mathbf{A}_i; \mathcal{M}_\pi)$ up to permutation of the components, and hence does not affect a regular loss function $\mathcal{L}$. $\square$

Finally, we show that there exists an optimal mechanism whose privacy cost matrix exhibits symmetries defined by the set of permutaitons $\Pi$.

LEMMA D.3. *Let $Wkload = \{\mathbf{A}_1 \ldots, \mathbf{A}_k\}$ be a workload of marginal queries. Let $\mathcal{L}$ be a regular loss function. Let $U$ be the set of all Gaussian linear mechanisms that can provide unbiased answers to the marginals in the $Wkload$. Let $\gamma$ be a real number. Then whenever either of the following optimization problems are feasible,*

$$\min_{\mathcal{M} \in U} pcost(\mathcal{M}) \quad s.t. \; \mathcal{L}(Var(\mathbf{A}_1; \mathcal{M}), \ldots, Var(\mathbf{A}_k; \mathcal{M})) \leq \gamma$$

$$\min_{\mathcal{M} \in U} \mathcal{L}(Var(\mathbf{A}_1; \mathcal{M}), \ldots, Var(\mathbf{A}_k; \mathcal{M})) \quad s.t. \; pcost(\mathcal{M}) \leq \gamma$$

*the feasible optimization problem is minimized by some mechanism of the form $\overline{\mathcal{M}}(\mathbf{x}) \equiv \overline{\mathbf{B}}\mathbf{x} + N(\mathbf{0}, \overline{\mathbf{\Sigma}})$ whose privacy cost matrix $\mathbf{\Gamma} \equiv \overline{\mathbf{B}}^T\overline{\mathbf{\Sigma}}^{-1}\overline{\mathbf{B}}$ has the following symmetries: for all renaming permutations $\pi \in \Pi$ (with $\mathbf{W}_\pi$ being the associated permutation matrix), we have $\mathbf{\Gamma} = \mathbf{W}_\pi^T\mathbf{\Gamma}\mathbf{W}_\pi$ (in other words, permuting the rows has no effect as long as the columns are permuted in the same way).*

*Proof.* Let $\mathcal{M}_{opt}(\mathbf{x}) \equiv \mathbf{B}_{opt}\mathbf{x} + N(\mathbf{0}, \mathbf{\Sigma}_{opt})$ be an optimal mechanism to one of these problems. It may not have the required symmetries, but from it we will construct an optimal mechanism that does.

For a permutation $\pi$ (and corresponding permutation matrix $\mathbf{W}_\pi$) and a positive number $\lambda$, consider the mechanism $\mathcal{M}_{\pi,\lambda}(\mathbf{x}) \equiv \mathbf{B}_{opt}\mathbf{W}_\pi\mathbf{x} + N(\mathbf{0}, \lambda\mathbf{\Sigma}_{opt})$. By Lemma D.2, this mechanism also answers the marginals in $Wkload$.

Now consider the mechanism $\overline{\mathcal{M}}$ which, on input $\mathbf{x}$ outputs the result of $\mathcal{M}_{\pi,|\Pi|}$ for all $\pi \in \Pi$.

The query matrix of $\overline{\mathcal{M}}$ is $\overline{\mathbf{B}} = \begin{bmatrix} \mathbf{B}_{opt}\mathbf{W}_{\pi_1} \\ \vdots \\ \mathbf{B}_{opt}\dot{\mathbf{W}}_{\pi_{|\Pi|}} \end{bmatrix}$ and the covariance matrix $\overline{\mathbf{\Sigma}}$ is a block diagonal matrix with the scaled matrix $|\Pi|\mathbf{\Sigma}_{opt}$ in each block. Clearly, by Lemma D.2, it also provides unbiased answers to the marginals in $Wkload$.

First, we claim that the $pcost(\overline{\mathcal{M}}) \leq pcost(\mathcal{M}_{opt})$ so that the privacy parameters are at least as good. Recall $pcost(\overline{\mathcal{M}})$ is the largest diagonal entry of:

$$\overline{\mathbf{B}}^T \overline{\boldsymbol{\Sigma}}^{-1} \overline{\mathbf{B}} = \frac{1}{|\Pi|} \sum_{\pi \in \Pi} \mathbf{W}_\pi^T \mathbf{B}_{opt}^T \boldsymbol{\Sigma}_{opt}^{-1} \mathbf{B}_{opt} \mathbf{W}_\pi, \tag{4}$$

Since the privacy cost $pcost(\mathcal{M}_{\pi,1})$ is the largest diagonal of $\mathbf{W}_\pi^T \mathbf{B}_{opt}^T \boldsymbol{\Sigma}_{opt}^{-1} \mathbf{B}_{opt} \mathbf{W}_\pi$ and equals $pcost(\mathcal{M}_{opt})$, Equation 4 (and convexity of the max function) shows that the $pcost(\overline{\mathcal{M}}) \leq pcost(\mathcal{M}_{opt})$.

Next we consider the loss function. Let $\mathbf{A}_i \in Wkload$ be a set of attributes and let $\mathbf{Q}_{\mathbf{A}_i}$ be the corresponding query matrix for the marginal on $\mathbf{A}_i$. Then the reconstructed variances of the answers to this marginal, based on the output of $\overline{\mathcal{M}}$ is:

$$Var(\mathbf{A}_i; \overline{\mathcal{M}}) = diag \left( \mathbf{Q}_{\mathbf{A}_i} (\overline{\mathbf{B}}^T \overline{\boldsymbol{\Sigma}}^{-1} \overline{\mathbf{B}})^\dagger \mathbf{Q}_{\mathbf{A}_i}^T \right)$$

$$= diag \left( \frac{1}{|\Pi|} \sum_{\pi \in \Pi} \mathbf{Q}_{\mathbf{A}_i} \left( \mathbf{W}_\pi^T \mathbf{B}_{opt}^T \boldsymbol{\Sigma}^{-1} \mathbf{B}_{opt} \mathbf{W}_\pi \right)^\dagger \mathbf{Q}_{\mathbf{A}_i}^T \right)$$

$$= \frac{1}{|\Pi|} \sum_{\pi \in \Pi} Var(\mathbf{A}_i; \mathcal{M}_{\pi,1})$$

For any $\pi \in \Pi$, Lemma D.2 tells us that $\mathcal{L}(Var(\mathbf{A}_1; \mathcal{M}_{opt}), \ldots, Var(\mathbf{A}_k; \mathcal{M}_{opt})) = \mathcal{L}(Var(\mathbf{A}_1; \mathcal{M}_{\pi,1}), \ldots, Var(\mathbf{A}_k; \mathcal{M}_{\pi,1}))$ and so regularity of $\mathcal{L}$ (which includes convexity), means that $\mathcal{L}(Var(\mathbf{A}_1; \overline{\mathcal{M}}), \ldots, Var(\mathbf{A}_k; \overline{\mathcal{M}})) \leq \mathcal{L}(Var(\mathbf{A}_1; \mathcal{M}_{opt}), \ldots, Var(\mathbf{A}_k; \mathcal{M}_{opt}))$.

Thus $\overline{\mathcal{M}}$ is no worse in privacy or utility than $\mathcal{M}_{opt}$ and hence is optimal.

Thus we consider the symmetries of the privacy cost matrix of $\overline{\mathcal{M}}$, which is given in Equation 4. Clearly it has the desired symmetry property that $\boldsymbol{\Gamma} = \mathbf{W}_\pi^T \boldsymbol{\Gamma} \mathbf{W}_\pi$ for any $\pi \in \Pi$ as the permutation space $\Pi$ is an algebraic group.

$\square$

## D.3 From permutations to interpretations

Let $\mathcal{M}_{opt}(\mathbf{x}) \equiv \mathbf{B}_{opt}\mathbf{x} + N(\mathbf{0}, \boldsymbol{\Sigma}_{opt})$ be an optimal mechanism that has the symmetries guaranteed by Lemma D.3. Our goal is to use the symmetries in the privacy cost matrix $\boldsymbol{\Gamma}_{opt} \equiv \mathbf{B}_{opt}^T \boldsymbol{\Sigma}_{opt}^{-1} \mathbf{B}_{opt}$ to examine the structure of $\boldsymbol{\Gamma}_{opt}$.

If $\gamma_{i,j}$ is the $(i,j)^{\text{th}}$ entry of $\boldsymbol{\Gamma}_{opt}$ and if there is a renaming permutation that maps $r_i$ (the record associated with index $i$) to some $r_{i'}$ (at index $i'$) and maps $r_j$ to some $r_{j'}$ then $\gamma_{i,j} = \gamma_{i',j'}$. Note that if $r_i$ and $r_j$ have the same values for attributes $Att_1$ and $Att_2$ then $r_{i'}$ and $r_{j'}$ must match on the same attributes because renaming permutations just change the names of values within each attribute. Thus we introduce notation for the set of attributes on which two records match:

DEFINITION D.4 (Common Attributes). *Define $\zeta$ to be the function that takes two records and outputs the set of attributes on which they match We emphasize that $\zeta(r_i, r_j)$ is a set of attributes, not attribute values.*

This discussion leads to the following result which characterizes the privacy cost matrix of an optimal mechanism.

LEMMA D.5. *Under the same conditions as Lemma D.3, there exists an optimal mechanism with a privacy cost matrix $\boldsymbol{\Gamma}_{opt}$ for which the following holds. In addition to the symmetry guaranteed by Lemma D.3, for every subset of attributes $S \subseteq \{Att_1, \ldots, Att_{n_a}\}$, there exists a number $c_S$ such that $\gamma_{i,j}$, the $(i,j)^{th}$ entry of $\boldsymbol{\Gamma}_{opt}$, is equal to $c_{\zeta(r_i, r_j)}$. In other words, the $(i,j)^{th}$ entry is completely determined by the set $\zeta(r_i, r_j)$ (recall $r_i$ the record value associated with index $i$ and $r_j$ is the record value associated with index $j$).*

*Proof.* By Lemma D.3, there exists an optimal mechanism with privacy cost matrix $\mathbf{\Gamma}_{opt}$ that is invariant under renaming permutations of its rows as long as the columns are permuted in the same way. Thus if $r_i$ is the record value corresponding to position $i$ and $r_j$ is the record value corresponding to position $j$, there exists a renaming permutation that maps $r_i$ to some $r_{i'}$ and $r_j$ to some $r'_j$ if and only if the attributes on which $r_i$ and $r_j$ match are the same as the attributes on which $r_{i'}$ and $r_{j'}$ match each other (in symbols: $\zeta(r_i, r_j) = \zeta(r_{i'}, r_{j'})$). When there exists such a renaming permutation then $\gamma_{i,j} = \gamma_{i',j'}$. Thus the value of $\gamma_{i,j}$ is completely determined by $\zeta(r_i, r_j)$ and the result follows. □

From Theorem 4.2, we know that the rows of the matrices of $\mathbf{R_A}$, for all $\mathbf{A} \subseteq \{Att_1, \ldots, Att_{n_a}\}$ are a linearly independent basis for $\mathbb{R}^d$, where $d = \prod_{i=1}^{n_a} |Att_i|$. Thus we call the rows a *residual basis*.

DEFINITION D.6. *A row vector* $\mathbf{v}$ *is a* residual basis vector *if it is a row in* $\mathbf{R_A}$ *for some* $\mathbf{A} \subseteq \{Att_1, \ldots, Att_{n_a}\}$.

We now provide an interpretation of the residual bases. First, for an attribute $Att_\ell$, define the vector $e_{i,j}^{(\ell)}$ to be a vector of length $|Att_\ell|$ such that the element at position $i$ is 1, the element at position $j$ is -1 and everywhere else is 0. In other words, $e_{i,j}^{(\ell)} = e_i^{(\ell)} - e_j^{(\ell)}$ (recall $e_i^{(\ell)}$ is 1 in position $i$ and 0 everywhere else and is a one-hot encoding of the attribute $a_i^{(\ell)}$). Now, each element of the residual basis has the form $\mathbf{v}^{(1)} \otimes \cdots \otimes \mathbf{v}^{(n_a)}$ where, for each $\ell$, $\mathbf{v}^{(\ell)}$ is either the vector $\mathbf{1}_{|Att_\ell|}^T$ or a vector $e_{1,i_\ell}^{(\ell)}$. When the vector for attribute $Att_\ell$ is the vector $\mathbf{1}_{|Att_\ell|}^T$, we say that all attribute values of $Att_\ell$ are *selected*. When the vector for $Att_\ell$ is $e_{1,i_\ell}^{(\ell)}$, then we say attribute value $a_1^{(\ell)}$ is *positively selected* and $a_{i_\ell}^{(\ell)}$ is *negatively selected* (the other attribute values of $Att_\ell$ are not selected at all). The attributes for which the kron term is not $\mathbf{1}_{|Att_\ell|}^T$ are called the *discriminative* attributes.

As an example of this notation and terminology, consider Table 8. Suppose we have three attributes: $Att_1$ takes values 'a' or 'b'; $Att_2$ takes values 'y' or 'n'; $Att_3$ takes values 1 or 2 or 3.

| | ay1 | ay2 | ay3 | an1 | an2 | an3 | by1 | by2 | by3 | bn1 | bn2 | bn3 |
|---|---|---|---|---|---|---|---|---|---|---|---|---|
| bn1: $[0,1] \otimes [0,1] \otimes [1,0,0]$ | 0 | 0 | 0 | 0 | 0 | 0 | 0 | 0 | 0 | 1 | 0 | 0 |
| $[1,1] \otimes [1,-1] \otimes [1,-1,0]$ | 1 | -1 | 0 | -1 | 1 | 0 | 1 | -1 | 0 | -1 | 1 | 0 |
| $[1,-1] \otimes [1,1] \otimes [1,0,-1]$ | 1 | 0 | -1 | 1 | 0 | -1 | -1 | 0 | 1 | -1 | 0 | 1 |

Table 8: Kron product representations.

In this case, the data vector $\mathbf{x}$ would have 12 components. The first component corresponds to the number of appearances of record "a,y,1" in the dataset, the second component corresponds to record "a,y,2" and so on. The records corresponding to each index of $\mathbf{x}$ are listed in order as the column headings in Table 8. The first row shows the representation of record "b,n,1" which is composed of the second value (b) for $Att_1$, the second value (n) for $Att_2$ and the first value (1) for $Att_3$. Hence its kron representation is $[0,1] \otimes [0,1] \otimes [1,0,0]$ and when the kron product is evaluated, the resulting vector has a 1 in the index corresponding to "bn1" (10th column) and 0 everywhere else.

The second and third rows show the expansions of two residual basis vectors $[1,1] \otimes [1,-1] \otimes [1,-1,0]$ (its discriminative attributes are $Att_2$ and $Att_3$) and $[1,-1] \otimes [1,1] \otimes [1,0,-1]$ (its discriminative attributes are $Att_1$ and $Att_3$). Consider again the kron product $[1,1] \otimes [1,-1] \otimes [1,-1,0]$. Note that the first part of the kron product, $[1,1]$ refers to the first attribute and selects both of its values (sets them to 1). The second part of the kron product $[1,-1]$ refers to the $Att_2$ and positively selects the first attribute value 'y' (sets it to 1) and negatively selected the second attribute value 'n' (sets it to -1). The third part is $[1,-1,0]$ and it positively selects the first attribute value, negatively selects the second, but the third attribute value is not selected at all (i.e., the 3rd position is 0). These attribute selections can help us determine what the kron product looks like when it is expanded as follows. For the residual basis vector $\mathbf{v}^{(1)} \otimes \cdots \otimes \mathbf{v}^{(n_a)}$ the value at the index associated with a record $r$ is

- 0 if $r$ has an attribute whose value is not selected by the residual basis vector's kron product. In this case we say the residual basis vector assigns a 0 to record $r$. For example, in the residual basis vector corresponding to kron product $[1,1] \otimes [1,-1] \otimes [1,-1,0]$, the third value of the third attribute is not selected. For any record that assigns the attribute value 3 to $Att_3$, this residual basis vector assigns a 0 to such a record.

- 1 if for every attribute, the value assigned to it by $r$ is selected (posititvely or negatively), and the number of negatively selected attribute values is even. In this case we say the residual basis vector assigns a 1 to record $r$.

- -1 if the attribute value for each attribute is selected, and the number of negatively selected attribute values is odd. In this case we say the residual basis vector assigns a $-1$ to record $r$.

For example, for the residual basis vector $[1,1] \otimes [1,-1] \otimes [1,-1,0]$, the attribute value 3 for $Att_3$ is not selected. Hence the value at indices corresponding to records an3,bn3,ay3,by3 are all 0 (see Table 8). Next, consider the record an2. The value "a" is positively selected, "n" is negatively selected, and "2" is negatively selected. Hence all attributes are selected and an even number of attributes are negatively selected. Therefore the value at the index associated with an2 is 1. Now for the record by2. The "b" is positively selected, "y" is positively selected, and "2" is negatively selected. Hence there are an odd number of negative selections and so the value at the index associated with by2 is -1.

With this discussion and associated notation, we can now show that each residual basis vector is an eigenvector of the optimal privacy cost matrix, and the eigenvalue only depends on which attributes are discriminative.

THEOREM D.7. *Under the same conditions as Lemma D.3, there exists an optimal mechanism such that the eigenvectors of its privacy cost matrix $\mathbf{\Gamma}$ are the residual basis vectors (Definition D.6). Furthermore, if two residual basis vectors $\mathbf{v}^{(1)} \otimes \cdots \otimes \mathbf{v}^{(n_a)}$ and $\mathbf{w}^{(1)} \otimes \cdots \otimes \mathbf{w}^{(n_a)}$ have the same discriminative attributes (i.e., for all $i$, $\mathbf{w}^{(i)} \neq \mathbf{1}_{|Att_i|}^T$ if and only if $\mathbf{v}^{(i)} \neq \mathbf{1}_{|Att_i|}^T$) then the two residual basis vectors have the same eigenvalues (in other words, all rows of the same residual matrix have the same eigenvalues).*

*Proof.* Recall from Definition D.4 that $\zeta(r_i, r_j)$ is the set of attributes on which $r_i$ and $r_j$ are equal.

Let $\mathbf{\Gamma}$ be the privacy cost matrix guaranteed by Lemma D.5 with the properties guaranteed by Lemma D.5, namely that for every subset of attributes $S \subseteq \{Att_1, \ldots, Att_{n_a}\}$, there exists a number $c_S$ such that $\gamma_{i,j}$, the $(i,j)^{\text{th}}$ entry of $\mathbf{\Gamma}$, is equal to $c_{\zeta(r_i,r_j)}$ – the constant associated with the set $\zeta(r_i, r_j)$, where $r_i$ the record value associated with index $i$ and $r_j$ is the record value associated with index $j$.

Let $r_\ell$ be a record associated with index $\ell$. We consider the dot product between a residual basis vector $\mathbf{v} = \mathbf{v}^{(1)} \otimes \cdots \otimes \mathbf{v}^{(n_a)}$ and the $\ell^{\text{th}}$ row of $\mathbf{\Gamma}$. Since the entries of the $\ell^{th}$ row are $c_{\zeta(r_\ell, r_1)}, \ldots, c_{\zeta(r_\ell, r_d)}$ and the entries of $\mathbf{v}$ are 0,1,-1, this dot product can be expressed as:

$$\sum_{\substack{r \text{ assigned} \\ \text{value 1 by } \mathbf{v}}} c_{\zeta(r_\ell, r)} - \sum_{\substack{r \text{ assigned} \\ \text{value -1 by } \mathbf{v}}} c_{\zeta(r_\ell, r)} \tag{5}$$

We analyze this in three cases.

**Case 1: $\mathbf{v}$ assigns a $0$ to $r_\ell$.** In this case, there is an attribute for which $r_\ell$ has a value that is not selected. Without loss of generality, we may assume this is the first attribute $Att_1$ so that $\mathbf{v}^{(1)} = e_{1,i}$ (the vector with a 1 at the first index and -1 at the $i^{\text{th}}$ index for some $i$ and 0 everywhere else) and the value of $Att_1$ for $r_\ell$ is therefore not $a_1^{(1)}$ or $a_i^{(1)}$ (because $r_\ell$ got assigned 0 by $\mathbf{v}$ due to attribute $Att_1$). Now, if a record $r$ appears in the left summation of Equation 5 then its value for $Att_1$ is either $a_1^{(1)}$ or $a_i^{(1)}$ and it does not match $r_\ell$ on the first attribute. But this means that we can transform $r$ into a record $r'$ by replacing $a_1^{(1)}$ and $a_i^{(1)}$ with each other. This $r'$ would be on the right hand side of the summation (because we are flipping the sign of the selection by $\mathbf{v}$ of attribute $Att_1$ in $r'$). Furthermore $r'$ also does not match $r_\ell$ on $Att_1$ and therefore $r$ matches $r_\ell$ on exactly the same attributes as $r'$ matches $r_\ell$. Thus $\zeta(r_\ell, r) = \zeta(r_\ell, r')$. Thus the summation term from record $r$ is cancelled out by $r'$ in Equation 5. Using the same argument, we see that every term in the left summation is canceled out by a unique term in the right summation, and vice versa. Hence, if $\mathbf{v}$ assigns a 0 to record $r_\ell$ (i.e., has a 0 in index $\ell$ when its kron product representation is expanded) then the dot product between $\mathbf{v}$ and the $\ell^{\text{th}}$ row of $\mathbf{\Gamma}$ is 0.

**Case 2: $\mathbf{v}$ assigns a $1$ to $r_\ell$.** In this case, every attribute of $r_\ell$ has a value that is (either positively or negatively) selected by $\mathbf{v}$ and an even number are negatively selected. Our goal is to show that if

some other record $r_t$ is also assigned a 1 by $\mathbf{v}$, then the dot product between $\mathbf{v}$ and $\ell^{\text{th}}$ row of $\boldsymbol{\Gamma}$ is the same as the dot product between $\mathbf{v}$ and the $t^{\text{th}}$ row of $\boldsymbol{\Gamma}$. That is, we want to show:

$$\sum_{\substack{r \text{ assigned} \\ \text{value 1 by } \mathbf{v}}} c_{\zeta(r_\ell, r)} - \sum_{\substack{r \text{ assigned} \\ \text{value -1 by } \mathbf{v}}} c_{\zeta(r_\ell, r)} = \sum_{\substack{r \text{ assigned} \\ \text{value 1 by } \mathbf{v}}} c_{\zeta(r_t, r)} - \sum_{\substack{r \text{ assigned} \\ \text{value -1 by } \mathbf{v}}} c_{\zeta(r_t, r)} \tag{6}$$

Let $S$ be the set of attributes on which $r_\ell$ and $r_t$ disagree. Now define a mapping $\phi$ between records such that $\phi$ only modifies attributes in $S$. For each attribute $Att$ in $S$, it maps the value that record $r_\ell$ has into the value that $r_t$ has an vice versa. (For example, suppose $S = \{Att_1, Att_2\}$ and $r_\ell$ has values $a_2^{(1)}$ and $a_3^{(2)}$ for those attributes, respectively, and suppose that $r_t$ has values $a_4^{(1)}$ and $a_5^{(2)}$ for those attributes. Then $\phi$ changes $a_2^{(1)}$ in $Att_1$ to $a_4^{(1)}$ and changes $a_4^{(1)}$ into $a_2^{(1)}$; for $Att_2$ it changes $a_3^{(2)}$ into $a_5^{(2)}$ and changes $a_5^{(2)}$ into $a_3^{(2)}$. Thus $\phi(r_\ell) = r_t$ and $\phi(r_t) = r_\ell$ and $\phi$ is its own inverse. Furthermore, for any record $r$, $\zeta(r_\ell, r) = \zeta(\phi(r_\ell), \phi(r)) = \zeta(r_t, \phi(r))$ since renaming attribute values the same way in two records does not affect the set of attributes on which they match (and the last equality is because $\phi(r_\ell) = r_t$).

We next note that since $r_t$ and $r_\ell$ are both assigned 1 by $\mathbf{v}$, then they must differ on an even number of discriminative attributes of $\mathbf{v}$ (if they differ on a discriminative attribute, one must have a value that is positively selected and the other must have a value that is negatively selected – there cannot be a 0 because $r_\ell$ and $r_t$ are not assigned a 0 by $\mathbf{v}$). Therefore, due to its definition, $\phi$ modifies an even number of discriminative attributes and therefore for any record $r$, both $r$ and $\phi(r)$ get assigned the same value by $\mathbf{v}$.

Putting these facts together, we get:

$$\sum_{\substack{r \text{ assigned} \\ \text{value 1 by } \mathbf{v}}} c_{\zeta(r_\ell, r)} - \sum_{\substack{r \text{ assigned} \\ \text{value -1 by } \mathbf{v}}} c_{\zeta(r_\ell, r)}$$

$$= \sum_{\substack{\phi(r) \text{ assigned} \\ \text{value 1 by } \mathbf{v}}} c_{\zeta(r_\ell, r)} - \sum_{\substack{\phi(r) \text{ assigned} \\ \text{value -1 by } \mathbf{v}}} c_{\zeta(r_\ell, r)} \quad \text{since } \phi \text{ doesn't change the summation set}$$

$$= \sum_{\substack{\phi(r) \text{ assigned} \\ \text{value 1 by } \mathbf{v}}} c_{\zeta(\phi(r_\ell), \phi(r))} - \sum_{\substack{\phi(r) \text{ assigned} \\ \text{value -1 by } \mathbf{v}}} c_{\zeta(\phi(r_\ell), \phi(r))} \quad \text{since } \phi \text{ preserves the outcome of } \zeta$$

$$= \sum_{\substack{\phi(r) \text{ assigned} \\ \text{value 1 by } \mathbf{v}}} c_{\zeta(r_t, \phi(r))} - \sum_{\substack{\phi(r) \text{ assigned} \\ \text{value -1 by } \mathbf{v}}} c_{\zeta(r_t, \phi(r))} \quad \text{since } \phi(r_\ell) = r_t$$

$$= \sum_{\substack{r' \text{ assigned} \\ \text{value 1 by } \mathbf{v}}} c_{\zeta(r_t, r')} - \sum_{\substack{r' \text{ assigned} \\ \text{value -1 by } \mathbf{v}}} c_{\zeta(r_t, r')} \quad \text{renaming the summation variable from } \phi(r) \text{ to } r'$$

and that proves Equation 6

**Case 3: $\mathbf{v}$ assigns a $-1$ to $r_\ell$.** In this case, every attribute of $r_\ell$ has a value that is (either positively or negatively) selected by $\mathbf{v}$ and an odd number are negatively selected. Our goal is to show that if some other record $r_t$ is assigned a 1 by $\mathbf{v}$, then the dot product between $\mathbf{v}$ and $\ell^{\text{th}}$ row of $\boldsymbol{\Gamma}$ is the negative of the dot product between $\mathbf{v}$ and the $t^{\text{th}}$ row of $\boldsymbol{\Gamma}$. That is, we want to show:

$$\sum_{\substack{r \text{ assigned} \\ \text{value 1 by } \mathbf{v}}} c_{\zeta(r_\ell, r)} - \sum_{\substack{r \text{ assigned} \\ \text{value -1 by } \mathbf{v}}} c_{\zeta(r_\ell, r)} = - \sum_{\substack{r \text{ assigned} \\ \text{value 1 by } \mathbf{v}}} c_{\zeta(r_t, r)} + \sum_{\substack{r \text{ assigned} \\ \text{value -1 by } \mathbf{v}}} c_{\zeta(r_t, r)} \tag{7}$$

As in the previous case, we define $\phi$ in the same way and reasoning as before we see that for any record $r$, $\zeta(r_\ell, r) = \zeta(\phi(r_\ell), \phi(r)) = \zeta(r_t, \phi(r))$ and since now $\phi$ must change an odd number of

discriminative attributes (since $r_\ell$ and $r_t$ are assigned -1 and 1 by $\mathbf{v}$) then for any record $r$, the value assigned to $r$ by $\mathbf{v}$ is the negative of the value assigned to $\phi(r)$ by $\mathbf{v}$. Thus we have:

$$
\sum_{\substack{r \text{ assigned} \\ \text{value 1 by } \mathbf{v}}} c_{\zeta(r_\ell, r)} - \sum_{\substack{r \text{ assigned} \\ \text{value -1 by } \mathbf{v}}} c_{\zeta(r_\ell, r)}
$$

$$
= \sum_{\substack{\phi(r) \text{ assigned} \\ \text{value -1 by } \mathbf{v}}} c_{\zeta(r_\ell, r)} - \sum_{\substack{\phi(r) \text{ assigned} \\ \text{value +1 by } \mathbf{v}}} c_{\zeta(r_\ell, r)} \quad \text{since } \phi \text{ flips the summation sets}
$$

$$
= \sum_{\substack{\phi(r) \text{ assigned} \\ \text{value -1 by } \mathbf{v}}} c_{\zeta(\phi(r_\ell), \phi(r))} - \sum_{\substack{\phi(r) \text{ assigned} \\ \text{value +1 by } \mathbf{v}}} c_{\zeta(\phi(r_\ell), \phi(r))} \quad \text{since } \phi \text{ preserves the outcome of } \zeta
$$

$$
= \sum_{\substack{\phi(r) \text{ assigned} \\ \text{value -1 by } \mathbf{v}}} c_{\zeta(r_t, \phi(r))} - \sum_{\substack{\phi(r) \text{ assigned} \\ \text{value +1 by } \mathbf{v}}} c_{\zeta(r_t, \phi(r))} \quad \text{since } \phi(r_\ell) = r_t
$$

$$
= \sum_{\substack{r' \text{ assigned} \\ \text{value -1 by } \mathbf{v}}} c_{\zeta(r_t, r')} - \sum_{\substack{r \text{ assigned} \\ \text{value +1 by } \mathbf{v}}} c_{\zeta(r_t, r')} \quad \text{renaming the summation variable from } \phi(r') \text{ to } r'
$$

and that proves Equation 7.

Thus what these 3 cases show us are that there exists some constant $\beta$ such that:

- If the i$^{\text{th}}$ position of the expansion of $\mathbf{v}$ is 0 (i.e., $r_i$ is assigned 0 by $\mathbf{v}$), then the i$^{\text{th}}$ position of $\mathbf{\Gamma v}$ is also 0 (the dot product between the i$^{\text{th}}$ row and $\mathbf{v}$ is 0).

- If the i$^{\text{th}}$ position of the expansion of $\mathbf{v}$ is 1 (i.e., $r_i$ is assigned 1 by $\mathbf{v}$), then the i$^{\text{th}}$ position of $\mathbf{\Gamma v}$ is $\beta$ (the dot product between the i$^{\text{th}}$ row and $\mathbf{v}$ is $\beta$).

- If the i$^{\text{th}}$ position of the expansion of $\mathbf{v}$ is -1 (i.e., $r_i$ is assigned -1 by $\mathbf{v}$), then the i$^{\text{th}}$ position of $\mathbf{\Gamma v}$ is $-\beta$ (the dot product between the i$^{\text{th}}$ row and $\mathbf{v}$ is $-\beta$).

Thus $\mathbf{v}$ is an eigenvector of $\mathbf{\Gamma}$ with eigenvalue $\beta$. That proves the first part of the theorem.

The next part of the theorem is to show that if two residual basis vectors have the same discriminative attributes, then they have the same eigenvalue. So let $\mathbf{v} = \mathbf{v}^{(1)} \otimes \cdots \otimes \mathbf{v}^{(n_a)}$ and $\mathbf{w} = \mathbf{w}^{(1)} \otimes \cdots \otimes \mathbf{w}^{(n_a)}$ be two residual basis vectors that have the same discriminative attributes. Define a renaming permutation $\pi$ as follows:

- For an attribute $Att_\ell$ that is not discriminative for $\mathbf{v}$ (and hence also not for $\mathbf{w}$), $\pi$ does not rename its values (i.e., it acts as the identity for those attribute values).

- For a discriminative attribute $Att_\ell$, let $e_{1,i_\ell}$ be the kron component for $\mathbf{v}$ (i.e., $\mathbf{v}^{(\ell)} = e_{1,i_\ell}$) and let $e_{1,j_\ell}$ be the kron component for $\mathbf{w}$. Note the indices $i_\ell$ and $j_\ell$ are not equal to 1. In this case, we make $\pi$ do the following renamings:

  - $a_{i_\ell} \rightarrow a_{j_\ell}$
  - $a_{j_\ell} \rightarrow a_{i_\ell}$
  - The remaining attribute values are unchanged.

By considering which records are assigned 1,-1 and 0 by $\mathbf{v}$ and $\mathbf{w}$, it is clear that $\pi$ converts $\mathbf{v}$ into $\mathbf{w}$ (and vice versa). Let $\mathbf{W}$ be the matrix representation of the renaming permutation $\pi$, so that $\mathbf{Wv} = \mathbf{w}$ and $\mathbf{W}^T \mathbf{w} = \mathbf{v}$ (a permutation matrix is orthogonal, so its inverse is its transpose). Thus, letting $\beta$ denote the eigenvalue of $\mathbf{v}$ with respect to $\mathbf{\Gamma}$, we have:

$$
\beta \mathbf{v} = \mathbf{\Gamma v}
$$
$$
= \mathbf{\Gamma W}^T \mathbf{w}
$$

$$= \mathbf{W}^T \mathbf{\Gamma} \mathbf{W} \mathbf{W}^T \mathbf{w} \quad \text{due to the symmetry from Lemma D.3}$$
$$= \mathbf{W}^T \mathbf{\Gamma} \mathbf{w},$$

since $\mathbf{W}^T$ is the inverse of $\mathbf{W}$ and so

$$\beta \mathbf{w} = \beta \mathbf{W} \mathbf{v} = \mathbf{W} \mathbf{W}^T \mathbf{\Gamma} \mathbf{w} = \mathbf{\Gamma} \mathbf{w}$$

and thus $\mathbf{w}$ has the same eigenvalue as $\mathbf{v}$. □

Thus each residual basis matrix $\mathbf{R_A}$ has a useful property: its rows are linearly independent and are part of the same eigenspace (linear space of vectors with the same eigenvalue) of the privacy cost matrix $\mathbf{\Gamma}$ of an optimal mechanism. This allows us to prove the main result:

THEOREM 4.4. *Given a marginal workload $Wkload$ and a regular loss function $\mathcal{L}$, suppose the optimization problem (either Equation 1 or 2) is feasible. Then there exist nonnegative constants $\sigma_{\mathbf{A}}^2$ for each $\mathbf{A} \in closure(Wkload)$ (the constants do not depend on the data), such that the optimal linear Gaussian mechanism $\mathcal{M}_{opt}$ for loss function $\mathcal{L}$ releases $\mathcal{M}_{\mathbf{A}}(\mathbf{x}; \sigma_{\mathbf{A}}^2)$ for all $\mathbf{A} \in closure(Wkload)$. Furthermore, any matrix mechanism for this workload must produce at least this many noise measurements during its selection phase.*

*Proof of Theorem 4.4.* Let $ALL$ represent $closure(\{Att_1, \ldots, Att_{n_a}\})$ – all possible subsets of attributes. Theorem D.7 guarantees that there is an optimal mechanism whose privacy cost matrix $\mathbf{\Gamma}$ has eigenvectors equal to the rows of the residual matrices. Rows within the same residual matrix have the same eigenvalues. Since privacy cost matrices are symmetric positive semidefinite, this means that for every $\mathbf{A} \in ALL$, there exists a nonnegative number $\beta_{\mathbf{A}}$ such that:

$$\mathbf{\Gamma} \mathbf{R}_{\mathbf{A}}^T = \beta_{\mathbf{A}} \mathbf{R}_{\mathbf{A}}^T$$

By Theorem 3.5 of [47], if two Gaussian linear mechanisms have the same privacy cost matrix then each can be obtained by linearly processing the other. Thus they have the same privacy properties (under any postprocessing invariant privacy definition) and can be used to answer the same queries with the same exact accuracies (under any measure of accuracy). Thus we just need to construct the appropriate mechanism having privacy cost matrix $\mathbf{\Gamma}$.

For each $\mathbf{A}$, let $\mathbf{Z}_{\mathbf{A}}$ be a matrix with orthonormal rows that span the row space of $\mathbf{R}_{\mathbf{A}}$. Thus the rows of $\mathbf{Z}_{\mathbf{A}}$ are also eigenvectors of $\mathbf{\Gamma}$ (having common eigenvalue $\beta_{\mathbf{A}}$) and the rows of $\mathbf{Z}_{\mathbf{A}}$ are orthogonal to the rows of $\mathbf{Z}_{\mathbf{A}'}$ for $\mathbf{A} \neq \mathbf{A}'$ (a consequence of Theorem 4.2). Thus the set of rows of the $\mathbf{Z}_{\mathbf{A}}$ for all $\mathbf{A} \in ALL$ are a complete list of the eigenvecotrs of $\mathbf{\Gamma}$ (the are linearly independent and span $\mathbb{R}^d$). Thus the (symmetric positive semidefinite) privacy cost matrix $\mathbf{\Gamma}$ can be expressed as:

$$\mathbf{\Gamma} = \sum_{\mathbf{A} \in ALL} \beta_{\mathbf{A}} \mathbf{Z}_{\mathbf{A}}^T \mathbf{Z}_{\mathbf{A}}$$

and one mechanism that achieves this privacy cost matrix is the one that releases $\mathbf{Z}_{\mathbf{A}} \mathbf{x} + N(\mathbf{0}, \frac{1}{\beta_{\mathbf{A}}} \mathcal{I})$ for each $\mathbf{A} \in ALL$ for which $\beta_{\mathbf{A}} \neq 0$ (i.e., we can drop the eigenvectors with eigenvalue equal to 0 as they make no difference to the privacy cost matrix).

Now, since the rows of $\mathbf{R}_{\mathbf{A}}$ and $\mathbf{Z}_{\mathbf{A}}$ are independent linear bases of the same subspace, then there exists an invertible matrix $\mathbf{Y}_{\mathbf{A}}$ such that $\mathbf{R}_{\mathbf{A}} = \mathbf{Y}_{\mathbf{A}} \mathbf{Z}_{\mathbf{A}}$. Furthermore, $\mathbf{R}_{\mathbf{A}} \mathbf{R}_{\mathbf{A}}^T$ is invertible and $\mathbf{Z}_{\mathbf{A}} \mathbf{Z}_{\mathbf{A}}^T = \mathcal{I}$ by orthonormality of its rows. Therefore

$$\mathbf{R}_{\mathbf{A}}^T (\mathbf{R}_{\mathbf{A}} \mathbf{R}_{\mathbf{A}}^T)^{-1} \mathbf{R}_{\mathbf{A}} = \mathbf{Z}_{\mathbf{A}}^T \mathbf{Y}_{\mathbf{A}}^T (\mathbf{Y}_{\mathbf{A}} \mathbf{Z}_{\mathbf{A}} \mathbf{Z}_{\mathbf{A}}^T \mathbf{Y}^T)^{-1} \mathbf{Y}_{\mathbf{A}} \mathbf{Z}_{\mathbf{A}}$$
$$= \mathbf{Z}_{\mathbf{A}}^T \mathbf{Y}_{\mathbf{A}}^T \mathbf{Y}_{\mathbf{A}}^{-T} (\mathbf{Z}_{\mathbf{A}} \mathbf{Z}_{\mathbf{A}}^T)^{-1} \mathbf{Y}_{\mathbf{A}}^{-1} \mathbf{Y}_{\mathbf{A}} \mathbf{Z}_{\mathbf{A}}$$
$$= \mathbf{Z}_{\mathbf{A}}^T (\mathbf{Z}_{\mathbf{A}} \mathbf{Z}_{\mathbf{A}}^T)^{-1} \mathbf{Z}_{\mathbf{A}}$$
$$= \mathbf{Z}_{\mathbf{A}}^T \mathbf{Z}_{\mathbf{A}} \quad \text{by orthonormality of the rows of } \mathbf{Z}_{\mathbf{A}}$$

Thus we have

$$\mathbf{\Gamma} = \sum_{\mathbf{A} \in ALL} \beta_{\mathbf{A}} \mathbf{R}_{\mathbf{A}}^T (\mathbf{R}_{\mathbf{A}} \mathbf{R}_{\mathbf{A}}^T)^{-1} \mathbf{R}_{\mathbf{A}}$$

and a mechanism that achieves this privacy cost matrix is the one that releases $\mathbf{R_A x} + N(\mathbf{0}, \frac{1}{\beta_\mathbf{A}} \mathbf{R_A R_A^T})$ for each $\mathbf{A}$ for which $\beta_\mathbf{A} \neq 0$.

We next note that each covariance matrices we propose to use, $\mathbf{\Sigma_A}$, is proportional to $\mathbf{R_A R_A^T}$ (they are equal up to positive rescaling). If we define the positive constants $\kappa_\mathbf{A}$ such that $\mathbf{R_A R_A^T} = \kappa_\mathbf{A} \mathbf{\Sigma_A}$ then we note that the $\sigma_\mathbf{A}^2$ in the theorem statement are equal to $\kappa_\mathbf{A}/\beta_\mathbf{A}$.

Next, we show that the eigenvalues $\beta_\mathbf{A} > 0$ for $\mathbf{A} \in \text{closure}(Wkload)$ and 0 otherwise, so that the optimal mechanism would not make use of any submechanism $\mathcal{M}_\mathbf{A}$ for $\mathbf{A} \notin \text{closure}(Wkload)$.

First, by Theorem 4.2, the rows of $\mathbf{R_A}$, for $\mathbf{A} \in \text{closure}(Wkload)$ form an independent linear basis for the space spanned by the rows of the marginals $\mathbf{Q_A}$ for $\mathbf{A} \in Wkload$. If a noisy $\mathbf{R_A x}$ is not released for some $\mathbf{A} \in \text{closure}(Wkload)$, then an unbiased noisy answer to at least one of the workload marginals could not be computed. Hence, they must all be part of the optimal mechanism (and thus, because of linear independence, *any* mechanism needs to get at least as many scalar noisy answers as this). This shows that $\beta_\mathbf{A} > 0$ for all $\mathbf{A} \in \text{closure}(Wkload)$. On the other hand since the rows of $\mathbf{R_A}$ are orthogonal to the rows of $\mathbf{R_{A'}}$ for $\mathbf{A} \neq \mathbf{A'}$, getting answers to $\mathbf{R_{A'} x}$, for $\mathbf{A'} \notin \text{closure}(Wkload)$, cannot help estimate the answers to the marginals $\mathbf{Q_A}$ for $\mathbf{A} \in Wkload$ (by Theorem 4.2, $\mathbf{R_{A'}}$ are orthogonal to the matrices representing these marginals when $\mathbf{A'} \notin \text{closure}(Wkload)$). Hence an optimal privacy mechanism cannot waste privacy budget on these irrelevant queries. This shows that $\beta_{\mathbf{A'}} = 0$ for $\mathbf{A'} \notin \text{closure}(Wkload)$ and concludes the proof. $\qquad\square$

# E  The other proofs about base mechanisms

THEOREM 4.2. *Let $\mathbf{A}$ be a set of attributes and let $\mathbf{Q_A}$ be the matrix representation of the marginal on $\mathbf{A}$. Then the rows of the matrices $\mathbf{R_{A'}}$, for all $\mathbf{A'} \subseteq \mathbf{A}$, form a linearly independent basis of the row space of $\mathbf{Q_A}$. Furthermore, if $\mathbf{A'} \neq \mathbf{A''}$ then $\mathbf{R_{A'} R_{A''}^T} = \mathbf{0}$ (they are mutually orthogonal).*

*Proof of Theorem 4.2.* Consider two sets $\mathbf{A'} \neq \mathbf{A''}$ and represent there respective residual matrices as:

$$\mathbf{R_{A'}} = \mathbf{V}_1' \otimes \cdots \otimes \mathbf{V}_{n_a}'$$
$$\mathbf{R_{A''}} = \mathbf{V}_1'' \otimes \cdots \otimes \mathbf{V}_{n_a}''$$
$$\mathbf{R_{A'} R_{A''}^T} = (\mathbf{V}_1'(\mathbf{V}_1'')^T) \otimes \cdots \otimes (\mathbf{V}_{n_a}'(\mathbf{V}_{n_a}'')^T)$$

Since $\mathbf{A'} \neq \mathbf{A''}$ then one of them contains an attribute, say $Att_i$, that the other doesn't have. Therefore either $\mathbf{V}_i'$ or $\mathbf{V}_i''$ is the vector $\mathbf{1}_{|Att_i|}^T$ and the other is $\mathbf{Sub}_{|Att_i|}$. However, $\mathbf{1}_{|Att_i|}^T \mathbf{Sub}_{|Att_i|}^T = \mathbf{0}$ and $\mathbf{Sub}_{|Att_i|} \mathbf{1}_{|Att_i|} = \mathbf{0}$ and hence $\mathbf{R_{A'} R_{A''}^T} = \mathbf{0}$.

Next, for any set $\mathbf{A'}$, it is clear that the row space of $\mathbf{R_{A'}}$ is contained in the row space of the marginal matrix $\mathbf{Q_{A'}}$. It is also clear that if $\mathbf{A'} \subseteq \mathbf{A}$ then the row space of the marginal matrix $\mathbf{Q_{A'}}$ is contained in the row space of $\mathbf{Q_A}$ (because $\mathbf{Q_{A'}}$ represents a sub-marginal of $\mathbf{Q_A}$). Thus the rows of the matrices $\mathbf{R_{A'}}$, for all $\mathbf{A'} \subseteq \mathbf{A}$, are contained in the rowspace of $\mathbf{Q_A}$. Thus we just need to show that the combined rows of $\mathbf{R_{A'}}$, for all $\mathbf{A'} \subseteq \mathbf{A}$, are linearly independent and that the number of rows is the same as the number of rows of $\mathbf{Q_A}$.

First, each $\mathbf{R_{A'}}$ is a kronecker product of matrices with full row rank, and so $\mathbf{R_{A'}}$ has full row rank (therefore its rows are linearly independent). Furthermore, since $\mathbf{R_{A'} R_{A''}^T} = \mathbf{0}$ whenever $\mathbf{A'} \neq \mathbf{A''}$ this means that the row space of $\mathbf{R_{A'}}$ is orthogonal to the row space of $\mathbf{R_{A''}}$. Hence the combined rows of the $\mathbf{R_{A'}}$, for all $\mathbf{A'} \subseteq \mathbf{A}$, are linearly independent.

Next, the number of rows in $\mathbf{R_\emptyset}$ is 1 and the number of rows in $\mathbf{R_{A'}}$ is equal to $\prod_{Att_i \in \mathbf{A'}} (|Att_i| - 1)$ for $\mathbf{A'} \neq \emptyset$ and so the total number of rows in the residual matrices is $1 + \sum_{\substack{\mathbf{A'} \subseteq \mathbf{A} \\ \mathbf{A'} \neq \emptyset}} \prod_{Att_i \in \mathbf{A'}} (|Att_i| - 1)$.

By the distributive property of multiplication, this is exactly the same as the product:

$$\prod_{Att_i \in \mathbf{A}} ((|Att_i| - 1) + 1) = \prod_{Att_i \in \mathbf{A}} |Att_i|$$

which is the number of rows in $\mathbf{Q_A}$ and that proves that the combined rows of $\mathbf{R_{A'}}$, for all $\mathbf{A'} \subseteq \mathbf{A}$, form a linearly independent basis for the row span of $\mathbf{Q_A}$. $\qquad\square$

LEMMA E.1. *For any $i$,* $\mathbf{Sub}_{|Att_i|}^T(\mathbf{Sub}_{|Att_i|}\mathbf{Sub}_{|Att_i|}^T)^{-1}\mathbf{Sub}_{|Att_i|} = \mathcal{I}_{|Att_i|} - \frac{1}{|Att_i|}\mathbf{1}_{|Att_i|}\mathbf{1}_{|Att_i|}^T$

*Proof of Lemma E.1.* For the moment, let $\mathbf{Y}$ denote $\mathbf{Sub}_{|Att_i|}^T(\mathbf{Sub}_{|Att_i|}\mathbf{Sub}_{|Att_i|}^T)^{-1}\mathbf{Sub}_{|Att_i|}$. Then we know:

- $\mathbf{Y}$ is symmetric.

- $\mathbf{Y}$ is an $|Att_i| \times |Att_i|$ matrix and its rank is $|Att_i| - 1$ since the rank of $\mathbf{Sub}_{|Att_i|}$ is $|Att_i| - 1$.

- $\mathbf{Sub}_{|Att_i|}\mathbf{Y}\mathbf{Sub}_{|Att_i|}^T = \mathbf{Sub}_{|Att_i|}\mathbf{Sub}_{|Att_i|}^T$.

Now, one symmetric solution to the equation $\mathbf{Sub}_{|Att_i|}\mathbf{X}\mathbf{Sub}_{|Att_i|}^T = \mathbf{Sub}_{|Att_i|}\mathbf{Sub}_{|Att_i|}^T$ is $\mathbf{X} = \mathcal{I}_{|Att_I|}$ and if $\mathbf{X}_1$ is another symmetric solution then $\mathbf{Sub}_{|Att_i|}(\mathcal{I}_{|Att_i|} - \mathbf{X}_1)\mathbf{Sub}_{|Att_i|}^T = \mathbf{0}$.

This means that $\mathbf{Sub}_{|Att_i|}\mathbf{v} = \mathbf{0}$ for each eigenvector $\mathbf{v}$ of the symmetric matrix $\mathcal{I}_{|Att_i|} - \mathbf{X}_1$ that has a nonzero eigenvalue. Since the rank of $\mathbf{Sub}_{|Att_i|}$ is $|Att_i| - 1$, the only vectors $\mathbf{v}$ for which $\mathbf{Sub}_{|Att_i|}\mathbf{v} = \mathbf{0}$ are proportional to $\mathbf{1}_{|Att_i|}$ (the null space has rank 1) and so $\mathcal{I}_{|Att_i|} - \mathbf{X}_1 = -c\mathbf{1}_{|Att_i|}\mathbf{1}_{|Att_i|}^T$ for some constant $c$.

This means that $\mathbf{Y}$ (and any other symmetric solution) has the form $\mathcal{I}_{|Att_i|} + c\mathbf{1}_{|Att_i|}\mathbf{1}_{|Att_i|}^T$. To find $c$, we note that $\mathbf{Y}$ is not full rank.

By the Sherman-Morrison-Woodbury inversion formula, if $\mathcal{I}_{|Att_i|} + c\mathbf{1}_{|Att_i|}\mathbf{1}_{|Att_i|}^T$ is invertible, then its inverse is $\mathcal{I}_{|Att_i|} - c\mathbf{1}_{|Att_i|}\left(1 + c\mathbf{1}_{|Att_i|}^T\mathbf{1}_{|Att_i|}\right)^{-1}\mathbf{1}_{|Att_i|}^T = \mathcal{I}_{|Att_i|} - c\frac{\mathbf{1}_{|Att_i|}\mathbf{1}_{|Att_i|}^T}{1+c|Att_i|}$. Thus, to prevent invertibility, we must have $c = -1/|Att_i|$.

Therefore $\mathbf{Y} = \mathcal{I}_{|Att_i|} - \frac{1}{|Att_i|}\mathbf{1}_{|Att_i|}\mathbf{1}_{|Att_i|}^T$. $\qquad\square$

THEOREM 4.5. *The privacy cost of $\mathcal{M}_\mathbf{A}$ with noise parameter $\sigma_\mathbf{A}^2$ is $\frac{1}{\sigma_\mathbf{A}^2}\prod_{Att_i \in \mathbf{A}}\frac{|Att_i|-1}{|Att_i|}$ and the evaluation of $\mathcal{M}_\mathbf{A}$ given in Algorithm 1 is correct – i.e., the output has the distribution $N(\mathbf{R_A}x, \sigma_\mathbf{A}^2\Sigma_\mathbf{A})$.*

*Proof of Theorem 4.5.* Without loss of generality (and to simplify notation), assume $\mathbf{A} = \{Att_1, \dots, Att_\ell\}$ consists of the first $\ell$ attributes.

By definition, $pcost(\mathcal{M}_\mathbf{A}(\cdot; \sigma_\mathbf{A}^2))$ is the largest diagonal of $\frac{1}{\sigma^2}\mathbf{R_A}^T\Sigma_\mathbf{A}^{-1}\mathbf{R_A}$. Thus we can write:

$$\mathbf{R_A} = \left(\bigotimes_{i=1}^{\ell}\mathbf{Sub}_{|Att_i|}\right) \otimes \left(\bigotimes_{j=\ell+1}^{n_a}\mathbf{1}_{|Att_j|}^T\right)$$

$$\mathbf{R_A}^T = \left(\bigotimes_{i=1}^{\ell}\mathbf{Sub}_{|Att_i|}^T\right) \otimes \left(\bigotimes_{j=\ell+1}^{n_a}\mathbf{1}_{|Att_j|}\right)$$

$$\mathbf{H} = \left(\bigotimes_{i=1}^{\ell}\mathbf{Sub}_{|Att_i|}\right) \otimes \left(\bigotimes_{j=\ell+1}^{n_a}[1]\right) \quad \text{(rightmost krons use } 1 \times 1 \text{ matrices)}$$

$$\Sigma_\mathbf{A} = \mathbf{H}\mathbf{H}^T = \left(\bigotimes_{i=1}^{\ell}(\mathbf{Sub}_{|Att_i|}\mathbf{Sub}_{|Att_i|}^T)\right) \otimes \left(\bigotimes_{j=\ell+1}^{n_a}[1]\right)$$

$$\boldsymbol{\Sigma}_{\mathbf{A}}^{-1} = \left( \bigotimes_{i=1}^{\ell} (\mathbf{Sub}_{|Att_i|}\mathbf{Sub}_{|Att_i|}^T)^{-1} \right) \otimes \left( \bigotimes_{j=\ell+1}^{n_a} [\,1\,] \right)$$

$$\mathbf{R}_{\mathbf{A}}^T \boldsymbol{\Sigma}_{\mathbf{A}}^{-1} \mathbf{R}_{\mathbf{A}} = \left( \bigotimes_{i=1}^{\ell} \mathbf{Sub}_{|Att_i|}^T (\mathbf{Sub}_{|Att_i|}\mathbf{Sub}_{|Att_i|}^T)^{-1}\mathbf{Sub}_{|Att_i|} \right) \otimes \left( \bigotimes_{j=\ell+1}^{n_a} \mathbf{1}_{|Att_j|} [\,1\,] \mathbf{1}_{|Att_j|}^T \right)$$
$$(8)$$

Now, by Lemma E.1,

$$\mathbf{Sub}_{|Att_i|}^T (\mathbf{Sub}_{|Att_i|}\mathbf{Sub}_{|Att_i|}^T)^{-1}\mathbf{Sub}_{|Att_i|} = \mathcal{I}_{|Att_i|} - \frac{1}{|Att_i|} \mathbf{1}_{|Att_i|}\mathbf{1}_{|Att_i|}^T \qquad (9)$$

Since its diagonals are $\frac{|Att_i|-1}{|Att_i|}$, then combined with Equation 8 it proves the result for $pcost(\mathcal{M}_{\mathbf{A}}(\cdot, \sigma_{\mathbf{A}}^2))$.

We next consider the correctness of Algorithm 1. First, we need to show that for the matrix $\mathbf{H}$ defined in Line 3 in Algorithm 1, $\mathbf{H}\mathbf{Q}_{\mathbf{A}}\mathbf{x} = \mathbf{R}_{\mathbf{A}}\mathbf{x}$. Then we can write:

$$\mathbf{R}_{\mathbf{A}} = \left( \bigotimes_{i=1}^{\ell} \mathbf{Sub}_{|Att_i|} \right) \otimes \left( \bigotimes_{j=\ell+1}^{n_a} \mathbf{1}_{|Att_j|}^T \right)$$

$$\mathbf{Q}_{\mathbf{A}} = \left( \bigotimes_{i=1}^{\ell} \mathcal{I}_{|Att_i|} \right) \otimes \left( \bigotimes_{j=\ell+1}^{n_a} \mathbf{1}_{|Att_j|}^T \right) \qquad \text{rightmost product is a matrix with 1 row}$$

$$\mathbf{H} = \left( \bigotimes_{i=1}^{\ell} \mathbf{Sub}_{|Att_i|} \right) \otimes [\,1\,] \qquad \text{(rightmost term is a } 1 \times 1 \text{ matrix)}$$

$$\mathbf{H}\mathbf{Q}_{\mathbf{A}} = \left( \bigotimes_{i=1}^{\ell} \left( \mathbf{Sub}_{|Att_i|}\mathcal{I}_{|Att_i|} \right) \right) \otimes \left( [\,1\,] \left( \bigotimes_{j=\ell+1}^{n_a} \mathbf{1}_{|Att_j|}^T \right) \right)$$

$$= \mathbf{R}_{\mathbf{A}}$$

Next, we note that if $\mathbf{z}$ is distributed as $N(0, \mathbf{I}_m)$ (Line 4 in Algorithm 1) then $\sigma_{\mathbf{A}}\mathbf{H}\mathbf{z}$ has the distribution $N(0, \sigma^2 \mathbf{H}\mathbf{H}^T) = \boldsymbol{\Sigma}_{\mathbf{A}}$ and hence the algorithm is correct. $\qquad \square$

## F  Proofs related to the reconstruction step

LEMMA 4.6. *For any $Att_i$, let $\ell = |Att_i|$. The matrix $\mathbf{Sub}_\ell$ has the following block matrix, with dimensions $\ell \times (\ell - 1)$, as its pseudo-inverse (and right inverse):* $\mathbf{Sub}_\ell^\dagger = \frac{1}{\ell} \begin{bmatrix} \mathbf{1}_{\ell-1}^T \\ \mathbf{1}_{\ell-1}\mathbf{1}_{\ell-1}^T - \ell\mathcal{I}_{\ell-1} \end{bmatrix}.$

*Proof of Lemma 4.6.* First, if a matrix has a right inverse then that is the pseudo-inverse. Hence we just need to show that $\mathbf{Sub}_\ell\mathbf{Sub}_\ell^\dagger = \mathcal{I}_{\ell-1}$.

Note that the $j^{\text{th}}$ row of $\mathbf{Sub}_\ell$ has a 1 in position 1, -1 in position $j + 1$, and is 0 everywhere else.

Meanwhile, the $i^{\text{th}}$ column of our claimed representation of $\mathbf{Sub}_\ell^\dagger$ has a $-(\ell - 1)/\ell$ in position $i + 1$ and $1/\ell$ everywhere else.

Hence if $j \neq i$ then the dot product between row $j$ of $\mathbf{Sub}_\ell$ and column $i$ of $\mathbf{Sub}_\ell^\dagger$ is 0 since the nonzero elements of the row from $\mathbf{Sub}_\ell$ are being multiplied by $1/\ell$ and $1/\ell$.

If $i = j$ then the corresponding first elements that are multiplied are 1 and $1/\ell$ while the elements at position $i + 1$ being multiplied are $-1$ and $-(\ell - 1)/\ell$. Furthermore, $1(1/\ell) + (-1)(-(\ell - 1)/\ell) = 1$. $\qquad \square$

LEMMA F.1. *For any attribute $Att_i$, let $\ell = |Att_i|$. Then $\mathbf{Sub}_\ell^\dagger (\mathbf{Sub}_\ell\mathbf{Sub}_\ell^T)\mathbf{Sub}_\ell^{\dagger T} = \mathcal{I}_\ell - \frac{1}{\ell}\mathbf{1}_\ell\mathbf{1}_\ell^T$*

*Proof of [Lemma F.1](#).* Because $\mathbf{Sub}_\ell$ has linearly independent rows, the pseudo-inverse of it can be expressed as,

$$\mathbf{Sub}_\ell^\dagger = \mathbf{Sub}_\ell^T(\mathbf{Sub}_\ell\mathbf{Sub}_\ell^T)^{-1}$$

From lemma [E.1](#) we get,

$$\mathbf{Sub}_\ell^\dagger\mathbf{Sub}_\ell = \mathbf{Sub}_\ell^T(\mathbf{Sub}_\ell\mathbf{Sub}_\ell^T)^{-1}\mathbf{Sub}_\ell$$
$$= \mathcal{I}_\ell - \frac{1}{\ell}\mathbf{1}_\ell\mathbf{1}_\ell^T$$

Therefore,

$$\begin{aligned}\mathbf{Sub}_\ell^\dagger(\mathbf{Sub}_\ell\mathbf{Sub}_\ell^T)\mathbf{Sub}_\ell^{\dagger T} &= (\mathbf{Sub}_\ell^\dagger\mathbf{Sub}_\ell)(\mathbf{Sub}_\ell^\dagger\mathbf{Sub}_\ell)^T\\
&= (\mathcal{I}_\ell - \frac{1}{\ell}\mathbf{1}_\ell\mathbf{1}_\ell^T)(\mathcal{I}_\ell - \frac{1}{\ell}\mathbf{1}_\ell\mathbf{1}_\ell^T)\\
&= \mathcal{I}_\ell - \frac{1}{\ell}\mathbf{1}_\ell\mathbf{1}_\ell^T - \frac{1}{\ell}\mathbf{1}_\ell\mathbf{1}_\ell^T + \frac{1}{\ell^2}\mathbf{1}_\ell(\ell)\mathbf{1}_\ell^T\\
&= \mathcal{I}_\ell - \frac{1}{\ell}\mathbf{1}_\ell\mathbf{1}_\ell^T\end{aligned}$$

$\square$

THEOREM F.2. *Let $\mathbf{A}$ be a set of attributes and let $\mathbf{Q_A}$ be the matrix representation of the marginal on $\mathbf{A}$. Given the matrices $\mathbf{R_{A'}}$, for all $\mathbf{A'} \in closure(\mathbf{A})$, we have $\mathbf{Q_A} = \sum\limits_{\mathbf{A'}\in closure(\mathbf{A})} \mathbf{Q_A}\mathbf{R}_{\mathbf{A'}}^\dagger\mathbf{R_{A'}}$.*

*Proof of [Theorem F.2](#).*

$$\mathbf{Q_A} = \bigotimes_{i=1}^{n_a}\mathbf{K}_i \quad \text{where, for each } i,\ \mathbf{K}_i = \begin{cases}\mathcal{I}_{|Att_i|} & \text{if } Att_i \in \mathbf{A}\\ \mathbf{1}_{|Att_i|}^T & \text{if } Att_i \notin \mathbf{A}\end{cases}$$

$$\mathbf{R_{A'}} = \bigotimes_{i=1}^{n_a}\mathbf{V}_i \quad \text{where, for each } i,\ \mathbf{V}_i = \begin{cases}\mathbf{Sub}_{|Att_i|} & \text{if } Att_i \in \mathbf{A'}\\ \mathbf{1}_{|Att_i|}^T & \text{if } Att_i \notin \mathbf{A'}\end{cases}$$

It is straightforward to verify that the following is a right inverse (and hence pseudo-inverse) of $\mathbf{R_{A'}}$

$$\mathbf{R}_{\mathbf{A'}}^\dagger = \bigotimes_{i=1}^{n_a}\mathbf{V}_i^\dagger \quad \text{where, for each } i,\ \mathbf{V}_i^\dagger = \begin{cases}\mathbf{Sub}_{|Att_i|}^\dagger & \text{if } Att_i \in \mathbf{A'}\\ \frac{1}{|Att_i|}\mathbf{1}_{|Att_i|} & \text{if } Att_i \notin \mathbf{A'}\end{cases}$$

$$\mathbf{Q_A}\mathbf{R}_{\mathbf{A'}}^\dagger\mathbf{R_{A'}} = \bigotimes_{i=1}^{n_a}\mathbf{K}_i\mathbf{V}_i^\dagger\mathbf{V}_i \quad \text{where, for each } i,\ \mathbf{K}_i\mathbf{V}_i^\dagger\mathbf{V}_i = \begin{cases}\mathbf{Sub}_{|Att_i|}^\dagger\mathbf{Sub}_{|Att_i|} & \text{if } Att_i \in \mathbf{A'}\\ \frac{1}{|Att_i|}\mathbf{1}_{|Att_i|}\mathbf{1}_{|Att_i|}^T & \text{if } Att_i \in \mathbf{A}/\mathbf{A'}\\ \mathbf{1}_{|Att_i|}^T & \text{if } Att_i \notin \mathbf{A}\end{cases}$$

Because $\mathbf{Sub}_{|Att_i|}$ has linearly independent rows, the pseudo-inverse of it can be expressed as,

$$\mathbf{Sub}_{|Att_i|}^\dagger = \mathbf{Sub}_{|Att_i|}^T(\mathbf{Sub}_{|Att_i|}\mathbf{Sub}_{|Att_i|}^T)^{-1}$$

From lemma [E.1](#) we get,

$$\mathbf{Sub}_{|Att_i|}^\dagger\mathbf{Sub}_{|Att_i|} = \mathbf{Sub}_{|Att_i|}^T(\mathbf{Sub}_{|Att_i|}\mathbf{Sub}_{|Att_i|}^T)^{-1}\mathbf{Sub}_{|Att_i|}$$
$$= \mathcal{I}_{|Att_i|} - \frac{1}{|Att_i|}\mathbf{1}_{|Att_i|}\mathbf{1}_{|Att_i|}^T$$

Therefore,

$$\mathbf{Q_A}\mathbf{R}_{\mathbf{A'}}^\dagger\mathbf{R_{A'}} = \bigotimes_{i=1}^{n_a}\mathbf{T}_i \quad \text{where, for each } i,\ \mathbf{T}_i = \begin{cases}\mathcal{I}_{|Att_i|} - \frac{1}{|Att_i|}\mathbf{1}_{|Att_i|}\mathbf{1}_{|Att_i|}^T & \text{if } Att_i \in \mathbf{A'}\\ \frac{1}{|Att_i|}\mathbf{1}_{|Att_i|}\mathbf{1}_{|Att_i|}^T & \text{if } Att_i \in \mathbf{A}/\mathbf{A'}\\ \mathbf{1}_{|Att_i|}^T & \text{if } Att_i \notin \mathbf{A}\end{cases}$$

Without loss of generality (and to simplify notation), assume $\mathbf{A} = \{Att_1, \ldots, Att_\ell\}$ consists of the first $\ell$ attributes,

$$\mathbf{Q_A} = \left(\bigotimes_{i=1}^{\ell} \mathcal{I}_{|Att_i|}\right) \otimes \left(\bigotimes_{i=\ell+1}^{n_a} \mathbf{1}_{|Att_i|}^T\right)$$

$$\sum_{\mathbf{A}' \in \text{closure}(\mathbf{A})} \mathbf{Q_A R}_{\mathbf{A}'}^\dagger \mathbf{R}_{\mathbf{A}'} = \sum_{\mathbf{A}' \in \text{closure}(\mathbf{A})} \left(\bigotimes_{i=1}^{n_a} \mathbf{T}_i\right)$$

$$= \sum_{\mathbf{A}' \in \text{closure}(\mathbf{A})} \left(\left(\bigotimes_{i=1}^{\ell} \mathbf{T}_i\right) \otimes \left(\bigotimes_{i=\ell+1}^{n_a} \mathbf{1}_{|Att_i|}^T\right)\right)$$

$$= \left(\sum_{\mathbf{A}' \in \text{closure}(\mathbf{A})} \left(\bigotimes_{i=1}^{\ell} \mathbf{T}_i\right)\right) \otimes \left(\bigotimes_{i=\ell+1}^{n_a} \mathbf{1}_{|Att_i|}^T\right)$$

where, for each $i \le \ell$, $\mathbf{T}_i = \begin{cases} \mathcal{I}_{|Att_i|} - \frac{1}{|Att_i|}\mathbf{1}_{|Att_i|}\mathbf{1}_{|Att_i|}^T & \text{if } Att_i \in \mathbf{A}' \\ \frac{1}{|Att_i|}\mathbf{1}_{|Att_i|}\mathbf{1}_{|Att_i|}^T & \text{if } Att_i \in \mathbf{A}/\mathbf{A}' \end{cases}$

Because of the distributive property of the Kronecker product,

$$\bigotimes_{i=1}^{\ell} \mathcal{I}_{|Att_i|} = \bigotimes_{i=1}^{\ell} \left(\left(\mathcal{I}_{|Att_i|} - \frac{1}{|Att_i|}\mathbf{1}_{|Att_i|}\mathbf{1}_{|Att_i|}^T\right) + \frac{1}{|Att_i|}\mathbf{1}_{|Att_i|}\mathbf{1}_{|Att_i|}^T\right)$$

$$= \sum_{\mathbf{A}' \in \text{closure}(\mathbf{A})} \left(\bigotimes_{i=1}^{\ell} \mathbf{T}_i\right)$$

Therefore, combining everything together,

$$\sum_{\mathbf{A}' \in \text{closure}(\mathbf{A})} \mathbf{Q_A R}_{\mathbf{A}'}^\dagger \mathbf{R}_{\mathbf{A}'} = \left(\sum_{\mathbf{A}' \in \text{closure}(\mathbf{A})} \left(\bigotimes_{i=1}^{\ell} \mathbf{T}_i\right)\right) \otimes \left(\bigotimes_{i=\ell+1}^{n_a} \mathbf{1}_{|Att_i|}^T\right)$$

$$= \left(\bigotimes_{i=1}^{\ell} \mathcal{I}_{|Att_i|}\right) \otimes \left(\bigotimes_{i=\ell+1}^{n_a} \mathbf{1}_{|Att_i|}^T\right)$$

$$= \mathbf{Q_A}$$

$\square$

THEOREM 4.7. *Given a marginal workload $Wkload$ and positive numbers $\sigma_\mathbf{A}^2$ for each $\mathbf{A} \in closure(Wkload)$, let $\mathcal{M}$ be the mechanism that outputs $\{\mathcal{M}_\mathbf{A}(\mathbf{x}; \sigma_\mathbf{A}^2) : \mathbf{A} \in closure(Wkload)\}$ and let $\{\omega_\mathbf{A} : \mathbf{A} \in closure(Wkload)\}$ denote the privacy-preserving noisy answers (e.g., $\omega_\mathbf{A} = \mathcal{M}_\mathbf{A}(\mathbf{x}, \sigma^2)$). Then for any marginal on an attribute set $\mathbf{A} \in closure(Wkload)$, Algorithm 2 returns the unique linear unbiased estimate of $\mathbf{Q_A x}$ (i.e., answers to the marginal query) that can be computed from the noisy differentially private answers.*

*The variances $Var(\mathbf{A}; \mathcal{M})$ of all the noisy cell counts of the marginal on $\mathbf{A}$ is the vector whose components are all equal to $\sum_{\mathbf{A}' \subseteq \mathbf{A}} \left(\sigma_{\mathbf{A}'}^2 \prod_{Att_i \in \mathbf{A}'} \frac{|Att_i|-1}{|Att_i|} * \prod_{Att_j \in (\mathbf{A}/\mathbf{A}')} \frac{1}{|Att_j|^2}\right)$. The covariance between any two noisy answers of the marginal on $\mathbf{A}$ is $\sum_{\mathbf{A}' \subseteq \mathbf{A}} \left(\sigma_{\mathbf{A}'}^2 \prod_{Att_i \in \mathbf{A}'} \frac{-1}{|Att_i|} * \prod_{Att_j \in (\mathbf{A}/\mathbf{A}')} \frac{1}{|Att_j|^2}\right)$.*

*Proof of Theorem 4.7.* We first verify the correctness and uniqueness of the reconstruction in Algorithm 2. Uniqueness follows from the fact that the rows from all the matrices $\mathbf{R_A}$ (for $\mathbf{A} \in \text{closure}(Wkload)$) are linearly independent.

Consider Line 3 from Algorithm 2. It uses a $\mathbf{U}$ matrix that depends on both the attributes $\mathbf{A}$ of the marginal one wants to compute and a subset $\mathbf{A}'$ of it. So, for notational dependence, we write it as $\mathbf{U}_{\mathbf{A}\leftarrow\mathbf{A}'}$. It is straightforward to verify that $\mathbf{U}_{\mathbf{A}\leftarrow\mathbf{A}'} = \mathbf{Q}_{\mathbf{A}}\mathbf{R}_{\mathbf{A}'}^{\dagger}$. From Theorem F.2, $\mathbf{Q}_{\mathbf{A}}\mathbf{x} = \sum_{\mathbf{A}'\subseteq\mathbf{A}} \mathbf{Q}_{\mathbf{A}}\mathbf{R}_{\mathbf{A}'}^{\dagger}\mathbf{R}_{\mathbf{A}'}\mathbf{x} = \sum_{\mathbf{A}'\subseteq\mathbf{A}} \mathbf{U}_{\mathbf{A}\leftarrow\mathbf{A}'}\mathbf{R}_{\mathbf{A}'}\mathbf{x}$, and so Algorithm 2 is correct because each $\omega_{\mathbf{A}'}$ is an unbiased noisy version of $\mathbf{R}_{\mathbf{A}'}\mathbf{x}$.

Having established that the $\mathbf{q}$ returned by Line 5 in Algorithm 2 is an unbiased estimate of the marginal query answer $\mathbf{Q}_{\mathbf{A}}\mathbf{x}$, the next step is to compute the covariance matrix $E[\mathbf{q}\mathbf{q}^T]$.

$$
\begin{aligned}
E[\mathbf{q}\mathbf{q}^T] &= E\left[\sum_{\mathbf{A}'\subseteq\mathbf{A}} \mathbf{U}_{\mathbf{A}\leftarrow\mathbf{A}'}\left(\omega_{\mathbf{A}'}\omega_{\mathbf{A}'}^T\right)\mathbf{U}_{\mathbf{A}\leftarrow\mathbf{A}'}^T\right] \\
&= \sum_{\mathbf{A}'\subseteq\mathbf{A}} \mathbf{U}_{\mathbf{A}\leftarrow\mathbf{A}'}\left(\sigma_{\mathbf{A}'}^2\boldsymbol{\Sigma}_{\mathbf{A}'}\right)\mathbf{U}_{\mathbf{A}\leftarrow\mathbf{A}'}^T
\end{aligned}
$$

Without loss of generality (and to simplify notation), assume $\mathbf{A} = \{Att_1, \ldots, Att_\ell\}$ consists of the first $\ell$ attributes, $\mathbf{A}' = \{Att_1, \ldots, Att_t\}$ consists of the first $t \leq \ell$ attributes, then $\mathbf{A}/\mathbf{A}' = \{Att_{t+1}, \ldots, Att_\ell\}$.

By definition, $Var(A; \mathcal{M})$ is the diagonal of $E[\mathbf{q}\mathbf{q}^T] = \sum_{\mathbf{A}'\in\text{closure}(\mathbf{A})} \sigma_{\mathbf{A}'}^2\mathbf{U}_{\mathbf{A}\leftarrow\mathbf{A}'}\boldsymbol{\Sigma}_{\mathbf{A}'}\mathbf{U}_{\mathbf{A}\leftarrow\mathbf{A}'}^T$. Thus we can write:

$$
\mathbf{Q}_{\mathbf{A}} = \left(\bigotimes_{i=1}^{t}\mathcal{I}_{|Att_i|}\right) \otimes \left(\bigotimes_{j=t+1}^{\ell}\mathcal{I}_{|Att_j|}\right) \otimes \left(\bigotimes_{k=\ell+1}^{n_a}\mathbf{1}_{|Att_k|}^T\right)
$$

$$
\mathbf{R}_{\mathbf{A}'} = \left(\bigotimes_{i=1}^{t}\mathbf{Sub}_{|Att_i|}\right) \otimes \left(\bigotimes_{j=t+1}^{\ell}\mathbf{1}_{|Att_j|}^T\right) \otimes \left(\bigotimes_{k=\ell+1}^{n_a}\mathbf{1}_{|Att_k|}^T\right)
$$

$$
\mathbf{R}_{\mathbf{A}'}^{\dagger} = \left(\bigotimes_{i=1}^{t}\mathbf{Sub}_{|Att_i|}^{\dagger}\right) \otimes \left(\bigotimes_{j=t+1}^{\ell}\frac{1}{|Att_j|}\mathbf{1}_{|Att_j|}\right) \otimes \left(\bigotimes_{k=\ell+1}^{n_a}\frac{1}{Att_k}\mathbf{1}_{|Att_k|}\right)
$$

$$
\mathbf{U}_{\mathbf{A}\leftarrow\mathbf{A}'} = \mathbf{Q}_{\mathbf{A}}\mathbf{R}_{\mathbf{A}'}^{\dagger} = \left(\bigotimes_{i=1}^{t}\mathbf{Sub}_{|Att_i|}^{\dagger}\right) \otimes \left(\bigotimes_{j=t+1}^{\ell}\frac{1}{|Att_j|}\mathbf{1}_{|Att_j|}\right) \otimes \left(\bigotimes_{k=\ell+1}^{n_a}[1]\right)
$$

$$
\mathbf{U}_{\mathbf{A}\leftarrow\mathbf{A}'}^T = \left(\bigotimes_{i=1}^{t}\mathbf{Sub}_{|Att_i|}^{\dagger T}\right) \otimes \left(\bigotimes_{j=t+1}^{\ell}\frac{1}{|Att_j|}\mathbf{1}_{|Att_j|}^T\right) \otimes \left(\bigotimes_{k=\ell+1}^{n_a}[1]\right)
$$

$$
\boldsymbol{\Sigma}_{\mathbf{A}'} = \left(\bigotimes_{i=1}^{t}\mathbf{Sub}_{|Att_i|}\mathbf{Sub}_{|Att_i|}^T\right) \otimes \left(\bigotimes_{j=t+1}^{\ell}[1]\right) \otimes \left(\bigotimes_{k=\ell+1}^{n_a}[1]\right)
$$

$$
\begin{aligned}
\mathbf{U}_{\mathbf{A}\leftarrow\mathbf{A}'}\boldsymbol{\Sigma}_{\mathbf{A}'}\mathbf{U}_{\mathbf{A}\leftarrow\mathbf{A}'}^T &= \left(\bigotimes_{i=1}^{t}\mathbf{Sub}_{|Att_i|}^{\dagger}\mathbf{Sub}_{|Att_i|}\mathbf{Sub}_{|Att_i|}^T\mathbf{Sub}_{|Att_i|}^{\dagger T}\right) \\
&\otimes \left(\bigotimes_{j=t+1}^{\ell}\frac{1}{|Att_j|^2}\mathbf{1}_{|Att_j|}[1]\mathbf{1}_{|Att_j|}^T\right) \otimes \left(\bigotimes_{k=\ell+1}^{n_a}[1]\right)
\end{aligned} \tag{10}
$$

Now, by Lemma F.1,

$$
\mathbf{Sub}_{|Att_i|}^{\dagger}\mathbf{Sub}_{|Att_i|}\mathbf{Sub}_{|Att_i|}^T\mathbf{Sub}_{|Att_i|}^{\dagger T} = \mathcal{I}_{|Att_i|} - \frac{1}{|Att_i|}\mathbf{1}_{|Att_i|}\mathbf{1}_{|Att_i|}^T \tag{11}
$$

So the diagonals of $\mathbf{U}_{\mathbf{A}\leftarrow\mathbf{A}'}\boldsymbol{\Sigma}_{\mathbf{A}'}\mathbf{U}_{\mathbf{A}\leftarrow\mathbf{A}'}^T$ can be computed by multiplying $\frac{|Att_i|-1}{|Att_i|}$ for each $Att_i \in \mathbf{A}'$ and $1/|Att_j|$ for each $Att_j \in \mathbf{A} \setminus \mathbf{A}'$. Meanwhile, the off diagonals are all the same and can be computed by multiplying $\frac{-1}{|Att_i|}$ for each $Att_i \in \mathbf{A}'$ and $\frac{1}{|Att_j|^2}$ for each $Att_j \in \mathbf{A} \setminus \mathbf{A}'$.

Computing the variance and covariance of the marginal query answer is therefore the summation of these quantities for all $\mathbf{A}' \subseteq \mathbf{A}$ and is what the theorem states.

$\square$

# G   Computational Complexity Proofs

THEOREM 4.8. *Let $n_a$ be the total number of attributes. Let $\#cells(\mathbf{A})$ denote the number of cells in the marginal on attribute set $\mathbf{A}$. Then:*

1. *Expressing the privacy cost of the optimal mechanism $\mathcal{M}^*$ as a linear combination of the $1/\sigma_{\mathbf{A}}^2$ values takes $O(\sum_{\mathbf{A} \in Wkload} \#cells(\mathbf{A}))$ total time.*
2. *Expressing all of the $Var(\mathbf{A}; \mathcal{M}^*)$, for $\mathbf{A} \in Wkload$, as a linear combinations of the $\sigma_{\mathbf{A}}^2$ values can be done in $O(\sum_{\mathbf{A} \in Wkload} \#cells(\mathbf{A}))$ total time.*
3. *Computing all the noisy outputs of the optimal mechanism (i.e., $\mathcal{M}_{\mathbf{A}}(\mathbf{x}; \sigma_{\mathbf{A}}^2)$ for $\mathbf{A} \in closure(Wkload))$ takes $O\left(n_a \sum_{\mathbf{A} \in Wkload} \prod_{Att_i \in \mathbf{A}}(|Att_i| + 1)\right)$ total time after the true answers have been precomputed (Line 1 in Algorithm 1). Note that the total number of cells on marginals in $Wkload$ is $O\left(\sum_{\mathbf{A} \in Wkload} \prod_{Att_i \in \mathbf{A}} |Att_i|\right)$.*
4. *Reconstructing marginals for all $\mathbf{A} \in Wkload$ takes $O(\sum_{\mathbf{A} \in Wkload} |\mathbf{A}| \#cells(\mathbf{A})^2)$ total time.*
5. *Computing the variance of the cells for all of the marginals for $\mathbf{A} \in Wkload$ can be done in $O(\sum_{\mathbf{A} \in Wkload} \#cells(\mathbf{A}))$ total time.*

*Proof of Theorem 4.8.* First we establish that $|closure(Wkload)| \leq \sum_{\mathbf{A} \in Wkload} \#cells(\mathbf{A})$. Given an set $\mathbf{A} \in Wkload$, we note that it has $2^{|\mathbf{A}|}$ subsets, so that $|closure(\mathbf{A})| = 2^{|\mathbf{A}|}$. However, $\#cells(\mathbf{A})$ is at least $2^{|\mathbf{A}|}$ (because each attribute has at least 2 attribute values). We also note that $closure(Wkload) = \bigcup_{\mathbf{A} \in Wkload} closure(\mathbf{A})$. Hence

$$|closure(Wkload)| \leq \sum_{\mathbf{A} \in Wkload} |closure(\mathbf{A})| = \sum_{\mathbf{A} \in Wkload} \#cells(\mathbf{A})$$

To analyze the time complexity of symbolically representing the privacy cost, as a linear combination of the $1/\sigma_{\mathbf{A}}^2$ values (for all $\mathbf{A} \in closure(Wkload)$) we note that the coefficient of $1/\sigma_{\mathbf{A}}^2$ is $\prod_{Att_i \in \mathbf{A}} \frac{|Att_i| - 1}{|Att_i|}$. Thus computing the coefficient $1/\sigma_{\emptyset}^2$ takes $O(1)$ time. Then, computing the coefficient of $1/\sigma_{\{Att_i\}}^2$ can be computed from the coefficient of $1/\sigma_{\emptyset}^2$ in $O(1)$ additional time. Thus, we if go level by level, first computing the coefficients of $1/\sigma_{\mathbf{A}}^2$ with $|\mathbf{A}| = 1$ then for $|\mathbf{A}| = 2$, etc. then computing the coefficient for each new $\mathbf{A}$ takes incremental $O(1)$ time. Thus the overall time is $O(|closure(Wkload)|)$ and therefore is $O(\sum_{\mathbf{A} \in Wkload} \#cells(\mathbf{A}))$.

Let $n_{cells} = \sum_{\mathbf{A} \in Wkload} \#cells(\mathbf{A})$ To express the variance symbolically as a linear function of the $\sigma_{\mathbf{A}}^2$ values via Theorem 4.7, we note from the previous part that computing $\prod_{Att_i \in \mathbf{A}'} \frac{|Att_i| - 1}{|Att_i|}$ for all $\mathbf{A}' \in closure(Wkload)$ can be done in total $O(n_{cells})$ time. Similarly, computing $\prod_{Att_i \in \mathbf{A}'} \frac{1}{|Att_i|^2}$ for all $\mathbf{A}' \in closure(Wkload)$ also take total $O(n_{cells})$ time. Once this is pre-computed, then for any $\mathbf{A}' \subseteq \mathbf{A} \in closure(Wkload)$, the product $\prod_{Att_i \in \mathbf{A}'} \frac{|Att_i| - 1}{|Att_i|} * \prod_{Att_j \in (\mathbf{A}/\mathbf{A}')} \frac{1}{|Att_j|^2}$ can be computed in $O(1)$ time since $\mathbf{A} \setminus \mathbf{A}' \in closure(Wkload)$. Now, $Var(\mathbf{A}; \mathcal{M}^*) = \sum_{\mathbf{A}' \subseteq \mathbf{A}} \sigma_{\mathbf{A}'}^2 \prod_{Att_i \in \mathbf{A}'} \frac{|Att_i| - 1}{|Att_i|} * \prod_{Att_j \in (\mathbf{A}/\mathbf{A}')} \frac{1}{|Att_j|^2}$. This is a linear combination of $2^{|\mathbf{A}|}$ terms (one term for each variable $\sigma_{\mathbf{A}'}^2$ for $\mathbf{A}' \subseteq \mathbf{A}$). Each term is computed in $O(1)$ time after the pre-computation phase. Thus the symbolic representation of $Var(\mathbf{A}; \mathcal{M}^*)$ takes $O(2^{|\mathbf{A}|})$ time (which is at most the number of cells in the marginal on $\mathbf{A}$) time after precomputation. Thus computing $Var(\mathbf{A}; \mathcal{M}^*)$ for all $\mathbf{A} \in Wkload$ can be done in total $O(n_{cells})$ time after precomputation, but precomputation also takes $O(n_{cells})$ time. Thus the overall total time is $O(n_{cells})$.

We next analyze the time it takes to generate noisy answers once the true answers have been precomputed (Line 1 in Algorithm 1). This involves (1) computing the product $\mathbf{Hv}$ in the algorithm, (2) generating one Gaussian random variable for each column of $\mathbf{H}$ and (3) computing $\mathbf{Hz}$. Now, the first and third steps take the same amount of time. The second step generates one Gaussian for each row of $\mathbf{H}$ and hence, for each $\mathcal{M}_{\mathbf{A}}$ takes time $\Pi_{Att_i \in \mathbf{A}}(|Att_i| - 1)$.

For the first step, the fast kronecker-product multiplication algorithm (Algorithm 1 of [38]) has the following complexity. Given a kronecker product of $\ell$ matrices of sizes $(m_1 - 1) \times m_1, \ldots, (m_\ell - 1) \times m_\ell$ and a vector with $m_1 \times \cdots \times m_\ell$ components, their algorithm has $\ell$ iterations. In iteration $i$, the $i^{\text{th}}$ matrix (with size $m_{i-1} \times m_i$) is multiplied by a matrix with shape $(m_i, \prod_{j=1}^{i-1} m_j * \prod_{j=i+1}^{\ell} (m_j - 1))$. In our case, each $m_i$ is a subtraction matrix with two nonzero elements in each row. Thus, in each iteration, the product makes $2 \prod_{j=1}^{i-1} m_j * \prod_{j=i}^{\ell} (m_j - 1)$ scalar multiplication operations. There are $\ell$ iterations, so the multiplication algorithm uses $O(\ell \prod_{i=1}^{\ell} m_i)$ multiplications.

Now, to run algorithm $\mathcal{M}_{\mathbf{A}}$, the number of kron products $\ell$ is $|\mathbf{A}|$ and each $m_i$ is $|Att_i|$ for $Att_i \in \mathbf{A}$. Hence the running time of $\mathcal{M}_{\mathbf{A}}$ is $O(|\mathbf{A}| \prod_{Att_i \in \mathbf{A}} |Att_i|)$ which is at most $|\mathbf{A}|$ times the number of cells in the marginal on $\mathbf{A}$. Note that the constant in the big-O notation is bounded across all $\mathbf{A}$. Next, when adding up the complexity across all $\mathbf{A}' \in \text{closure}(\mathbf{A})$, we can replace $|\mathbf{A}'|$ with $|\mathbf{A}|$, and then the summation looks like the product $\prod_{Att_i \in \mathbf{A}} (|Att_i| + 1)$ when this product is expanded. Hence the time to run all $\mathbf{Q}_{\mathbf{A}'}$ for all $\mathbf{A}' \in \text{closure}(\mathbf{A})$ is $O(|\mathbf{A}| \prod_{Att_i \in \mathbf{A}} (|Att_i| + 1))$. Adding up over all $\mathbf{A} \in Wkload$ gets the results.

Next we consider the reconstruction phase. Using the same analysis of the fast kron-product vector multiplication, we see that in each iteration of Algorithm 2, there is a kron product vector multiplication. Using similar reasoning as for the previous item, each such multiplication takes $O(|\mathbf{A}| \prod_{Att_i \in \mathbf{A}}) |Att_i| = O(|\mathbf{A}| \#cells(\mathbf{A}))$ time. The number of iterations in the algorithm is $2^{|\mathbf{A}|} \leq \#cells(\mathbf{A})$. Thus the overall runtime is $O(\sum_{\mathbf{A} \in Wkload} |\mathbf{A}| \#cells(\mathbf{A})^2)$.

Finally, the variance computation is no harder than expressing the $Var(\mathbf{A}; \mathcal{M}^*)$ as linear combinations of the optimization variables and we have shown this to be $O(n_{cells})$. $\qquad\square$

# H  Closed Form Solution to the Weighted Sum of Variances Loss

By Theorem 4.5, the privacy cost is a linear combination of the $1/\sigma_{\mathbf{A}}^2$ values. By Theorem 4.7, each reconstructed marginal's cell variances are a linear combination of the $\sigma_{\mathbf{A}}^2$ values. Thus, minimizing the weighted sum of reconstructed marginal variances subject to the privacy cost being $\leq c$ can be formulated as a problem of the following type:

$$\mathop{\arg\min}_{\sigma_{\mathbf{A}}^2 : \mathbf{A} \in \text{closure}(Wkload)} \sum_{\mathbf{A} \in \text{closure}(Wkload)} v_{\mathbf{A}} \sigma_{\mathbf{A}}^2 \tag{12}$$

$$s.t. \sum_{\mathbf{A} \in \text{closure}(Wkload)} \frac{p_{\mathbf{A}}}{\sigma_{\mathbf{A}}^2} \leq c$$

where the $v_{\mathbf{A}}$ are the linear coefficients of the $\sigma_{\mathbf{A}}^2$ and the $p_{\mathbf{A}}$ are the linear coefficients of the $1/\sigma_{\mathbf{A}}^2$ in the privacy cost. The closed form solution is given by hte following lemma.

LEMMA H.1. *Given the optimization problem in Equation 12 The optimal objective function value is* $T = \left(\sum_{\mathbf{A}} \sqrt{v_{\mathbf{A}} p_{\mathbf{A}}}\right)^2 / c$, *the optimal value of each noise scale parameter is* $\sigma_{\mathbf{A}}^2 = \sqrt{T p_{\mathbf{A}} / (c v_{\mathbf{A}})}$.

*Proof.* Clearly, for the optimal solution, the inequality constraint must be tight (i.e., $= c$) because if it is not tight, we can lower variance while increasing privacy cost by dividing each $\sigma_{\mathbf{A}}^2$ by a number $> 1$. Thus we just need to solve the problem subject to $\sum_{\mathbf{A}} p_{\mathbf{A}} / \sigma_{\mathbf{A}}^2 = c$.

From Cauchy-Schwarz inequality,

$$\sum_{\mathbf{A}} v_{\mathbf{A}} \sigma_{\mathbf{A}}^2 = \left( \sum_{\mathbf{A}} v_{\mathbf{A}} \sigma_{\mathbf{A}}^2 \right) \left( \sum_{\mathbf{A}} \frac{p_{\mathbf{A}}}{\sigma_{\mathbf{A}}^2} \right) / c \geq \left( \sum_{\mathbf{A}} \sqrt{v_{\mathbf{A}} p_{\mathbf{A}}} \right)^2 / c = T$$

Equality holds when $\frac{v_{\mathbf{A}}}{p_{\mathbf{A}}} \sigma_{\mathbf{A}}^4 = t$ for all $\mathbf{A}$ (for some constant $t$). Since $c = \sum_{\mathbf{A}} \frac{p_{\mathbf{A}}}{\sigma_{\mathbf{A}}^2} = \sum_{\mathbf{A}} \sqrt{v_{\mathbf{A}} p_{\mathbf{A}}/t}$, then we must have $t = T/c$. Plugging this into the definition of $t$, we get $\sigma_{\mathbf{A}}^2 = \sqrt{T p_{\mathbf{A}}/(c v_{\mathbf{A}})}$. $\qquad\square$

Thus, if the loss function is the weighted sum of variances, ResidualPlanner does not need any optimization steps. The selection of the noise scales and the reconstruction phase are direct algorithms.

# I  Additional Experiments

In this section, we present additional experiments. Following [37], the experiments use the following type of workloads:

- All $k$-way marginals.
- All $\leq$ 3-way marginals. This includes all 0-way marginal (the total sum), all 1-way marginals, all 2-way marginals, and all 3-way marginals.
- Small marginals. This includes any $k$-way marginal that has at most 5000 cells.

We also use these metrics:

- RMSE: The total variance is the sum of the variances of the reconstructed cells in each marginal in the workload. Root Mean Squared Error is obtained by taking the total variance, dividing by the total number of cells in the workload marginals, then taking the square root. The SVD Bound (SVDB for short) [31] provides a theoretical lower bound on RMSE for any matrix mechanism. For marginals, the SVDB is tight, but its computation is not scalable.
- MaxVar: compute the variance of each reconstructed cell for each marginal in the workload, then take the maximum of these.
- Running time (in seconds) of the different stages of the algorithms (select and reconstruct).

Unless otherwise stated, ResidualPlanner uses the open-source ECOS optimizer [14] for solving the optimization problem it generates for the select step.

For all experiments, we require all mechanisms to have privacy cost $pcost(\mathcal{M}) = 1$. By definition 2.3, $\mathcal{M}$ satisfies $\rho$-zCDP with $\rho = 1/2$ [46] and satisfies $\mu$-Gaussian DP with $\mu = 1$ [15, 46].

Each experiment is repeated 5 times, we report the mean value of these 5 results and a confidence interval consisting of $\pm 2$ standard deviations. This is most useful for running time, as the variance loss metrics have negligible variance across all algorithms.

## I.1  Scalability

In this section, we study the scalability of ResidualPlanner. This is done using the Synth$-n^d$ dataset, where $d$ is the number of attributes and $n$ is the domain size of each attribute. We use all $\leq$ 3-way marginals as a fixed workload and vary $n$ or $d$ to get the computation time for HDMM and ResidualPlanner.

### I.1.1  Varying Attribute Domain Size $n$ in the Selection Step.

This experiment considers what happens when the attribute domain size $n$ get larger. We fix the number of attributes $d = 5$ and vary the domain size $n$ for each attribute, where $n$ ranges from 2 to 1024. We evaluate the running time and accuracy of the selection step

Table 9 shows the running time for the selection step of HDMM and ResidualPlanner. The RMSE on the workload that the selection step guarantees is also measured. Both HDMM and ResidualPlanner

have no trouble here. HDMM is nearly optimal in RMSE and ResidualPlanner is optimal, as shown by agreement with the SVD Bound. ResidualPlanner is faster, but both methods are fast in this experiment setting.

Table 9: Selection step on Synth$-n^d$ dataset where $d = 5$ and $n$ varies. The workload is all $\leq$ 3-way marginals. Metrics are running time and RMSE.

| $n$ | $Time_{HDMM}$ | $Time_{ResPlan}$ | $RMSE_{HDMM}$ | $RMSE_{ResPlan}$ | SVDB |
|---|---|---|---|---|---|
| 2 | $0.069 \pm 0.018$ | $0.001 \pm 0.000$ | 1.903 | 1.890 | 1.890 |
| 4 | $0.064 \pm 0.006$ | $0.001 \pm 0.000$ | 2.685 | 2.681 | 2.681 |
| 8 | $0.070 \pm 0.021$ | $0.001 \pm 0.000$ | 3.156 | 3.156 | 3.156 |
| 16 | $0.076 \pm 0.020$ | $0.001 \pm 0.000$ | 3.367 | 3.366 | 3.366 |
| 32 | $0.105 \pm 0.020$ | $0.001 \pm 0.000$ | 3.422 | 3.423 | 3.423 |
| 64 | $0.114 \pm 0.033$ | $0.001 \pm 0.000$ | 3.408 | 3.407 | 3.407 |
| 128 | $0.137 \pm 0.048$ | $0.001 \pm 0.000$ | 3.371 | 3.367 | 3.367 |
| 256 | $0.187 \pm 0.050$ | $0.001 \pm 0.000$ | 3.331 | 3.322 | 3.322 |
| 512 | $0.183 \pm 0.020$ | $0.001 \pm 0.000$ | 3.294 | 3.283 | 3.283 |
| 1024 | $0.353 \pm 0.058$ | $0.001 \pm 0.000$ | 3.328 | 3.251 | 3.251 |

Table 10 shows the running time and Max Variance comparison for the selection step. HDMM can only optimize for RMSE, not max variance, so this table shows that RMSE is not a good substitute when one needs to optimize for Max Variance.

Table 10: Selection step on Synth$-n^d$ dataset where $d = 5$ and $n$ varies. The workload is all $\leq$ 3-way marginals. Metrics are running time and Max Variance.

| $n$ | $Time_{HDMM}$ | $Time_{ResPlan}$ | $MaxVar_{HDMM}$ | $MaxVar_{ResPlan}$ |
|---|---|---|---|---|
| 2 | $0.069 \pm 0.018$ | $0.008 \pm 0.001$ | 8.091 | 4.148 |
| 4 | $0.064 \pm 0.006$ | $0.008 \pm 0.001$ | 44.693 | 9.760 |
| 8 | $0.070 \pm 0.021$ | $0.008 \pm 0.001$ | 180.343 | 15.643 |
| 16 | $0.076 \pm 0.020$ | $0.008 \pm 0.001$ | 588.115 | 20.067 |
| 32 | $0.105 \pm 0.020$ | $0.008 \pm 0.001$ | 1649.341 | 22.811 |
| 64 | $0.114 \pm 0.033$ | $0.008 \pm 0.001$ | 5560.807 | 24.345 |
| 128 | $0.137 \pm 0.048$ | $0.008 \pm 0.001$ | 12229.480 | 25.157 |
| 256 | $0.187 \pm 0.050$ | $0.008 \pm 0.001$ | 8168.716 | 25.574 |
| 512 | $0.183 \pm 0.020$ | $0.008 \pm 0.001$ | 32159.958 | 25.786 |
| 1024 | $0.353 \pm 0.058$ | $0.008 \pm 0.001$ | 277825.955 | 25.893 |

### I.1.2 Impact of varying the number of attributes in the Selection Step.

Next, we fix the domain size of each attribute to be $n = 10$ and vary the number of attributes $d$, where $d$ ranges from 2 to 200. This experiment can test some of the limits of ResidualPlanner. While HDMM cannot perform selection when the number of attributes is 20 or larger, ResidualPlanner has no trouble optimizing RMSE even for 200 attributes. However, optimizing for Max Variance is much more difficult. ResidualPlanner can do this for $d = 100$ but the underlying optimization took more than 1 hour for $d = 200$ and we killed the process.

Table 11 shows the running time and RMSE comparison for the selection step. The running time of HDMM increases sharply and it quickly runs out of memory. At the same point, the SVD Bound can no longer be computed. Meanwhile, ResidualPlanner continues to run efficiently.

Table 12 shows the running time and Max Variance comparison on the Selection step. Optimizing for Max Variance is much harder for ResidualPlanner compared to RMSE and we killed the process for $d = 200$. Meanwhile, HDMM is not able to run at $d = 20$ (we emphasize again, it optimizes for RMSE even if one cares about Max Variance). There is an interesting phenomenon with HDMM that takes place for $d$ between 8 and 15. In this case, HDMM always produces a max variance of 1000. This maximum is always achieved for the sum query (a zero-dimensional marginal) for the following reason. For $d$ beween 8 and 15, HDMM decides to add noise to all 3-way marginals and nothing else (even though the workload is all $\leq$ 3 marginals). The privacy loss budget is split equally among them. Thus, each of the $\binom{d}{3}$ marginals it measures gets $N(0, \binom{d}{3})$ noise. The sum query gets reconstructed

Table 11: Selection step on Synth$-n^d$ dataset where $n = 10$ and $d$ varies. The workload is all $\leq$ 3-way marginals. Metrics are running time and RMSE.

| $d$ | $Time_{HDMM}$ | $Time_{ResPlan}$ | $RMSE_{HDMM}$ | $RMSE_{ResPlan}$ | SVDB |
|---|---|---|---|---|---|
| 2 | $0.013 \pm 0.003$ | $0.001 \pm 0.0008$ | 1.379 | 1.379 | 1.379 |
| 4 | $0.028 \pm 0.007$ | $0.002 \pm 0.001$ | 2.346 | 2.345 | 2.345 |
| 6 | $0.065 \pm 0.012$ | $0.002 \pm 0.0008$ | 4.278 | 4.275 | 4.275 |
| 8 | $0.167 \pm 0.019$ | $0.004 \pm 0.001$ | 6.726 | 6.638 | 6.638 |
| 10 | $0.639 \pm 0.059$ | $0.009 \pm 0.001$ | 9.629 | 9.348 | 9.348 |
| 12 | $4.702 \pm 0.315$ | $0.015 \pm 0.001$ | 12.904 | 12.359 | 12.359 |
| 14 | $46.054 \pm 12.735$ | $0.025 \pm 0.002$ | 16.506 | 15.642 | 15.642 |
| 15 | $201.485 \pm 13.697$ | $0.030 \pm 0.017$ | 18.421 | 17.378 | 17.378 |
| 20 | Out of memory | $0.079 \pm 0.017$ | Out of memory | 26.916 | Out of memory |
| 30 | Out of memory | $0.247 \pm 0.019$ | Out of memory | 49.713 | Out of memory |
| 50 | Out of memory | $1.207 \pm 0.047$ | Out of memory | 107.258 | Out of memory |
| 100 | Out of memory | $9.913 \pm 0.246$ | Out of memory | 303.216 | Out of memory |
| 200 | Out of memory | $80.120 \pm 1.502$ | Out of memory | 855.330 | Out of memory |

as follows. For any single noisy 3-way marginal, one can estimate the sum by adding up the cells in the marginal. Since each cell has variance $\binom{d}{3}$ and there are $n^3 = 1,000$ cells, the sum estimate from a single 3-way marginal has a variance of $1000\binom{d}{3}$. But one can obtain an independent estimate to the sum query from each of the $\binom{d}{3}$ noisy 3-way marginals. By averaging these noisy estimates, one can obtain an estimate of the sum query with variance $1,000$.

Table 12: Selection step on Synth$-n^d$ dataset where $n = 10$ and $d$ varies. The workload is all $\leq$ 3-way marginals. Metrics are running time and Max Variance.

| $d$ | $Time_{HDMM}$ | $Time_{ResPlan}$ | $MaxVar_{HDMM}$ | $MaxVar_{ResPlan}$ |
|---|---|---|---|---|
| 2 | $0.013 \pm 0.003$ | $0.007 \pm 0.001$ | 13.745 | 3.306 |
| 4 | $0.028 \pm 0.007$ | $0.010 \pm 0.005$ | 132.620 | 10.480 |
| 6 | $0.065 \pm 0.012$ | $0.009 \pm 0.001$ | 461.132 | 26.904 |
| 8 | $0.167 \pm 0.019$ | $0.015 \pm 0.003$ | 1000.000 | 56.961 |
| 10 | $0.639 \pm 0.059$ | $0.018 \pm 0.001$ | 1000.000 | 105.031 |
| 12 | $4.702 \pm 0.315$ | $0.028 \pm 0.001$ | 1000.000 | 175.496 |
| 14 | $46.054 \pm 12.735$ | $0.041 \pm 0.001$ | 1000.000 | 272.738 |
| 15 | $201.485 \pm 13.697$ | $0.050 \pm 0.001$ | 1000.000 | 332.769 |
| 20 | Out of memory | $0.123 \pm 0.023$ | Out of memory | 768.941 |
| 30 | Out of memory | $0.461 \pm 0.024$ | Out of memory | 2540.440 |
| 50 | Out of memory | $4.011 \pm 0.112$ | Out of memory | 11597.037 |
| 100 | Out of memory | $121.224 \pm 3.008$ | Out of memory | 91960.917 |

### I.1.3 Scalability of the Reconstruction Step.

We conduct similar experiments, but now we measure the time in the reconstruction step. To complement the reconstruction scalability experiments from the main paper on the Synth$-n^d$ synthetic dataset, we first fix the number of attributes $d = 5$ and vary the domain size $n$ for each attribute, where $n$ ranges from 2 to 512. The reconstruction time for ResidualPlanner does not depend on the metric that the select step was optimized for. Again we compare with HDMM [38] and a version of HDMM with improved reconstruction scalability called HDMM+PGM [38, 41] (the PGM settings used 50 iterations of its Local-Inference estimator, as the default 1000 was too slow). Table 13 shows the results. Again, at some point HDMM runs out of memory while ResidualPlanner runs efficiently. HDMM runs of out memory because of choices it had made in the selection step. When $n = 128$ it decided to measure a 5-way marginal, which is so large (requiring $128^5$ space) that it caused HDMM and HDMM+PGM to have memory issues.

We next fix $n = 3$ and vary $d$. Table 14 shows ResidualPlanner is clearly faster. Furthermore, HDMM and HDMM+PGM are hampered by the failure of the selection step (when selection fails, there is nothing to reconstruct). It is interesting to compare HDMM+PGM behavior when $n = 3$ in Table

Table 13: Running time (in seconds) of the reconstruction step on Synth$-n^d$ dataset where $d = 5$ and $n$ varies. The workload is all $\leq$ 3-way marginals.

| $n$ | HDMM | HDMM + PGM | ResPlan |
|---|---|---|---|
| 2 | $0.005 \pm 0.002$ | $2.466 \pm 0.278$ | $0.008 \pm 0.002$ |
| 4 | $0.005 \pm 0.000$ | $1.894 \pm 0.146$ | $0.011 \pm 0.008$ |
| 8 | $0.008 \pm 0.000$ | $1.871 \pm 0.122$ | $0.011 \pm 0.008$ |
| 16 | $0.064 \pm 0.036$ | $1.936 \pm 0.131$ | $0.016 \pm 0.001$ |
| 32 | $1.924 \pm 0.060$ | $3.211 \pm 0.220$ | $0.045 \pm 0.007$ |
| 64 | $56.736 \pm 1.460$ | $12.574 \pm 0.512$ | $0.217 \pm 0.021$ |
| 128 | Out of memory | Out of memory | $1.244 \pm 0.059$ |
| 256 | Out of memory | Out of memory | $12.090 \pm 0.504$ |
| 512 | Out of memory | Out of memory | $166.045 \pm 13.803$ |

14 with $n = 10$ in Table 2 from the main paper. Clearly HDMM+PGM is faster for $n = 10$ than $n = 3$. This counterintuitive result can be explained by the complex workings of HDMM as follows. When $n = 3$, the selection step in HDMM returns some 4-way marginals. But when $n = 10$, HDMM only returns $\leq$ 3-way marginals. The 4-way marginals make the reconstruction step harder for both HDMM and HDMM + PGM.

Table 14: **Time for Reconstruction Step in seconds** on Synth$-n^d$ dataset. $n = 3$ and the number of attributes $d$ varies. The workload consists of all marginals on $\leq$ 3 attributes each. Times are reported with $\pm 2$ standard deviations. Reconstruction can only be performed if the select step completed.

| $d$ | HDMM | HDMM + PGM | ResidualPlanner |
|---|---|---|---|
| 2 | $0.001 \pm 0.0001$ | $0.256 \pm 0.030$ | $0.005 \pm 0.002$ |
| 6 | $0.009 \pm 0.001$ | $3.293 \pm 0.253$ | $0.020 \pm 0.004$ |
| 10 | $0.334 \pm 0.010$ | $51.568 \pm 3.391$ | $0.086 \pm 0.004$ |
| 12 | $3.882 \pm 0.101$ | $180.708 \pm 5.437$ | $0.153 \pm 0.002$ |
| 14 | $55.856 \pm 0.361$ | $314.252 \pm 3.991$ | $0.280 \pm 0.072$ |
| 15 | $231.283 \pm 0.554$ | $713.526 \pm 4.957$ | $0.307 \pm 0.005$ |
| 20 | Unavailable (select step failed) | Unavailable (select step failed) | $0.758 \pm 0.023$ |
| 30 | Unavailable (select step failed) | Unavailable (select step failed) | $2.700 \pm 0.200$ |
| 50 | Unavailable (select step failed) | Unavailable (select step failed) | $12.480 \pm 0.208$ |
| 100 | Unavailable (select step failed) | Unavailable (select step failed) | $99.787 \pm 2.113$ |

## I.2   Comparison on Real Datasets.

In this section, we compare RMSE and Max Variance on the real datasets: CPS, Adult, and Loans. The different workloads are 1-way, 2-way, 3-way, 4-way, 5-way marginals, all $\leq$ 3-way marginals, and Small Marginals.

### I.2.1   RMSE Comparisons

We provide an expanded comparison of RMSE on the 3 real datasets from the main paper. Here we add more workloads. Table 15, 16 and 17 show the comparison of RMSE on the CPS, Adult, and Loans datasets respectively.

We notice that ResidualPlanner matches the theoretical SVD Bound while HDMM is slightly worse, but still accurate. We conclude that when optimizing RMSE, the main advantage of ResidualPlanner is superior scalability.

Since ResidualPlanner is optimal, the purpose of the accuracy comparisons is a sanity check. For RMSE, we compare the quality of ResidualPlanner to the theoretically optimal lower bound known as the SVD bound [31] (they match, as shown in Table 18).

Table 15: Comparison of RMSE on CPS(5D) dataset.

| Workload | HDMM | ResPlan | SVDB |
|---|---|---|---|
| 1-way Marginals | 1.756 | 1.744 | 1.744 |
| 2-way Marginals | 2.103 | 2.035 | 2.035 |
| 3-way Marginals | 2.089 | 2.048 | 2.048 |
| 4-way Marginals | 1.648 | 1.627 | 1.627 |
| 5-way Marginals | 1.000 | 1.000 | 1.000 |
| $\leq$ 3-way Marginals | 2.301 | 2.276 | 2.276 |
| Small Marginals | 2.525 | 2.525 | 2.525 |

Table 16: Comparison of RMSE on Adult(14D) dataset.

| Workload | HDMM | ResPlan | SVDB |
|---|---|---|---|
| 1-way Marginals | 3.081 | 3.047 | 3.047 |
| 2-way Marginals | 6.504 | 6.359 | 6.359 |
| 3-way Marginals | 11.529 | 10.515 | 10.515 |
| 4-way Marginals | 16.618 | 14.656 | 14.656 |
| 5-way Marginals | 20.240 | 17.844 | 17.844 |
| $\leq$ 3-way Marginals | 11.555 | 10.665 | 10.665 |
| Small Marginals | 10.006 | 9.945 | 9.945 |

### I.2.2 Max Variance

The next comparison is on optimization for Max Variance. We repeat that HDMM only optimizes for RMSE and this shows that optimizing for RMSE is highly suboptimal when one cares about max variance.

In contrast to RMSE, where the optimization problem generated by ResidualPlanner's selection step can be solved in closed form, for Max Variance, the optimization needs a convex solver. Hence we include comparisons between the open source ECOS [14] optimizer to the commercial Gurobi optimizer [21]. Thus, our results have columns labeled ResidualPlanner+ECOS and ResidualPlanner+Gurobi.

Tables 19, 20 and 21 show the results for the CPS, Adult, and Loans datasets, respectively. There is one item to note about numerical stability. Although Gurobi is generally faster and more numerically stable, the differences do not matter much. Situations where EOCS was worse are highlighted in red. For example, in Table 19 for the CPS dataset, the dataset has only 5 attributes, so a 5-way marginal is basically the entire dataset. The optimal mechanism for 5-way marginals simply adds $N(0, 1)$ noise to each cell and optimizing for RMSE is equal to optimizing Max Variance for this special case. As we see, the Max Variance for ResidualPlanner+ECOS is $1.008$ which is $0.8\%$ worse than optimal. The reason for this is the numerical precision with which ECOS can solve the optimization problem that ResidualPlanner gives it. In general, however, it looks like open source optimizers should work fairly reliably for them to be used in real applications of ResidualPlanner.

Table 17: Comparison of RMSE on Loans(12D) dataset.

| Workload | HDMM | ResPlan | SVDB |
|---|---|---|---|
| 1-way Marginals | 2.903 | 2.875 | 2.875 |
| 2-way Marginals | 5.747 | 5.634 | 5.634 |
| 3-way Marginals | 9.478 | 8.702 | 8.702 |
| 4-way Marginals | 12.537 | 11.267 | 11.267 |
| 5-way Marginals | 14.872 | 12.678 | 12.678 |
| $\leq$ 3-way Marginals | 9.406 | 8.876 | 8.876 |
| Small Marginals | 8.262 | 8.206 | 8.206 |

Table 18: RMSE Comparisons to the theoretical lower bound SVD Bound [31]

| | Adult Dataset | | CPS Dataset | | Loans Dataset | |
|---|---|---|---|---|---|---|
| Workload | ResPlan | SVDB | ResPlan | SVDB | ResPlan | SVDB |
| 1-way Marginals | 3.047 | 3.047 | 1.744 | 1.744 | 2.875 | 2.875 |
| 2-way Marginals | 6.359 | 6.359 | 2.035 | 2.035 | 5.634 | 5.634 |
| 3-way Marginals | 10.515 | 10.515 | 2.048 | 2.048 | 8.702 | 8.702 |
| $\leq$ 3-way Marginals | 10.665 | 10.665 | 2.276 | 2.276 | 8.876 | 8.876 |

Table 19: Comparison of Max Variance on CPS(5D) dataset.

| Workload | HDMM | ResPlan + ECOS | ResPlan + Gurobi |
|---|---|---|---|
| 1-way Marginals | 13.672 | 4.346 | 4.346 |
| 2-way Marginals | 47.741 | 7.897 | 7.897 |
| 3-way Marginals | 71.549 | 7.706 | 7.706 |
| 4-way Marginals | 15.538 | 4.142 | 4.141 |
| 5-way Marginals | 1.000 | 1.008 | 1.000 |
| $\leq$ 3-way Marginals | 415.073 | 13.216 | 13.216 |
| Small Marginals | 223.579 | 11.774 | 11.774 |

Table 20: Comparison of Max Variance on Adult(14D) dataset.

| Workload | HDMM | ResPlan + ECOS | ResPlan + Gurobi |
|---|---|---|---|
| 1-way Marginals | 41.772 | 12.047 | 12.047 |
| 2-way Marginals | 599.843 | 67.802 | 67.802 |
| 3-way Marginals | 5675.238 | 236.843 | 236.843 |
| 4-way Marginals | 26959.322 | 575.213 | 575.213 |
| 5-way Marginals | 79817.002 | 1030.948 | 1030.948 |
| $\leq$ 3-way Marginals | 6677.253 | 253.605 | 253.605 |
| Small Marginals | 2586.980 | 126.902 | 126.902 |

Table 21: Comparison of Max Variance on Loans(12D) dataset.

| Workload | HDMM | ResPlan + ECOS | ResPlan + Gurobi |
|---|---|---|---|
| 1-way Marginals | 33.256 | 10.640 | 10.640 |
| 2-way Marginals | 437.478 | 52.217 | 52.217 |
| 3-way Marginals | 3095.997 | 156.638 | 156.638 |
| 4-way Marginals | 13776.417 | 320.778 | 320.778 |
| 5-way Marginals | 26056.289 | 474.244 | 474.243 |
| $\leq$ 3-way Marginals | 4317.709 | 180.817 | 180.817 |
| Small Marginals | 2330.883 | 89.873 | 89.873 |

