# OpenReview forum: "An Optimal and Scalable Matrix Mechanism for Noisy Marginals under Convex Loss Functions"
_NeurIPS.cc/2023/Conference — NeurIPS 2023 spotlight_

### Official Review · Reviewer_rP9C · 2023-07-05

**Soundness:** 3 good
**Presentation:** 3 good
**Contribution:** 3 good
**Rating:** 7
**Confidence:** 4

**Summary:**

The paper studies a timely problem of improving the scalability of differentially private query releases using a matrix mechanism. The central approach is to carefully design a set of the residual basis for the representation of workload queries, such that the workload queries could be represented as Kronecker products of block matrices with small sizes. This design of basis representation allows subsequent reconstruction steps of the matrix mechanism to be computed in a closed-form, efficient manner. Consequently, this allows a significantly more efficient reconstruction step (from noisy query answers) than prior matrix mechanisms. The efficiency is proved in computation complexity and empirically shown through improved runtime compared to previous matrix mechanisms via extensive experiments. Finally, the authors show that their mechanism easily extends to different objectives (such as a weighted sum of variance and max variance) for query release. They also illustrate the improved empirical error of the proposed mechanism for the max variance objective (which is not optimized by prior matrix mechanisms).


**Minor comments:** Some statements need to be clarified. E.g., in theorem 4.5, what does " is correct" mean? Does it mean the output is an unbiased estimator?

**Strengths:**

- Significantly improved efficiency of the matrix mechanism. The idea of designing a careful set of query representation basis is novel. The reason for efficiency improvements (via more convenient pseudo-inverse matrix computation for Kronecker product of known block matrices) is also explained in detail.
- The proposed mechanism is general and extends to different objectives for private query release (such as max variance). By contrast, prior works mainly focus on the sum of variance objectives. However, the significance of this generalized mechanism needs to be clarified (see question Q2 for more details).

**Weaknesses:**

- The comparison with prior matrix mechanisms sometimes needs more clarity. The improvement and the necessary cost for improvement need more clarification. -- The proposed mechanism seems to sacrifice the accuracy of higher-order marginals for lower-order marginals. Is this an unavoidable cost for improved efficiency? See question Q1 for more details.
- The difficulty and significance of extending the matrix mechanism to more general objectives, such as max variance, need to be clarified. See question Q2 for more details.

**Questions:**

Q1: Theorem 4.7 shows that the variance for the proposed mechanism is larger as the number of attributes for marginal increases. In other words, 3-way marginals have a larger variance than 2-way marginals. Is that correct? If so, how does this compare to the original matrix mechanism (where I suppose the variance for 3-way and 2-way marginals are the same)? Are we sacrificing the accuracy of higher-order marginals for lower-order marginals? Is this unavoidable?

Q2: One of the main contributions of the paper is the investigation of other objectives (besides the sum of variance) for matrix mechanisms in a scalable manner. However, why would optimizing for other objectives be difficult for prior matrix mechanisms? Maybe the authors could clarify more regarding this to help understand the significance of this generalized mechanism for different objectives.

Q3: The optimality statement Theorem 4.4 puzzles me, as it seems only about an optimal number of noisy answers. In that sense, all previous matrix mechanism-based solutions are also optimal. (As they answer the same number of noisy queries as ResidualPlanner in this paper.) In that sense, I wonder whether there is an optimality improvement. Additionally, could the authors comment on the optimality of ResidualPlanner in terms of the optimization objective, e.g., the smallest sum of variance (weighted) or max variance, among all matrix mechanisms?

**Limitations:**

The paper achieves significant scalability improvement for the matrix mechanisms, and uses an interesting idea of optimizing noise added to different marginals for the overall query release objective. However, it needs to be clarified whether the algorithm inevitably generates more noisy answers to higher-order marginals. The authors also adequately discussed the limitation of marginal queries and possible future works of generalizing the mechanisms for other noise distributions in the limitation section.

---

> ### Author Rebuttal · Authors · 2023-08-07
>
> **The optimality statement Theorem 4.4 puzzles me, as it seems only about an optimal number of noisy answers. In that sense, all previous matrix mechanism-based solutions are also optimal. (As they answer the same number of noisy queries as ResidualPlanner in this paper.) In that sense, I wonder whether there is an optimality improvement. Additionally, could the authors comment on the optimality of ResidualPlanner in terms of the optimization objective, e.g., the smallest sum of variance (weighted) or max variance, among all matrix mechanisms?**
>
> Sorry, we were not clear enough in theorem 4.4.
> As we are re-reading Theorem 4.4, we can see the confusion. As we will clarify in the paper,
> 1) The user picks a privacy budget, a workload of marginals and a regular loss function L (e.g.,  weighted sum of squares or max squared error over the workload).
> 2) Based on this, residual planner decides how much noise to add to the different parts of the strategy queries (i.e., it determines the $\sigma_A$ parameters in the base mechanisms).
> 3) Then it reconstructs answers the workload queries. The resulting noisy workload answers are optimal meaning that subject to the privacy constraints, no other matrix mechanism with Gaussian noise can have a lower value of the loss function (e.g., HDMM cannot have a better sum of variances, if that is what the user sets the loss function to be).
>
>
> Regarding the number of noisy answers -- this refers to the number of noisy answers produced in the measurement step (not the reconstruction step). HDMM and prior matrix mechanisms were overparametrized -- they used more noisy answers than necessary, that is why they needed to use least squares postprocessing instead of inversion.
>
>
> **in theorem 4.5, what does " is correct" mean? Does it mean the output is an unbiased estimator?**
>
> Sorry, it was not phrased well. It simply means that the output of the algorithm is the same as if one naively computed $R_A \vec{x} + $Gaussian noise with the given covariance. In the algorithm, the matrix multiplication and generation of correlated noise are performed more quickly than a naive matrix multiply or a scipy multivariate Gaussian library call (in particular, scipy would try to invert the covariance matrix).
>
> **Theorem 4.7 shows that the variance for the proposed mechanism is larger as the number of attributes for marginal increases. In other words, 3-way marginals have a larger variance than 2-way marginals. Is that correct?**
>
> Thank you for considering the implications of our results and this theorem in particular. In short, that is not necessarily the case.
> The variance in Theorem 4.7 is a summation and the summation has two product terms in it. For the second product (i.e., where the $1/|Att_j|^2$ appears), it generally has more terms, causing the product to decrease. At the same time, more things are being added in the overall summation. These two opposing forces mean that, depending on the objective function and desired workload, in some cases the larger marginals have larger variance and
> in some cases they have smaller variance.
>
> **Why would optimizing for other objectives be difficult for prior matrix mechanisms?**
>
> That is a very good question. In theory, one could throw a generic semidefinite solver at the problem, but it would be painfully slow, even for toy problems -- the number of variables is large and computing the objective function takes a lot of time. Prior work took a specific objective function (usually weighted sum of variances) and took advantage of its mathematical properties to speed up the algorithm. For example, HDMM figured out how to compute the sum of variances faster with the help of the kronecker product. These tricks from prior work are very specific to the objective functions they study.

---

> > ### Comment · Reviewer_rP9C · 2023-08-16
> >
> > Thank the authors for the rebuttal, which helped me better understand the paper. I have a follow-up question. The authors mentioned that depending on the objective function and the desired workload, the bound in Theorem 4.7 for large marginals may be better or worse. How does the bound in Theorem 4.7 depend on the objective function (does it implicitly come from $\sigma^2_A$)?

---

> > > ### Author Response · Authors · 2023-08-17
> > > **A worked out example for two scenarios (apologies, this is long):**
> > >
> > > Thanks for the question. Yes, the objective function completely determines the $\sigma^2_A$ parameters and they are the ones that basically determine the marginal variances. To explain this better, we ran two small examples for which the variance calculation can be done by hand in this post (calculations appear at the end).
> > >
> > > The toy dataset has 5 attributes, each with 3 possible values. In this discussion, the variance of a marginal is the largest variance of its cells (all cells within the same marginal have the same variance, so the variance of a marginal  is basically the variance of any cell).
> > >
> > > Example 1: the objective function is to minimize the max variance among **all** marginals while satisfying mu-Gaussian DP with mu=1. In this case:
> > >
> > > All marginals end up with the same variance of 7.594 and the $\sigma^2_A$ parameters are:
> > > -    $\sigma^2_{\emptyset} = 7.594$
> > > -   $\sigma^2_{A} = 10.125$ when $A$ consists of 1 attribute
> > > -   $\sigma^2_{A} = 13.5$ when $A$ consists of 2 attributes
> > > -   $\sigma^2_{A} = 18$ when $A$ consists of 3 attributes
> > > -   $\sigma^2_{A} = 24$ when $A$ consists of 4 attributes
> > > -   $\sigma^2_{A} = 32$ when $A$ consists of all of the attributes
> > >
> > >
> > > Example 2: the objective function is to minimize the weighted max variance among all marginals (i.e., minimize $\max_{m} weight_m * var(m)$). We set the weight of a marginal to be 3 if it is the 5-way marginal and 1 otherwise. This objective function basically says we want more accuracy on the 5-way marginal (it has higher weight). Again the privacy constraint is 1-Gaussian DP.
> > >
> > > In the optimal solution, each cell in the 5-way marginal has variance 2.718, the 4-way marginals have variance 5.528, and the other marginals have variance 8.154. The $\sigma^2_A$ parameters are:
> > >
> > > -   $\sigma^2_{\emptyset} = 8.154$
> > > -   $\sigma^2_{A} = 10.871$ when $A$ consists of 1 attribute
> > > -   $\sigma^2_{A} = 14.495$ when $A$ consists of 2 attributes
> > > -   $\sigma^2_{A} = 19.327$ when $A$ consists of 3 attributes
> > > -   $\sigma^2_{A} = 12.477$ when $A$ consists of 4 attributes
> > > -   $\sigma^2_{A} = 4.159$ when $A$ consists of all of the attributes
> > >
> > >
> > > For the variance calculation, we show this for Example 2.
> > >
> > > For the variance of a one-way marginal, say the marginal on attribute 1 (i.e., $ A= \\{ 1 \\} $), the calculation involves $\sigma^2_{\\{1\\}}$ and $\sigma^2_{\emptyset}$ (because $\emptyset\subseteq\\{1\\}$) and the summation has two terms. The term involving $\sigma^2_{\emptyset}$ is multiplied by 1 (the first product in the variance expression for theorem 4.7, since an empty product is 1) and that is multiplied by 1/9 (the second product, since each attribute has 3 possible values). Meanwhile, the term involving $\sigma^2_{\\{1\\}}$ would be multiplied by 2/3 (first product) and 1 (since the second product is empty). Thus the variance for the marginal on attribute 1 is:
> > > $8.154 * 1 * 1/9 + 10.871 * 2/3 * 1 \approx 8.154$.
> > >
> > > For the variance on the 5-way marginal, it consists of 1 term for $A^\prime=\emptyset$, 5 terms for when $A^\prime$ has 1 attribute since there are 5 such $A^\prime$ (but all the terms have the same value), 10 terms for when $A^\prime$  has 2 attributes (all have the same value), 10 terms for when $A^\prime$ has 3 attributes (again these terms all are equal), 5 terms when $A^\prime$ has 4 attributes, and 1 term for when $A^\prime=\\{1,2,3,4,5\\}$. First, we note that the total number of terms is 32, which is not bad because the number of cells in the 5-way marginal is $3^5$ (i.e., much larger). The expression for the variance is:
> > >
> > > $1 * (8.154 * (2/3)^0 * (1/9)^5)\qquad\qquad\qquad$   (the term  involving $\sigma^2_\emptyset$)
> > >
> > > $+ 5 * (10.871 * (2/3)^1 * (1/9)^4)\qquad\qquad\qquad$   (the sum of the 5 terms having $A^\prime$ with 1 attribute)
> > >
> > > $+ 10 * (14.495 * (2/3)^2 * (1/9)^3)\qquad\qquad\qquad$  (the sum of the 10 terms having $A^\prime$ with 2 attributes, etc.)
> > >
> > > $+ 10 * (19.327 * (2/3)^3 * (1/9)^2)$
> > >
> > > $+ 5 * (12.477 * (2/3)^4 * (1/9)^1)$
> > >
> > > $+ 1 * (4.159 * (2/3)^5 * (1/9)^0)$
> > >
> > > $\approx 2.718$.
> > >
> > > So the main takeaway here is that when evaluating the variance of the 5-way marginal, the 1/9 terms in the products reduce the influence of any $\sigma_{A^\prime}^2$ for which $A^\prime$ has a small number of attributes. Hence the most important value is $\sigma^2_{\\{1,2,3,4,5\\}}$ followed by the $\sigma_{A^\prime}$ for the 5 sets $A^\prime$ that contain 4 attributes, etc.

---

> > > > ### Comment · Reviewer_rP9C · 2023-08-21
> > > >
> > > > Thanks for the detailed clarifications, which addressed my concerns. I have raised the score. The example is very helpful, I suggest the authors add it in the paper or appendix.

---

### Official Review · Reviewer_zYAs · 2023-07-07

**Soundness:** 4 excellent
**Presentation:** 4 excellent
**Contribution:** 4 excellent
**Rating:** 9
**Confidence:** 4

**Summary:**

INTRODUCTION:
This paper presents significant advancements in the matrix mechanism, a technique used for releasing differentially private linear queries or count statistics. Releasing these types of queries, for instance, “count of people living in NYC who are male” (also known as a 2-way marginal query), is a fundamental task in the differential privacy community. The proposed matrix mechanism is considerably more efficient compared to previous approaches.

BASICS OF MATRIX MECHANISM:

In differential privacy, when given a collection of linear queries, a simple approach is to add independent Gaussian noise to each query response. The scale of this noise is tuned to the sensitivity of the queries, which denotes the amount of change in the output corresponding to alterations in individual data points. Informally, this implies that as the number of queries to be released increases, more noise must be added to each query answer to maintain privacy.

ROLE OF MATRIX MECHANISM:

The matrix mechanism is designed to reduce the amount of added noise for achieving differential privacy, aiming for better query estimates while preserving the same level of privacy. This mechanism operates by refactoring the set of queries and constructing a function with lower sensitivity, yet retaining most information from the original query set. This is effective because queries are often correlated, and this refactoring process capitalizes on such correlations. However, matrix mechanisms traditionally face computational inefficiencies in high-dimensional settings, as they require performance-intensive matrix operations.

EXISTING SOLUTION AND ITS LIMITATION:

The High-Dimensional Matrix Mechanism (HDMM) is a notable prior attempt to enhance the scalability of the original matrix mechanism. Despite its significant improvements, HDMM has limitations. Specifically, HDMM struggles to approximate 3-way marginal queries on datasets comprising more than 10 attributes (each with a cardinality of 10).

NEW CONTRIBUTION:

This paper breaks new ground by proposing an efficient and scalable method named 'ResidualPlanner.' This method can estimate marginal queries on datasets with 100 features (each with a cardinality of 10), outperforming the previous HDMM method. The scalability and speed of ResidualPlanner underscore a significant advancement in the application of matrix mechanisms for differential privacy. Moreover, this method handles optimizing over different loss functions. For example, prior work only optimizes over the square error over all queries, whereas this method allows for optimizing over max error.

**Strengths:**

The ResidualPlanner heralds a substantial advancement in the release of linear statistical queries, an important task in the field of differential privacy. Notably, it enhances efficiency by facilitating the release of an exponentially higher number of linear queries across high-dimensional datasets, compared to its predecessor, the HDMM method. This accomplishment bears significant practical implications for the field of differential privacy, given that real-world datasets are often high dimensional.

The innovation behind ResidualPlanner's improved computational performance lies in a novel technique that produces a compact representation of the query matrix, Q. This approach deviates from the traditional matrix mechanism, which tends to be overparameterized and consequently computationally intensive. By avoiding this overparameterization, ResidualPlanner yields improved computational efficiency.


**Weaknesses:**

Certain sections of this paper have a high degree of technicality, which poses readability challenges. Thus, the paper could greatly benefit from providing more intuitive explanations of their techniques. For instance, the definition and context of the Kronecker product are noticeably absent, which detracts from the clarity of the relevant sections.

Additionally, the discourse on Algorithm 1 is notably brief. The paper lacks a clear explanation of the method's purpose and its necessity in the overall context. Providing further elaboration on these aspects would greatly improve the paper's readability and comprehension.


**Questions:**

In line 69, is there a typo in the example Q_{Att_2}?
- Line 130, Typo: “privcy”.

**Limitations:**

Yes

---

> ### Author Rebuttal · Authors · 2023-08-07
>
> **Thus, the paper could greatly benefit from providing more intuitive explanations of their techniques. For instance, the definition and context of the Kronecker product are noticeably absent, which detracts from the clarity of the relevant sections.**
>
> Sorry for the omission, we will add that. In the supplementary material, we also have a run-through (with commentary) of ResidualPlanner on a toy example to illustrate how it works. The NeurIPS tight page restrictions tend to force papers to be terse in the main part but we took great care in preparing the supplementary material (adding intuition, etc.) knowing that, if accepted, it would be accessible with the main paper.
>
> **Additionally, the discourse on Algorithm 1 is notably brief. The paper lacks a clear explanation of the method's purpose and its necessity in the overall context.**
>
> We will add more explanation. It is just there for completeness to show that the measurement step can be performed quickly without expanding the kronecker product and shows how the generate the correlated noise quickly (the general scipy routine for sampling multivariate Gaussian noise must invert the covariance matrix and that is slow; we avoid that).
>
> **In line 69, is there a typo in the example $Q_{Att_2}$?**
>
> Yes, thanks for finding the typos.

---

> > ### Comment · Reviewer_zYAs · 2023-08-16
> >
> > I read the authors rebuttal to all reviews and I believe they have sufficiently addressed all questions. Therefore, I will maintain my score.

---

### Official Review · Reviewer_zbBa · 2023-07-07

**Soundness:** 3 good
**Presentation:** 2 fair
**Contribution:** 3 good
**Rating:** 7
**Confidence:** 3

**Summary:**

The paper studies scalability of matrix mechanism when answering marginal queries. The paper looks at a restricted yet large enough class which they call regular loss function, satisfying three key properties. Scalability of the matrix mechanism is often poor as it requires heavy SVD computation to compute the matrix inverses. This has been studied by several researchers since the inception of matrix mechanism in 2010, and often heuristics are proposed that provides some degree of scalability. The subject of this paper is to put it on a more firm ground.

**Strengths:**

As I discussed above, the matrix mechanism has been often suggested as a mechanism that provides almost optimal accuracy guarantees (in terms of means-squared error). However, one major challenge, that several past work has not adequately addressed and becomes problematic if the problem size grows even in the medium range, is the scalability issue. This is because, without any structure in the queries, finding optimal factorization to be used in the matrix mechanism requires either solving a semidefinite program or solving a generalized regression problem, whose complexity is cubic in the size of the dimension. The scalability issue is to an extent that even considering a 2-marginal on a medium size dataset leads to out-of-memory on really high-powered clusters.

To address these scalability issues, several recent works have proposed using some heuristics, like removing the consistency constraints. This leads to a suboptimal matrix mechanism. This paper suggests the use of a new mechanism that can optimize over a large class of convex objective functions (the ones that are regular) and return solutions that are optimal under Gaussian noise addition. The matrix mechanism consists of designing two matrices: strategy matrix and reconstruction matrix. The authors propose a new strategy matrix which is in the form of a tensor product of $Sub_d$ matrix. This construction only works in full-rank structure, so they perform a Gaussian elimination to get linear independence.

**Weaknesses:**

Before stating the weakness section, I would like to make a disclaimer that I have been unable to read the supplementary material, so if there are some questions that I raised here and are present in the supplementary material, please just state the page number and I will have a look.

1. The authors mention that their result is optimal for the Gaussian mechanism. I am unsure what this statement means. Is it the case that any other mechanism will add Gaussian noise with a larger variance than what they suggest? I am a little skeptical of this statement considering that their strategy matrix is a fixed matrix and then their reconstruction matrix just depends on the inverse of the strategy matrix. This is true for a given strategy matrix, but why does this hold for an arbitrary strategy matrix?

2. They design their algorithm under the assumption that the loss function is convex and regular; however, HDMM is more general. Can we simply not add another constraint in HDMM that takes into account the regularity of the loss function and achieve competitive run time? Is there a fundamental reason why that cannot be done?

3. The error of the matrix mechanism is at least the trace norm of the query matrix (the SVD bound). So, to me, it feels like, for the worst case k-marginal, the mean squared error would scale like $\Omega((d/k)^{2\sqrt{k}})$. I might be making a mistake in my calculation, but translating it to \ell_\infty error, this sounds about right (comparing with the known upper bound result from ICALP 2012 paper making it tight). Is the regularity assumption the one that gets you around this bound?

4. I really would like to see a concrete error bound (at least for some structured marginal queries that satisfy their requirements) to get a feel of their bound. Can the author provide me with one?


> I am currently keeping my score to just below acceptance, but I am highly inclined to move it above the threshold and would definitely do that if the authors satisfactorily answer my questions (and other reviewer's questions)

I revised my score to "Accept" based on the rebuttal and upon revisiting the paper once again post-rebuttal. I thank the authors for clarifying some of my misunderstandings in the first reading of the paper.

**Questions:**

Please see above.

---

> ### Author Rebuttal · Authors · 2023-08-07
>
> **This construction only works in full-rank structure, so they perform a Gaussian elimination to get linear independence.**
>
> This is a confusion due to our writing.  We propose a residual basis (formed by all possible R matrices as defined in Section 4.2). This basis is linearly independent, but not all basis elements are needed for the marginals that a user requests. We simply drop unnecessary bases (no Gaussian elimination is needed) as explained in lines 209-214. This is possible because the residual basis we propose has a useful property -- any set of marginals is compactly represented by a subset of the basis vectors. When a set of marginals is represented in matrix form, the number of linearly independent rows is equal to the number of residual basis vectors that we need.
>
> **What optimality means? ...**
>
> Sorry, we were not clear enough in theorem 4.4.
> As we are re-reading Theorem 4.4, we can see the confusion. As we will clarify in the paper,
> 1) The user picks a privacy budget, a workload of marginals and a regular loss function L (e.g.,  weighted sum of squares or max squared error over the workload).
> 2) Based on this, residual planner decides how much noise to add to the different parts of the strategy queries (i.e., it determines the $\sigma_A$ parameters in the base mechanisms).
> 3) Then it reconstructs answers the workload queries. The resulting noisy workload answers are optimal meaning that subject to the privacy constraints, no other matrix mechanism with Gaussian noise can have a lower value of the loss function (e.g., HDMM cannot have a better sum of variances, if that is what the user sets the loss function to be).
>
> **"their strategy matrix is a fixed matrix"**
>
> Yes, however different strategy query answers get different amounts of noise added. These are the $\sigma^2_A$ parameters of the base mechanisms.
> These parameters depend on the workload marginals
> and the objective function. That turns out to be sufficient for optimality (theorem 4.4) because of the residual basis used by the base mechanisms. The residual basis has an important property -- for any subset of the workload marginals, the rows of the query matrices for those marginals span a linear space; the basis for that linear space is always a subset of the residual basis.
>
> **The loss function is convex and regular; however, HDMM is more general. Can we simply not add another constraint in HDMM that takes into account the regularity of the loss function**
>
> Perhaps we are misreading what is being asked. For marginals answered with Gaussian noise, HDMM is not more general. HDMM can only optimize for the weighted total variance of any given workload of marginals. This is a regular loss function, which means that ResidualPlanner can also optimize for the weighted total variance of any given workload of marginals (i.e., ResidualPlanner can solve the same problem). ResidualPlanner will return the optimal solution. HDMM returns approximate solutions (but can run out of memory on medium sized datasets). However, ResidualPlanner can optimize other loss functions, while HDMM only gives you one choice.
>
> **The error of the matrix mechanism is at least the trace norm of the query matrix (the SVD bound). I really would like to see a concrete error bound (at least for some structured marginal queries that satisfy their requirements) to get a feel of their bound. Can the author provide me with one?**
>
> When the objective function is total squared error on $k$-way marginals or on weighted data cube queries (as defined in the paper that introduced the SVD bound), the SVD bound is tight and therefore ResidualPlanner matches it. Just in case there is any confusion, ResidualPlanner does not restrict the workload marginals one can use.
>
> **So, to me, it feels like, for the worst case k-marginal, the mean squared error would scale like $\Omega ((d/k)^{2\sqrt{k}})$
> . I might be making a mistake in my calculation, but translating it to $\ell_\infty$ error, this sounds about right (comparing with the known upper bound result from ICALP 2012 paper making it tight: (https://arxiv.org/pdf/1205.1758.pdf Faster Algorithms for Privately
> Releasing Marginals). Is the regularity assumption the one that gets you around this bound?**
>
> This refers to "Faster algorithms for privately releasing marginals" by Thaler et al. (and Theorem 1.1 in their section 1.1)? Their performance criteria are not comparable for the following reason. Since they look at additive error for proportions (e.g., the DP estimate of the **fraction** of women in the dataset is within $\alpha$ of the true answer with high probability), then in the language of counts their result is about whether the differentially private **counts** are within $\alpha n$ of the true counts (where $n$ is the number of people in the dataset). Rescaling $\alpha$ by $n$ makes it pop up in their other constants/expressions and so their results have a dependence on $n$. Matrix mechanisms (and so also ResidualPlanner) provide data independent errors (no dependence on $n$). Thus there is no translation between the bounds.

---

> > ### Comment · Reviewer_zbBa · 2023-08-16
> > **Response to the rebuttal**
> >
> > I apologize for being so late in responding to the rebuttal. I really wanted to get back to the paper and cross-verify the response.
> >
> > Thanks once again for clarifying the paper to me, especially, the relation with HDMM. As I said in my initial review, my initial assessment was the preliminary one. I am more confident about the paper and raising my score to Accept.
> >
> > One qualms I had in trying to implement HDMM was that it was not scalable if we went beyond 3-marginal queries.
> >
> > Good work!

---

### Official Review · Reviewer_2C53 · 2023-07-16

**Soundness:** 3 good
**Presentation:** 3 good
**Contribution:** 4 excellent
**Rating:** 7
**Confidence:** 3

**Summary:**

This work presents a scalable and optimal, under some conditions, matrix mechanism, which can compute private marginals.  Releasing private marginals is one of the most fundamental problems in private data analytics, and as mentioned in the paper, an important primitive in synthetic data generation.  This work seems to improve on the existing work in every dimension, from scalability, generalizability, and accuracy.


**Strengths:**

The paper does a great job of presenting the related work and the underlying problem they want to solve.  The results are pretty amazing when compared to HDMM.  ResidualPlanner is more scalable, more accurate, and works for more general loss functions.   The paper has a nice outline that follows the general framework of the matrix mechanism, starting with the select step, followed by measure and reconstruct steps.  One takeaway from the table of results that was not mentioned is that ResidualPlanner also tends to have less variability in its run time


**Weaknesses:**

The paper is very notation heavy and the provided rationale behind the base mechanisms is lacking. I would have liked to have seen more discussion about why the general linear Gaussian Mechanism is necessary and why not just use an identity covariance and matrix B.  Furthermore, I would have liked to have seen more discussion on why we need more general loss functions and what applications they would be needed, in particular are other loss functions important for synthetic data?  I don’t think Table 3 is necessary in the paper, so that could be removed and more discussion could be added regarding the points above.  I like how there is some result on optimality, but I find it to be a little weak, in that it restricts optimality to linear Gaussian mechanisms.  Although this is a very general class of mechanisms, I would have liked to have seen some optimality guarantee for all DP or Gaussian DP mechanisms.  I am curious about the two problems that were presented in equations (1) and (2) where (1) follows the general approach of differential privacy (maximize utility subject to privacy), while (2) seems to follow something closer to “ex-post” privacy from Ligett, Neel, Roth, Waggoner, Wu ’17, where we want to return the best privacy subject to some accuracy constraint.

**Questions:**

Is there some connection between optimality for linear Gaussian mechanisms and optimality for Gaussian DP that would strengthen Theorem 4.4?

Does solving (2) provide some ex-post privacy guarantee, as in Ligett et al. ’17?

In the computational complexity section, it claims that the time complexity does not depend on the universe size, but how can that be if we are looking at all the k-way marginals with k<= 3?  If we keep the marginals the same but then increase the universe size in each dimension, I would expect the time complexity to increase.

Is the time in Tables 1 and 2 in minutes or seconds?

**Limitations:**

The paper points out some limitations.  Another limitation is that this assumes that each user can contribute at most one record (at least for user-level privacy).  How would bias introduced by clipping the number of records per user impact overall utility in this approach since this work is wanting to return unbiased results.

---

> ### Author Rebuttal · Authors · 2023-08-07
>
>
> **Why the general linear Gaussian Mechanism is necessary and why not just use an identity covariance and matrix B?**
>
> Great question. In theory, it is the same, but as a practical matter, if the covariance is the identity, then B will have irrational entries (a general formula for B even seems tricky). Using the approach we took, all matrices are integer-valued or rational, meaning that a secure implementation (avoiding floating point and rounding issues) is simplified. After the submission deadline, we used these properties to figure out how to use the Discrete Gaussian noise to replace the continuous Gaussian noise (the difficulty was in extending it to the rational covariance matrices we use).
>
> **Why we need more general loss functions?**
>
> One example is multiple users, with each user interested in the sum of squared error for the marginals they care about (different users care about different marginals). One could take a max over each user's loss function to create the overall loss function (minimize the worst user's unhappiness). For another perspective, economists criticize the DP community for using error metrics that are too simple. By expanding the list of supported loss functions, the ball is back in their court to quantify the utility they seek from differentially private data.
>
>
> **Is there some connection between optimality for linear Gaussian mechanisms and optimality for Gaussian DP that would strengthen Theorem 4.4?**
>
> In general, Gaussian DP allows data dependent algorithms while matrix mechanisms provide data independent guarantees, so the requested connection for each regular loss function is probably not possible. We feel that linear Gaussian mechanisms have some extremely important practical properties -- even after the reconstruct phase, the noise distribution of each marginal is known exactly (Gaussian with known covariance matrix).  This allows  statisticians to perform valid statistical inference on the resulting data products and is one of the most important criteria for them.
>
> **Does solving (2) provide some ex-post privacy guarantee, as in Ligett et al. ’17?**
>
> Since the accuracy guarantees are data independent (as with all matrix mechanisms), it is not an ex-post guarantee, but rather another *a priori* way of exploring the privacy/utility tradeoff. For example, if you want to add Gaussian noise to a sensitivity 1 query and guarantee variance $\leq \sigma^2$, then the best privacy you can get is $\mu$-Gaussian DP with $\mu=1/\sigma$ (i.e., if you set this as your privacy budget, then you will meet the accuracy constraint).
>
> **In the computational complexity section, it claims that the time complexity does not depend on the universe size**
>
> Sorry, it was not phrased well. We will rephrase it as: "Although the universe size grows exponentially with the number of attributes, the complexity of ResidualPlanner depends directly on quantities that typically grow polynomially (e.g., the number of desired marginals and total number of cells in those marginals)."
>
>
> **Is the time in Tables 1 and 2 in minutes or seconds?**
>
> The time is measured in seconds. This was easy to miss in the long captions so we will add units after every number.
>
> **Notation heavy**
>
> Unfortunately we find this happens when linear algebra is involved. We also felt the paper was notation heavy though the notation seems necessary. We tried to minimize the cognitive burden by using more descriptive notation where possible (e.g., $Att_i$ and the $Sub$ matrices). For this reason, we also included in the supplementary material  both a table of notation and an example  run-through of ResidualPlanner on a toy problem to help readers along.  We will be happy to incorporate additional suggestions.
>
> **Another limitation is that this assumes that each user can contribute at most one record.**
>
> This is a good point that we will add.

---

> > ### Comment · Reviewer_2C53 · 2023-08-17
> >
> > I have reviewed the rebuttal.  Thanks for addressing my questions.  I will maintain my score.

---

### Author Rebuttal · Authors · 2023-08-07

We thank the reviewers for carefully reading the paper and the insightful comments. The questions pointed out where we need additional clarification and explanation. As suggested, these can be added by moving Table 3 to the supplementary material.
Since we have enough space in the comments for each reviewer, we address each review separately.

---

### Decision · Program_Chairs · 2023-09-21

**Decision:**

Accept (spotlight)

**Comment:**

All reviewers agree that this paper provides a substantial advancement that broadens the applicability of the matrix mechanism with significant practical applications. All of the reviewer questions were addressed during the discussion period. I agree that Table 3 should be moved to the supplementary material so that points from the discussion may be included in the main body of the paper.